# The variation and evolution of complete human centromeres

Glennis A. Logsdon[1,14], Allison N. Rozanski[1], Fedor Ryabov[2], Tamara Potapova[3], Valery A. Shepelev[4], Claudia R. Catacchio[5], David Porubsky[1], Yafei Mao[6], DongAhn Yoo[1], Mikko Rautiainen[7,15], Sergey Koren[7], Sergey Nurk[7,16], Julian K. Lucas[8,9], Kendra Hoekzema[1], Katherine M. Munson[1], Jennifer L. Gerton[3], Adam M. Phillippy[7], Mario Ventura[5], Ivan A. Alexandrov[10,11,12] & Evan E. Eichler[1,13] ✉

Human centromeres have been traditionally very difficult to sequence and assemble owing to their repetitive nature and large size[1]. As a result, patterns of human centromeric variation and models for their evolution and function remain incomplete, despite centromeres being among the most rapidly mutating regions[2,3]. Here, using long-read sequencing, we completely sequenced and assembled all centromeres from a second human genome and compared it to the finished reference genome[4,5]. We find that the two sets of centromeres show at least a 4.1-fold increase in single-nucleotide variation when compared with their unique flanks and vary up to 3-fold in size. Moreover, we find that 45.8% of centromeric sequence cannot be reliably aligned using standard methods owing to the emergence of new α-satellite higher-order repeats (HORs). DNA methylation and CENP-A chromatin immunoprecipitation experiments show that 26% of the centromeres differ in their kinetochore position by >500 kb. To understand evolutionary change, we selected six chromosomes and sequenced and assembled 31 orthologous centromeres from the common chimpanzee, orangutan and macaque genomes. Comparative analyses reveal a nearly complete turnover of α-satellite HORs, with characteristic idiosyncratic changes in α-satellite HORs for each species. Phylogenetic reconstruction of human haplotypes supports limited to no recombination between the short (p) and long (q) arms across centromeres and reveals that novel α-satellite HORs share a monophyletic origin, providing a strategy to estimate the rate of saltatory amplification and mutation of human centromeric DNA.

Advances in long-read sequencing technologies and assembly algorithms have now enabled the complete assembly of complex repetitive regions in the human genome, including centromeres[4–8]. In addition to these technological advances, completion of the first human genome was aided by the use of a complete hydatidiform mole (CHM)[4]—an abnormality of development in which only the paternal chromosomal complement is retained. The particular cell line, CHM13, simplified the assembly process because the presence of a single human haplotype eliminated allelic variation that can otherwise complicate the assembly of structurally complex regions[6,9]. This combination of technologies and resources therefore provided the first complete sequence of each centromere from a single human genome[4,5]. Notwithstanding these advances, human centromeres still pose a challenge to sequencing and assembly. In a recent analysis of human genomes sequenced as part of the Human Pangenome Reference Consortium (HPRC), no other human genome was completely sequenced across its centromeres[10]. The centromeres, in particular, were among the most gap-ridden regions[11] and were excluded from the construction of a pangenome[10]. Additional methods and approaches are still required to fully sequence and assemble these regions[12].

Human centromeres are among the most diverse and rapidly evolving regions of the genome[13,14]. The bulk of human centromeric DNA is composed of tandemly repeating, approximately 171 bp α-satellite DNA, which is organized into HOR units that can extend for megabase

[1]Department of Genome Sciences, University of Washington School of Medicine, Seattle, WA, USA. [2]Masters Program in National Research University Higher School of Economics, Moscow, Russia. [3]Stowers Institute for Medical Research, Kansas City, MO, USA. [4]Institute of Molecular Genetics, Moscow, Russia. [5]Department of Biosciences, Biotechnology and Environment, University of Bari Aldo Moro, Bari, Italy. [6]Bio-X Institutes, Key Laboratory for the Genetics of Developmental and Neuropsychiatric Disorders, Ministry of Education, Shanghai Jiao Tong University, Shanghai, China. [7]Genome Informatics Section, Computational and Statistical Genomics Branch, National Human Genome Research Institute, National Institutes of Health, Bethesda, MD, USA. [8]Department of Biomolecular Engineering, University of California, Santa Cruz, Santa Cruz, CA, USA. [9]UC Santa Cruz Genomics Institute, University of California, Santa Cruz, CA, USA. [10]Department of Human Molecular Genetics and Biochemistry, Tel Aviv University, Tel Aviv, Israel. [11]Department of Anatomy and Anthropology, Sackler Faculty of Medicine, Tel Aviv University, Tel Aviv, Israel. [12]Dan David Center for Human Evolution and Biohistory Research, Tel Aviv University, Tel Aviv, Israel. [13]Howard Hughes Medical Institute, University of Washington, Seattle, WA, USA. [14]Present address: Department of Genetics, Epigenetics Institute, Perelman School of Medicine, University of Pennsylvania, Philadelphia, PA, USA. [15]Present address: Institute for Molecular Medicine Finland (FIMM), Helsinki Institute of Life Science (HiLIFE), University of Helsinki, Helsinki, Finland. [16]Present address: Oxford Nanopore Technologies, Oxford, United Kingdom. ✉e-mail: eee@gs.washington.edu

pairs (Mb) of sequence. Centromeres are particularly variable among humans owing to the action of unequal crossing over, concerted evolution and saltatory amplification[12,15]. Thus, a single human genome, such as CHM13, cannot adequately represent human genetic diversity. Although most of the human genome has been examined for allelic variation at the base-pair level, studies of centromeric DNA are far more limited, based on early pulsed-field gels and Southern blots[1,16,17], monomer α-satellite analyses with short reads[18,19], or analyses restricted to select regions or chromosomes[5,20,21]. Here we present a complete set of centromeres from another human genome using a second hydatidiform mole cell line (CHM1)[6,9,22]. We compare two complete sets of human centromeres to establish a baseline for single-nucleotide and structural variation, and we relate these differences to shifts in the site of kinetochore attachment. We also compare the rate of mutational change of centromeric DNA by sequencing select chromosomes from other non-human primate (NHP) species and comparing our findings to finished centromeres from the HPRC[10] and Human Genome Structural Variation Consortium (HGSVC)[23].

## The complete sequence of CHM1 centromeres

To assemble each CHM1 centromere, we used an approach similar to that used for the assembly of the CHM13 centromeres (Supplementary Fig. 1). First, we generated approximately 66-fold sequence coverage of Pacific Biosciences (PacBio) high-fidelity (HiFi) sequencing data and about 98-fold coverage of Oxford Nanopore Technologies (ONT) data from the complete hydatidiform mole cell line CHM1 (Supplementary Table 1). We initially used the whole-genome assembler hifiasm[24] to generate a highly accurate backbone genome assembly. Only four centromeres were contiguously assembled (from chromosomes 2, 7, 19 and 20), with the remaining 19 fragmented into multiple contigs. We resolved the remaining centromeres using singly unique nucleotide k-mers (SUNKs) to barcode the PacBio HiFi contigs, bridging them with ultra-long (>100 kb) ONT reads that share a similar barcode, as described previously[21]. Finally, we improved the base accuracy of the assemblies by replacing the ONT sequences with locally assembled PacBio HiFi contigs, generating complete CHM1 centromere assemblies with an estimated base accuracy >99.9999% (QV > 60; Methods).

Owing to the potential for somatic rearrangement arising during cell culture, especially for centromeric regions[25,26], we carefully assessed the CHM1 cell line for chromosomal rearrangements (Supplementary Figs. 2 and 3 and Supplementary Notes 1 and 2) and validated the integrity and biological relevance of each CHM1 centromere. First, we mapped native long-read sequencing data generated from the CHM1 genome to each centromere assembly and confirmed the integrity of all chromosomes, with two exceptions (Supplementary Figs. 4–8 and Supplementary Note 2). We next applied an algorithm, VerityMap[27], that identifies discordant k-mers between the centromere assemblies and PacBio HiFi reads and found no evidence of discordance (Methods). Third, we used a method, GAVISUNK[28], that compares SUNKs in the centromere assemblies to those in the ONT reads generated from the same sample and observed support for each SUNK with orthogonal ONT data (Supplementary Figs. 9–11). Fourth, we compared the sequence of each CHM1 centromere assembly to those generated by an independent assembler, Verkko[29], and found that they were highly concordant, with greater than 99.99% sequence identity between each pair (Supplementary Figs. 12 and 13). Finally, we compared both the CHM1 and CHM13 genomes directly to 56 genomes (112 haplotypes) sequenced as part of the HPRC[10] and HGSVC[23]. While many of these additional human genomes are not yet completely assembled across the centromeres, 20.9% of human haplotypes match ≥99% to the newly assembled centromeric regions (Supplementary Table 2, Extended Data Fig. 1 and Supplementary Figs. 14 and 15). In fact, we found that 46.9% of these haplotypes are a better match to CHM1 than to CHM13 (Methods and Supplementary Table 2). Although the data support the

biological relevance of CHM1 centromeres (similar to the T2T-CHM13 centromeres), both genomes are aberrations of normal development followed by cell culture propagation. Thus, caution should be taken until all structures and configurations have been confirmed in additional human samples.

## Genetic variation among human centromeres

The complete assembly of each CHM1 centromere enables, in principle, a comprehensive comparison of centromeric allelic sequence and structure between two human genomes (Fig. 1). In light of the considerable variation between centromeres and the challenge in creating optimal alignments (especially among α-satellite HORs), we analysed the blocks of monomeric α-satellite DNA in the pericentromere separately from the α-satellite HOR arrays, and we considered three different alignment strategies, including one designed to specifically handle variation in tandem repeats[30] (Methods). We initially compared the centromeres from the CHM1 and CHM13 genomes and then extended our analysis to both complete and incompletely sequenced centromeres from 56 human genomes (Supplementary Tables 3–6). Comparison of the CHM1 and CHM13 centromeres revealed that 63.0–71.5% of α-satellite HORs (depending on the chromosome) could be reliably aligned (that is, greater than 90% identity; Supplementary Table 3). Extending this analysis to those from 56 diverse human genomes from the HPRC and HGSVC, we found that this drops to 53.2–55.3% (Supplementary Table 6), underscoring the considerable variation in these genomes and the emergence of new α-satellite HOR structures in some human haplotypes but not others. For the portions that could be aligned, the results were comparable among the three methods (Supplementary Table 3), and we report the full contig alignment statistics with respect to single-nucleotide variation below (Methods).

In comparing the CHM1 and CHM13 centromeres to each other, we found that sequence identity increases as we transition from heterochromatin to euchromatin. For example, the mean sequence identity for the alignable portions of the CHM1 and CHM13 α-satellite HOR arrays is 98.6 ± 1.6%, in contrast to monomeric/diverged α-satellites at 99.8 ± 0.4% and other pericentromeric satellite DNA (β-satellite, γ-satellite and human satellites) at 99.1 ± 1.5% (Extended Data Fig. 2 and Supplementary Table 4). Extending further into the non-satellite pericentromeric DNA, the sequence identity begins to approximate rates of allelic variation corresponding to the euchromatic portions of the genome (99.9 ± 0.3%; Extended Data Fig. 2 and Supplementary Table 4). However, we note that this varies considerably depending on the chromosome (Fig. 2a, Extended Data Fig. 3 and Supplementary Figs. 16 and 17), and the presence of imperfectly aligned α-satellite repeats further complicates such calculations. The centromeres of some chromosomes, such as 19 and X, show the highest degree of concordance between their α-satellite HOR arrays, whereas all others show greater divergence in both sequence identity and structure (Fig. 2a, Extended Data Fig. 3 and Supplementary Figs. 16 and 17). A comparison of the chromosome 5 D5Z2 α-satellite HOR array, for example, reveals tracts that have as much as 4% sequence divergence, with clear expansions of α-satellite HORs in the CHM1 α-satellite HOR array (Fig. 2a).

Comparison with 56 incompletely assembled HPRC/HGSVC reference genomes[10,23] generally confirms that this wide variance in sequence identity is a chromosome-specific property (Extended Data Fig. 4 and Supplementary Figs. 18 and 19). Whereas most α-satellite HOR arrays share at least 97% sequence identity, chromosomes 1, 5, 10, 12, 13 and 19 represent clear outliers, with 16.6% of α-satellite HOR arrays aligning very divergently (<97% sequence identity; Extended Data Fig. 4 and Supplementary Figs. 18 and 19). Importantly, neither set of fully resolved human centromeres is a better match for the majority of HPRC/HGSVC genomes, nor does either adequately capture the full extent of human genetic diversity (Supplementary Figs. 20–42). For example, the mean sequence identity among the 56 HPRC/HGSVC genomes to either CHM1

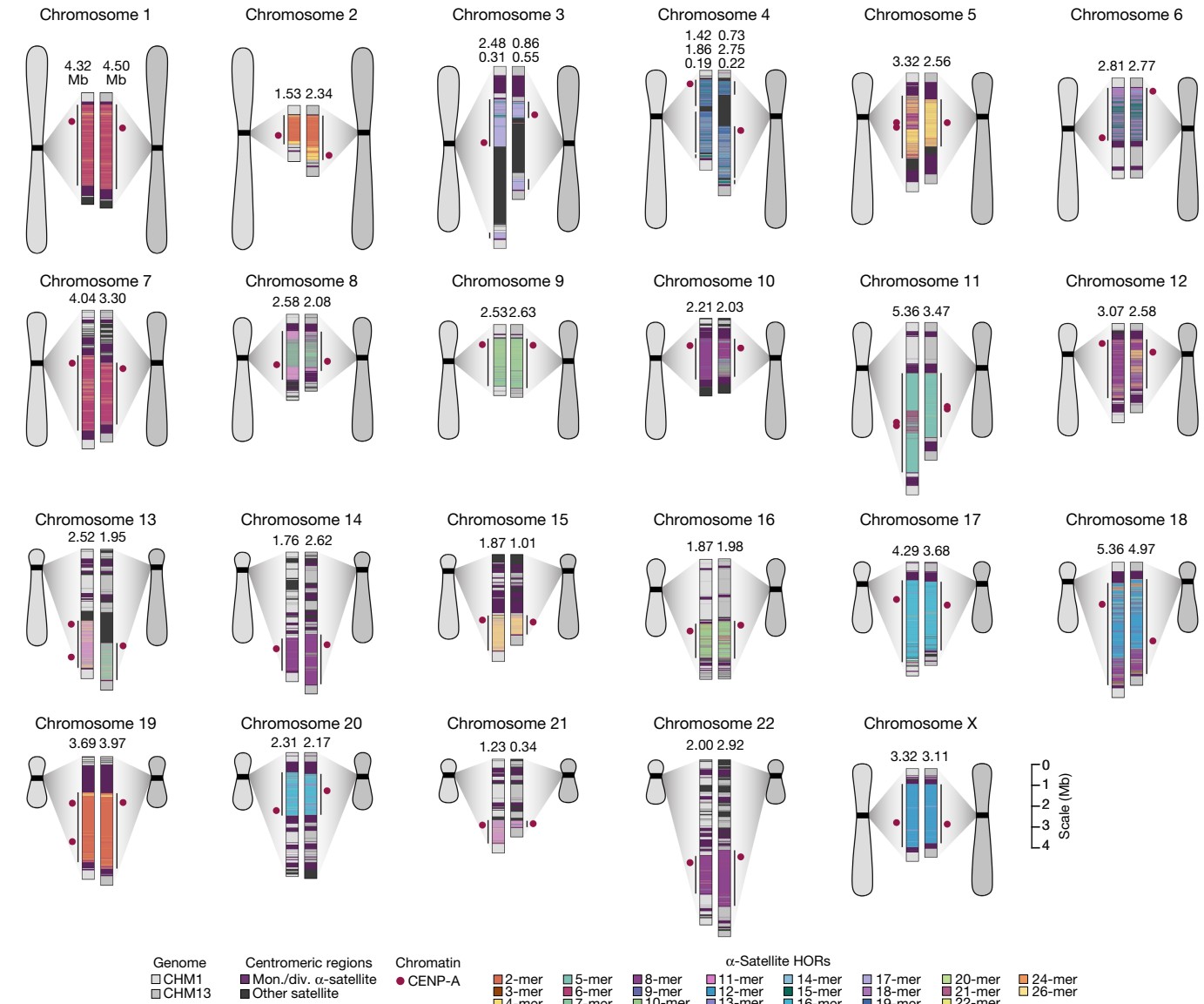

**Fig. 1 | Overview of the centromeric genetic and epigenetic variation between two human genomes.** Complete assembly of centromeres from two hydatidiform moles, CHM1 and CHM13, reveals both small- and large-scale variation in centromere sequence, structure and epigenetic landscape. The CHM1 and CHM13 centromeres are shown on the left and right, respectively, between each pair of chromosomes. The length (in Mb) of the α-satellite higher-order repeat (HOR) array(s) is indicated, and the location of centromeric chromatin, marked by the presence of the histone H3 variant CENP-A, is indicated by a dark red circle. Transposable elements that are polymorphic in these regions are shown in Supplementary Fig. 73. Mon./div., monomeric/diverged.

or CHM13 is 98.0 ± 2.3% (Supplementary Table 6). Similarly, we find that 11 centromeres are a better match to CHM1, while 12 are a better match to CHM13 (Supplementary Table 2). However, if we require that more than 75% of all HPRC haplotypes match better to either CHM1 or CHM13, only five centromeres meet this requirement for CHM1 (chromosomes 2, 12, 13, 19 and 22), while seven do for CHM13 (chromosomes 3, 4, 7, 10, 11, 14 and 15; Supplementary Table 2). These analyses reflect an extraordinary degree of single-nucleotide and structural diversity of human centromeres.

Comparison of the length of the α-satellite HOR arrays reveals that CHM1 arrays are around 1.3-fold larger, on average, than their CHM13 counterparts, with 16 out of 23 chromosomes containing a larger array in CHM1 than in CHM13 (Figs. 2b and 3a and Supplementary Table 7). Of these, five arrays are more than 1.5-fold larger in CHM1 than in CHM13 (chromosomes 3, 4, 11, 15 and 21), with the greatest variation in length occurring on chromosome 21 (3.6-fold; Fig. 3a). This variation in length between CHM1 and CHM13 α-satellite HOR arrays falls within the normal range of variation (1.7-fold to 79.7-fold; median, 2.3-fold), based on released haplotype-phased genome assemblies from the HPRC[10] and HGSVC[23] (Fig. 2c). Our analysis shows, for example, that human α-satellite HOR arrays range in size from 0.03 Mb on chromosome 4 to 6.5 Mb on chromosome 11. Chromosomes 3, 4 and 21 represent some of the smallest α-satellite HOR arrays and show the greatest variation in length among human haplotypes (Fig. 2b; 13.3-fold, 19.0-fold and 9.0-fold difference, respectively). Almost all of the large-scale structural variation is due to variation in α-satellite HOR array organization and size, although the patterns are considerably more complex than simple insertion, deletion or inversion processes.

Comparison of the CHM1 and CHM13 centromeres identifies eight with distinctly different α-satellite HOR array structures (chromosomes 5, 7, 8 and 10–14; Fig. 3b,c and Supplementary Fig. 43). This includes four arrays with a high abundance of previously uncharacterized α-satellite HORs (chromosomes 5, 7, 10 and 14; Supplementary Fig. 44 and Supplementary Table 8). The centromeric *D5Z2* α-satellite

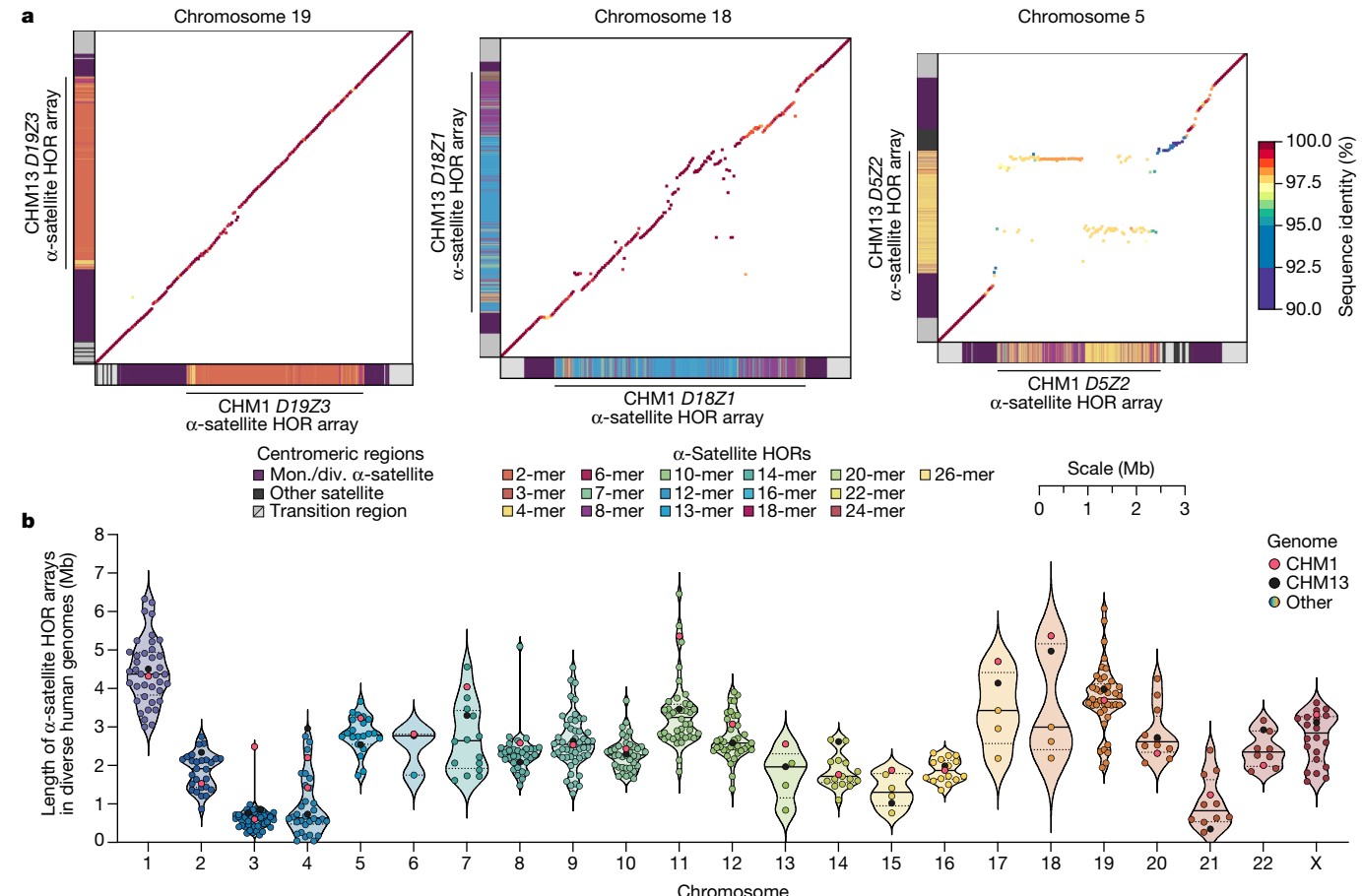

**Fig. 2 | The variation in sequence and structure between two sets of human centromeres. a**, The allelic variation between CHM1 and CHM13 centromeric/pericentromeric haplotypes. Diagonal lines are coloured according to per cent sequence identity. The α-satellite HOR array structure is shown on the axes, along with the organization of each centromeric/pericentromeric region. **b**, The length of the active α-satellite HOR arrays among the CHM1 (red), CHM13 (black) and complete HPRC/HGSVC (various colours) centromeres. $n = 626$. The α-satellite HOR arrays range in size from 0.03 Mb on chromosome 4 to 6.5 Mb on chromosome 11. Data are mean (solid black bar) and 25% and 75% quartiles (dotted black bars).

array from CHM1 chromosome 5, for example, is significantly more diverse, containing two novel α-satellite HOR variants that are four and six α-satellite monomers in length (Fig. 3b and Supplementary Fig. 44a). Phylogenetic and comparative analysis of these HOR variants reveals that they are both derivatives of an ancestral ten-monomer α-satellite HOR, which resides at the edge of the *D5Z2* α-satellite HOR array. These novel HORs, confirmed by analysis of the HPRC genomes[10], probably arose from repeated deletions of α-satellite monomers in the ancestral HOR, giving rise to novel four- and six-monomer HOR variants (Supplementary Fig. 44a). Moreover, specific α-satellite HORs appear to be more consolidated, forming distinct evolutionary layers that are not as apparent or are completely absent in the other haplotype. A clear 870 kb evolutionary layer, for example, is apparent in the CHM1 chromosome 5 centromeric *D5Z2* α-satellite HOR array, and it corresponds to a cluster of highly identical eight-monomer α-satellite HORs (Fig. 3b). This evolutionary layer is absent from the CHM13 centromere, of which the eight-monomer α-satellite HORs are more dispersed along with four-monomer HORs. Similarly, the CHM1 chromosome 11 *D11Z1* centromere evolved a 1.2 Mb layer in the core of its α-satellite HOR array that is missing from the CHM13 centromere (Fig. 3c). This novel layer is composed of six-monomer α-satellite HORs that are found only rarely in the CHM13 centromere. We observed new evolutionary layers in the CHM1 chromosome 10, 12 and 13 α-satellite HOR arrays, all of which have divergent array structures. The remaining centromeres have a similar number of evolutionary layers between the two genomes, ranging from two

to six, with the majority having four (Extended Data Fig. 5 and Supplementary Figs. 45–67).

## Epigenetic differences among centromeres

The kinetochore is a proteinaceous complex marked by the presence of nucleosomes containing the histone H3 variant CENP-A, which is critical for both meiotic and mitotic segregation of chromosomes. Previous studies have shown that the kinetochore typically resides within a region of hypomethylated DNA, named the centromere dip region (CDR)[5,31], that colocalizes with CENP-A immunostaining[21]. We assessed the DNA methylation pattern and CENP-A chromatin organization of each CHM1 centromere and compared it with its CHM13 counterpart. Although CHM1 centromeric α-satellite HOR arrays are typically larger, the majority of CHM1 kinetochore sites (18 out of 23) are smaller than their CHM13 counterparts, with an average size of 178 kb versus 214 kb, respectively (Fig. 4a and Supplementary Table 7). Moreover, 16 out of 23 CHM1 kinetochore sites are located further than 100 kb away from their corresponding location in the CHM13 centromere, with six located further than 500 kb away (chromosomes 4, 6, 11, 12, 18 and 20), when measuring the distance from the α-satellite HOR-to-monomeric transition region (Fig. 4b and Supplementary Table 7). Consistent with earlier observations[5], we identified five chromosomes with evidence of two kinetochore sites, separated by >150 kb (chromosomes 1, 2, 13, 17 and 19). In the case of chromosomes 13 and 19, the two distinct kinetochore sites are located more than 1 Mb apart from each other (Extended Data Fig. 6).

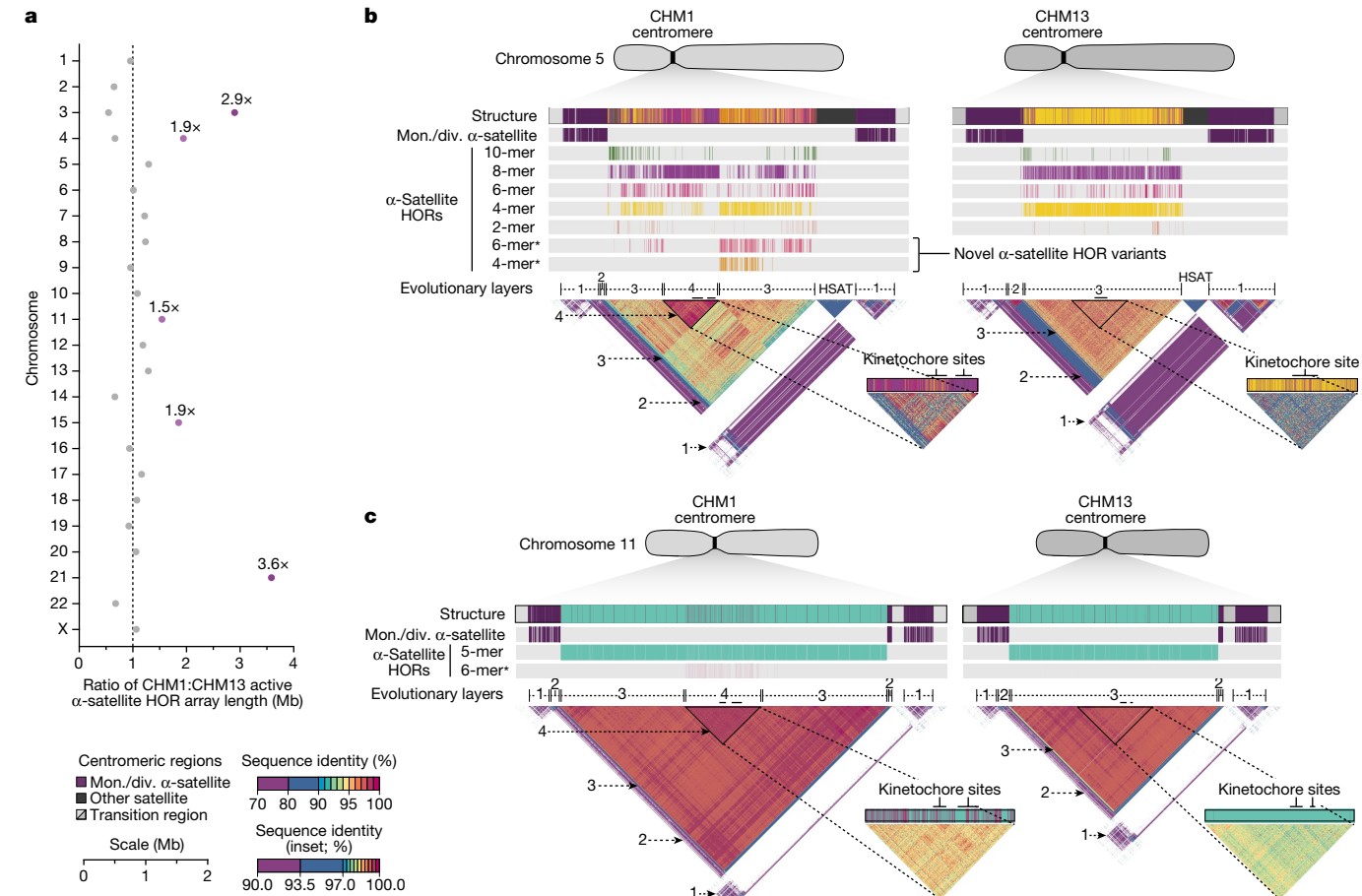

**Fig. 3 | Variation in the length and sequence composition of human centromeric α-satellite HOR arrays. a**, Ratio of the length of the active α-satellite HOR arrays in the CHM1 genome compared with those in the CHM13 genome. **b,c**, Comparison of the CHM1 and CHM13 chromosome 5 *D5Z2* α-satellite HOR arrays (**b**) and CHM1 and CHM13 chromosome 11 *D11Z1* α-satellite HOR arrays (**c**). The CHM1 chromosome 5 *D5Z2* array contains two novel α-satellite HOR variants (Supplementary Fig. 44a) as well as a new evolutionary layer (layer 4; indicated by an arrow), which is absent from the CHM13 array. Similarly, the CHM1 chromosome 11 *D11Z1* α-satellite HOR array contains a six-monomer HOR variant that is much more abundant than in the CHM13 array

and comprises a new evolutionary layer, or a stretch of sequence that has evolved separately from neighbouring sequences (layer 4; indicated with an arrow), although this 1.21 Mb segment is more highly identical to the flanking sequence. The inset shows each of the new evolutionary layers with a higher stringency of sequence identity, as well as the relative position of the kinetochore. Notably, the α-satellite HOR variants comprising the new evolutionary layers in both CHM1 chromosomes 5 and 11 have divergent CpG methylation patterns despite their identical structure (Supplementary Fig. 74). Asterisk, α-satellite HORs variants that are either novel or present in higher abundance in the CHM1 centromere relative to the CHM13 centromere.

To test whether these two kinetochore sites represent two distinct cell populations or, alternatively, an early-stage somatic mutational event resulting in two kinetochores on the same chromosome, we performed immunostaining combined with fluorescence in situ hybridization (immuno-FISH) analysis of stretched CHM1 metaphase chromosome spreads. We found that the chromosome 13 centromere has a single kinetochore, marked by the inner-kinetochore protein CENP-C, within the *D13Z2* α-satellite HOR array, while the chromosome 19 centromere has two kinetochores within the *D19Z3* α-satellite HOR array (Extended Data Fig. 7). Assessment of the underlying sequence and structure of the chromosome 13 *D13Z2* α-satellite HOR array reveals a 631 kb deletion in approximately half of CHM1 cells (Supplementary Fig. 5c and Supplementary Note 2), which may have contributed to the repositioning of the kinetochore in a subpopulation of cells, whereas the chromosome 19 centromere has no such deletion and may have had two kinetochores present from the first few cell divisions. Centromeres with two kinetochores (known as dicentrics) have been previously observed in humans and other species and have been shown to be viable, even with interkinetochore distances of up to 12 Mb (refs. 32,33).

The chromosome 6 centromere shows the greatest variation in kinetochore position, with a difference of 2.4 Mb between the two

haplotypes. This change spans 87–88% of the length of the α-satellite HOR array itself and coincides with an alteration in the underlying α-satellite HOR sequence and structure, switching from a mixture of 16- and 18-monomer α-satellite HORs to a mixture of 15- and 18-monomer HORs (Fig. 4c). Given the complete sequence of CHM1 and CHM13 centromeres and the availability of incomplete assemblies from 56 diverse human genomes, we assessed whether the sequences underlying the kinetochore were more likely to be conserved compared with α-satellite HORs that were not associated with the kinetochore. While we observed clear examples of sequence conservation underlying the kinetochore for specific chromosomes involving both CHM1 and CHM13 (for example, chromosomes 4, 5, 7, 12, 13, 16 and 18), other kinetochore regions appeared to be more similar (chromosomes 1–3, 6, 8, 9, 17, 20, 21 and X) or more divergent (chromosomes 10, 11, 14, 15, 19 and 22) than other portions of the α-satellite HOR array (Supplementary Figs. 20–42).

## Diverse evolutionary trajectories

Our analyses (Figs. 1–4) revealed that human centromeres vary non-uniformly depending on the chromosome. In particular, specific human chromosomes show either highly variable α-satellite HOR array

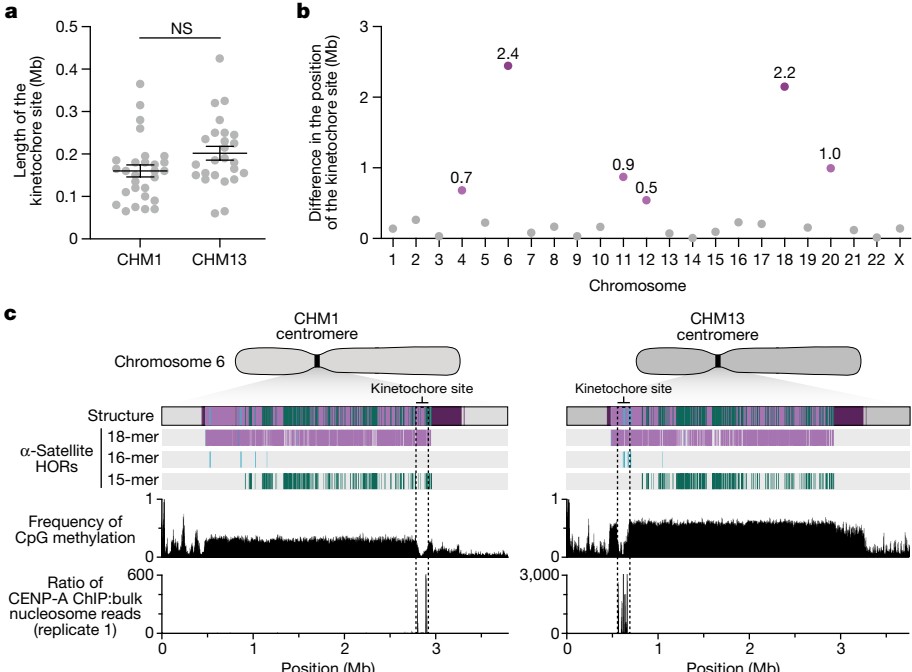

**Fig. 4 | Variation at the site of the kinetochore among two sets of human centromeres. a**, Comparison of the length of the kinetochore site, marked by hypomethylated DNA and CENP-A-containing chromatin, between the CHM1 and CHM13 centromeres. $n$ = 28 and 25 kinetochore sites for the CHM1 and CHM13 centromeres, respectively. Data are mean ± s.e.m. Statistical analysis was performed using a two-sided Kolmogorov–Smirnov test; NS, not significant. **b**, The difference in the position of the kinetochore among the CHM1 and CHM13 centromeres. **c**, Comparison of the CHM1 and CHM13 chromosome 6 centromeres, which differ in kinetochore position by 2.4 Mb.

lengths (such as chromosome 21), diverse α-satellite HOR organizations (such as chromosomes 5, 10 and 12) or divergent epigenetic landscapes (such as chromosome 20). By contrast, the X chromosome is among the most conserved, with nearly identical sequences and structures among diverse human genomes (Supplementary Fig. 42). These findings imply that centromeres may have different mutation rates and diverse evolutionary trajectories that shape their variation. To test this hypothesis, we sequenced and assembled orthologous centromeres from four primate species, focusing on the completion of these six centromeres, in an effort to reconstruct their evolutionary history over a 25-million-year window of primate evolution. Each was specifically selected because it represents different forms of centromeric diversity, but additional analyses, such as sampling all centromeres across multiple individuals, will need to be performed to fully assess the complete diversity. To assemble these centromeres, we first generated PacBio HiFi data (38- to 100-fold coverage) from diploid human, chimpanzee, orangutan and macaque genomes (Methods), producing whole-genome assemblies ranging from 6.1 to 6.3 Gb in size (Supplementary Table 1). Using ultra-long ONT data (14- to 20-fold coverage), we then ordered, oriented and joined the PacBio HiFi contigs together from each centromere, creating 31 contiguous assemblies of primate centromeres for these six chromosomes (Fig. 5). Mapping of long-read sequencing data to each centromere showed uniform coverage, indicating a lack of large structural errors and validating the overall organization (Supplementary Figs. 68–71). With the exception of the X chromosome from a male chimpanzee, both haplotypes were completely sequenced for each diploid female sample, providing additional insights into their overall organization and variation (Fig. 5).

Comparative analysis of these six sets of NHP centromeres revealed, as expected[2,18,19,34], diverse α-satellite HOR array organization and structures, with arrays varying in size by more than 18.6-fold (the smallest residing on human chromosome 21, and the largest residing on macaque chromosome 20). Distinct species-specific differences also became apparent during this analysis (Fig. 5). For example, we estimate that common chimpanzee α-satellite HOR arrays are, on average, 67.8% the size of their human counterparts—a reduction observed in both chimpanzee haplotypes. Like humans, chimpanzee α-satellite HOR arrays show evidence of clear evolutionary layers, with the pairwise sequence identity of these layers dropping as they move toward pericentromeric DNA. This layered α-satellite HOR organization consists mainly of a single, continuous block of higher-order α-satellite repeats that are >95% identical to each other, except for on chromosomes 12 and 20, which have two or three discrete blocks of α-satellite HORs that are only 90–95% identical to each other. By contrast, orangutan centromere organization differs radically from either human or chimpanzee. We found that orangutan α-satellite HOR arrays are composed of three to four distinct blocks of α-satellite HORs that are only 80–90% identical to each other, creating a mosaic of independent HOR expansions with a patchwork-quilt pattern based on sequence identity (Fig. 5). Finally, macaque centromeric α-satellite arrays are substantially larger in size, with an average length of 12.2 ± 1.6 Mb. In contrast to apes, which possess complex HOR structures, macaque centromeric arrays are composed of dimeric α-satellite units[18,35] that are 93–97% identical across all centromeres.

Assessment of the α-satellite suprachromosomal families (SFs), which are classes of α-satellite HORs defined by their monomer type and homology[36], revealed three unexpected findings. First, we identified an African ape centromere that is primarily composed of SF5 α-satellite repeats: the chimpanzee chromosome 5 centromere. While all human and chimpanzee α-satellite arrays are mainly composed of α-satellites from SF1–3 with the exception of the Y chromosome (SF4)[37], we found that both chimpanzee chromosome 5 centromeres harbour SF5 α-satellite. Second, we found that all four chimpanzee chromosome 20 and 21 α-satellite HOR arrays are composed of SF1 α-satellite, as opposed to SF2 as in human. Third, we found multiple examples of large-scale inversion polymorphisms across α-satellite arrays, such as one of two orangutan chromosome 20 centromeres, which contains a

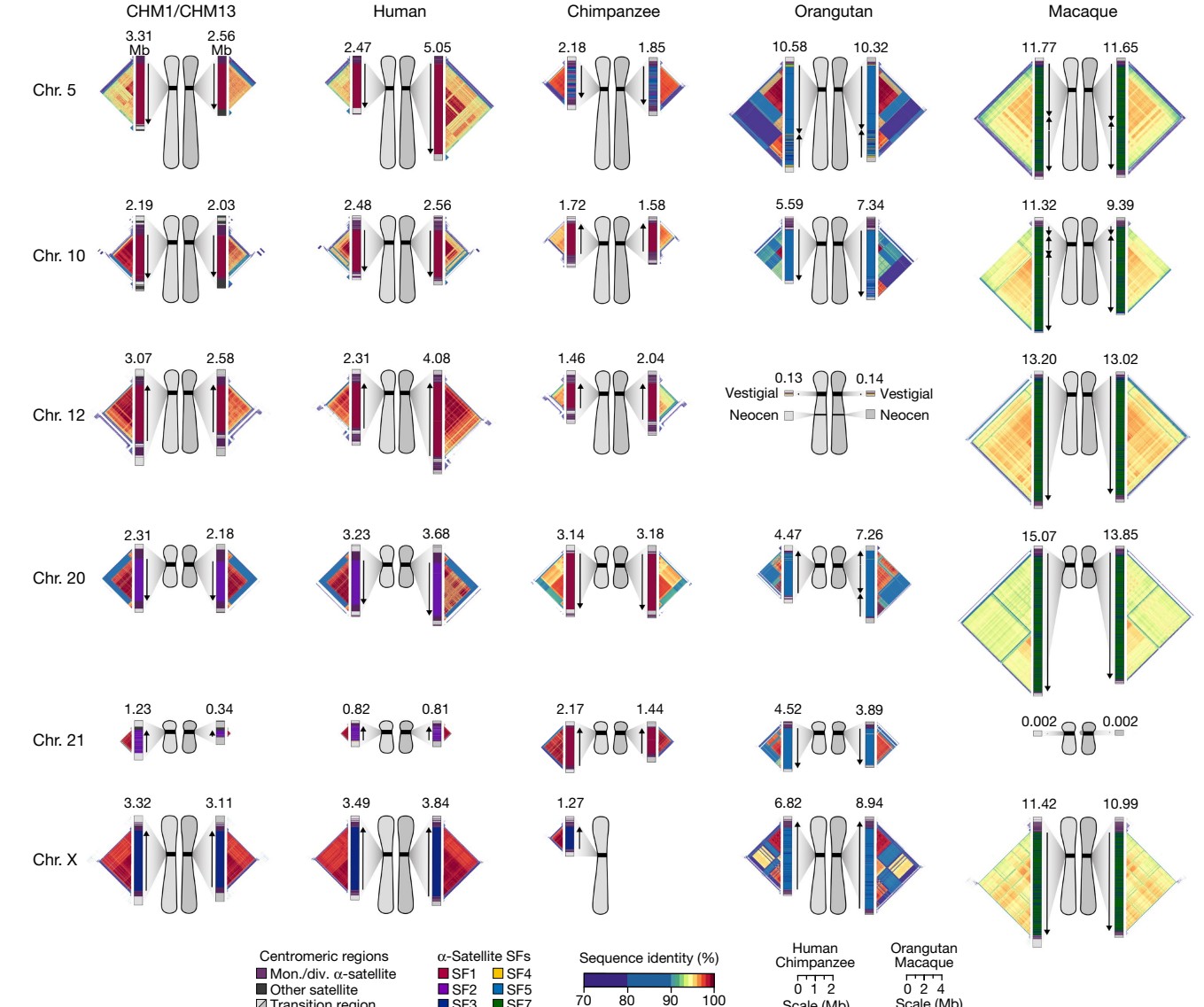

**Fig. 5 | Sequence and structure of six sets of centromeres from diverse primate species.** Complete assembly of centromeres from chromosomes 5, 10, 12, 20, 21 and X in human, chimpanzee, orangutan and macaque reveals diverse α-satellite SF organization and evolutionary landscapes. Sequence identity maps generated using StainedGlass[44] are shown for each centromere (Methods and Supplementary Figs. 75–80), with the size of the α-satellite higher-order (human, chimpanzee and orangutan) or dimeric (macaque) repeat array indicated in Mb. The α-satellite SF for each centromeric array is indicated (vertical bar colour), with arrows illustrating the orientation of the repeats within the array. Chromosome 12 in orangutan has a neocentromere, while the chromosome 21 centromere in macaque is no longer active due to a chromosomal fusion event in that lineage[45]. All chromosomes are labelled according to the human phylogenetic group nomenclature[46]. The human diploid genome used as a control (second column) is HG00733—a 1000 Genomes sample of Puerto Rican origin. Note that the orangutan and macaque centromeres are drawn at half the scale with respect to the other apes.

large 3.2 Mb inversion (Supplementary Fig. 72), and all four macaque chromosome 5 and 10 centromeres.

Despite these species-specific patterns, a common feature of all primate centromeres is the presence of two to five distinct evolutionary layers, marked by the most highly identical α-satellite sequences at the centre of the α-satellite array that become increasingly divergent towards the periphery. These more divergent higher-order and dimeric repeats are flanked by blocks of monomeric α-satellite DNA. We performed phylogenetic and comparative analyses of all six complete orthologous centromere sets and observed that monomeric α-satellite is generally more closely related to the Old World monkey dimeric satellites of macaques. Notwithstanding this general topology, distinct chromosome-specific patterns emerge (Fig. 6 and Extended Data Fig. 8). The chromosome 5 centromere, for example, has evolved human-specific α-satellite that defines the active *D5Z2* α-satellite HOR array, while more ancient α-satellite sequences are located within inactive *D5Z1* α-satellite HOR arrays (Fig. 6a). This is in contrast to the chromosome 12 centromere, which contains α-satellite HORs that are shared among orangutan and chimpanzee (Extended Data Fig. 8b). Finally, the chromosome X centromere is composed of α-satellite HORs and monomers that are evolutionarily similar to each other, and in contrast to the other centromeres, are also similar to macaque's α-satellite monomers (Fig. 6b).

## Mutation rate estimation

As our analyses showed that monomeric α-satellite sequences mutate less quickly and can be readily aligned among human and non-human apes, we focused first on the pericentromeric DNA flanking the α-satellite HOR array. On the basis of the complete sequence from

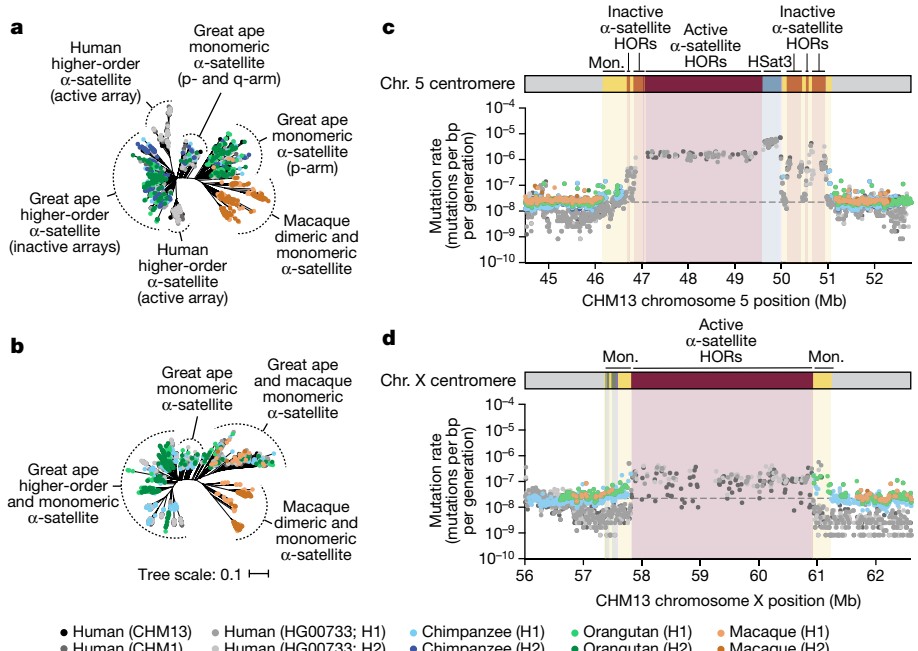

**Fig. 6 | Centromeres evolve with different evolutionary trajectories and mutation rates. a,b**, Phylogenetic trees of human, chimpanzee, orangutan and macaque α-satellites from the higher-order and monomeric (mon.) α-satellite regions of the chromosome 5 (**a**) and X (**b**) centromeres, respectively. **c,d**, The mutation rate of the chromosome 5 (**c**) and X (**d**) centromeric regions, respectively. Individual data points from 10 kb pairwise sequence alignments are shown. Note that the regions corresponding to the active α-satellite HORs have only approximate mutation rates based on human–human comparisons. Owing to unequal rates of mutation and the emergence of new α-satellite HORs, interspecies comparisons are not possible in these regions. HSat3, human satellite 3.

human and NHP centromeric transition regions, we estimated the mutation rate of the approximately 1–2 Mb region flanking the α-satellite HOR arrays using established evolutionary models (Methods) and found that the mutation rate increases from 1.1- to 4.1-fold compared with the unique portions in each of the six centromeres (Fig. 6c,d and Extended Data Fig. 8e–h). The greatest increase in the mutation rate was observed for the chromosome 5 centromere (4.1-fold), while the smallest increase was observed for the chromosome X centromere (1.1-fold), consistent with the observed rapid and slower structural diversity for this chromosome. Owing to nearly complete evolutionary turnover of the α-satellite HORs, biologically meaningful alignment comparisons among humans and NHPs could not be made. However, analyses of the sequence alignments among the four human haplotypes suggest a potential mutation rate increase of at least an order of magnitude, given the caveat that substantial portions of the α-satellite repeats do not align.

To understand the nature of evolutionary change within the α-satellite HOR arrays and especially the emergence of new HORs, we used a population genetics approach leveraging the genetic diversity present in the HPRC[10] and HGSVC[23] genomes. We reasoned that less divergent sequence comparisons within the human species would enable more accurate alignments and, therefore, better reconstruction of the series of mutational events occurring within the α-satellite HOR arrays. Given the relative stability of the flanking monomeric satellite DNA, we constructed phylogenetic trees using the chimpanzee sequence as an outgroup and estimated separation times for different human haplotypes, assuming a chimpanzee and human divergence time of 6 million years (Fig. 7 and Extended Data Figs. 9–11). Under the assumption that there is limited to no recombination across the α-satellite HOR array, we then compared the topologies of both the p- and q-arms, focusing specifically on haplotypes in which we had documented the emergence of novel α-satellite HOR arrays. Despite being anchored in sequence separated 2–3 Mb apart, the p- and q-arm topologies of the resulting trees were similar, consistent with the notion

of suppressed or limited recombination across the region. Importantly, haplotypes containing new α-satellite HORs most often share a monophyletic origin (Fig. 7b and Extended Data Figs. 9–11). For example, in the case of chromosome 12, we estimate the new HORs emerged approximately 13–23 thousand years ago; Fig. 7b), while for chromosome 11, they emerged approximately 80–153 thousand years ago (Extended Data Fig. 11a). This suggests a single origin for the new α-satellite HORs, followed by the saltatory spread of >1 Mb of new HORs to this subset of human haplotypes. As we are specifically selecting haplotypes that show a saltatory amplification of α-satellite HORs, these rate estimates should not be considered genome- or even centromere-wide rates of change.

By directly comparing the structure of the α-satellite HOR arrays with the nearest human haplotype lacking the newly derived HORs, we computed the difference in the number of base pairs, α-satellite monomers, α-satellite HORs and distinct structural changes (Supplementary Table 9). Using these α-satellite HORs as a benchmark, our results suggest 392–2,490 nucleotide differences (or up to two α-satellite HORs) per generation, on average, to create the new HORs on chromosomes 11 and 12 (Fig. 7b and Supplementary Fig. 11a). Given the average length of each α-satellite HOR array and the estimated coalescent time, this translates to considerably different rates for the emergence of these new α-satellite HORs on chromosomes 11 (~30–60 nucleotide differences per Mb per generation) and 12 (~500–1,000 nucleotide differences per Mb per generation; Supplementary Table 9). While caution should be exercised given the focus on new α-satellite HOR structures and the limited number of human haplotypes compared, a notable finding is both the speed at which these new HORs emerged and the interdigitated nature of new α-satellite HORs intermixed with relic ancestral HORs. Our results suggest approximately 100 distinct structural changes (insertions and deletions) as this new HOR variant evolved. This pattern implicates mechanisms other than simple unequal crossover for the spread of novel α-satellite HORs within centromeres. The change in array structure is probably due to saltatory amplification of newly

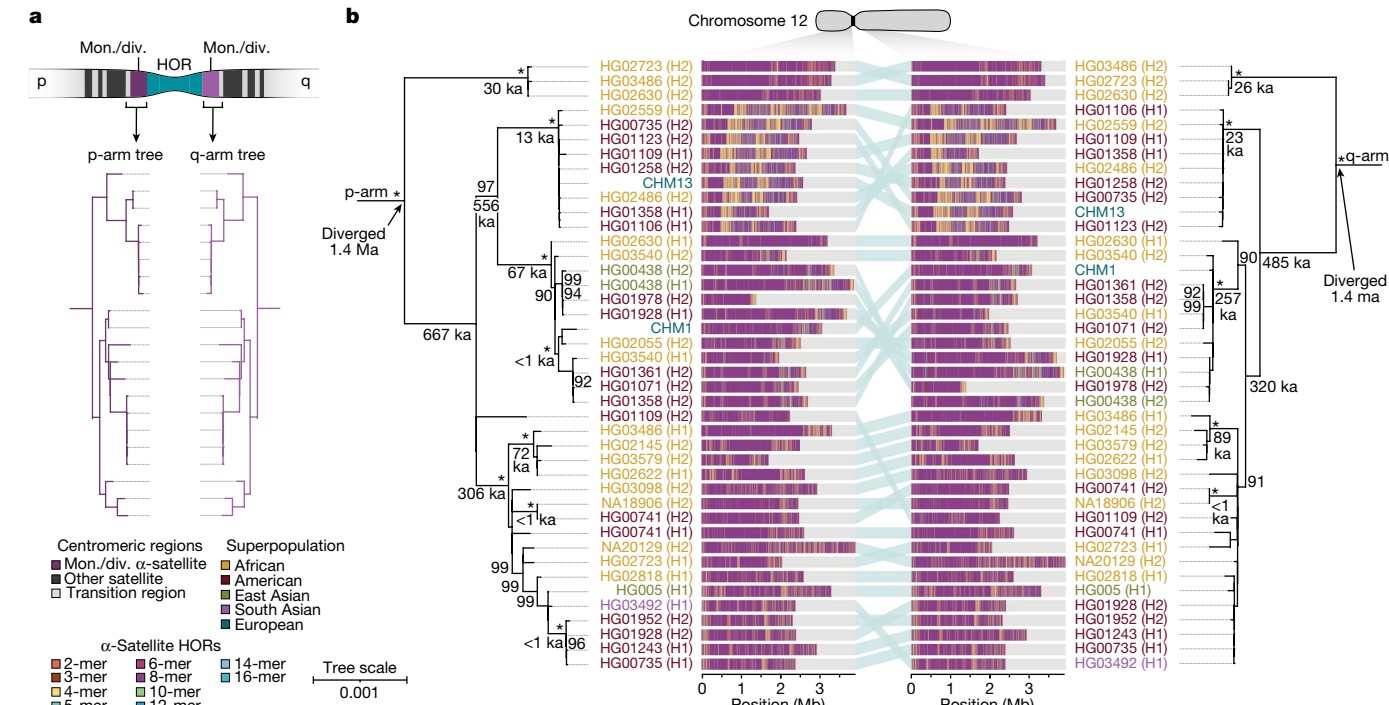

**Fig. 7 | Phylogenetic reconstruction of human centromeric haplotypes and the saltatory amplification of new α-satellite HORs. a**, The strategy to determine the phylogeny and divergence times of completely sequenced centromeres using monomeric α-satellite or unique sequence flanking the canonical α-satellite HOR array from both the p- and q-arms. Chimpanzee was used as an outgroup with an estimated species divergence time of 6 million years ago (Ma). **b**, Maximum-likelihood phylogenetic trees depicting the p- and q-arm topologies along with the estimated divergence times reveal a monophyletic origin for the emergence of new α-satellite HORs within the chromosome 12 (*D12Z3*) α-satellite HOR array. This array shows a complex pattern of new α-satellite HOR insertions and deletions over a short period of evolutionary time. The asterisks indicate nodes with 100% bootstrap support, and nodes with 90–99% bootstrap support are indicated numerically. Nodes without an asterisk or number have bootstrap support <90%. The haplotypes from the p- and the q-arm trees are linked with a light teal bar, as shown in the schematic in **a**. Note that most differences in the order of the haplotypes occur at the terminal branches, where the order of sequence taxa can be readily reshuffled to establish near-complete concordance. Thus, there are no significant changes in the overall topologies of the phylogenetic tree. ka, thousand years ago.

emerged α-satellite HOR variants at multiple sites in the original HOR array, leading to an overall increase in array size from 554 kb to 2 Mb, on average (Fig. 7b and Extended Data Figs. 9–11).

## Discussion

Here we present a detailed comparative analysis of two completely assembled reference sets of human centromeres compared to a diversity panel of human and NHP centromeres. We show a demonstrable acceleration of single-nucleotide and structural variation transitioning from euchromatin to heterochromatin, with most of this excess occurring within the core of the centromeric α-satellite HOR arrays. An important caveat is that a substantial fraction (45–47%) of the completely sequenced centromeres cannot be readily aligned to either of the two references, owing in part to the emergence of new α-satellite HOR structures (Supplementary Tables 6 and 8 and Supplementary Figs. 43 and 44). These initial mutation rate estimates therefore probably represent an underestimate until a greater diversity of human and NHP centromeres is sampled. Notably, we find that the predicted site of the kinetochore attachment varies considerably in location, with eight differing by more than 500 kb in these two human genomes (Fig. 4b). While some of this repositioning corresponds to the emergence of novel α-satellite HORs (Fig. 3c), overall we have not found a one-to-one correspondence between the sites of kinetochore attachment and areas of rapid evolutionary turnover and homogenization as predicted by the kinetochore-associated recombination machinery model[15,34] (Supplementary Figs. 20–42). This notable plasticity in kinetochore position despite the conserved, essential function of these regions underscores the centromere paradox[3], an unresolved conundrum regarding the contradictory phenomenon of rapidly evolving centromeric DNA and proteins despite their essential role in ensuring faithful chromosome transmission.

Comparison of the sequence and structure across six sets of orthologous primate centromeres suggests near-complete lineage-specific turnover of α-satellite HORs as well as unique features specific to each lineage. We find that chimpanzee α-satellite HOR arrays are around 67% smaller than their human counterparts. Orangutan α-satellite HOR arrays are organized as a mosaic patchwork of distinct α-satellite HOR blocks with a high degree of divergence. Macaque centromeres are consistently the largest, but are also more homogenous and composed of dimeric α-satellites that are approximately 95% similar in sequence to each other, with blocks of polymorphic inversions present on some centromeres. Using the emergence of these HOR structures within human as a marker of evolutionary mutability, our coalescent approach suggests that centromeric α-satellite HOR arrays can mutate multiple orders of magnitude more quickly than unique DNA (estimated at 30–1,000 nucleotides per generation per Mb based on our analysis of newly emerged HORs on chromosomes 11 and 12; Supplementary Table 9). These changes in DNA occur most frequently in concert with gains and losses of α-satellite HOR units and do not appear to do so in a contiguous manner but, instead, are intermixed with ancestral HORs. These patterns are consistent with saltatory as opposed to a constant rate of mutation[38], potentially as a result of meiotic drive for the newly minted HORs[39]. Mechanisms involving DNA double-stranded break formation followed by homologous or unidirectional gene conversion between sister chromatids, as has been recently suggested for

*Arabidopsis thaliana*[40], may account for this pattern. The emergence of new α-satellite HORs may also contribute to increased centromere strength[41], which can lead to non-Mendelian chromosome segregation and biased chromosome retention in oocytes[42,43]. Now that centromeres can be fully sequenced, it will be critical to study the mutational processes in multigenerational families to understand the mechanisms shaping these rapidly evolving regions of our genome.

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

## Methods

### Cell lines

CHM1hTERT (CHM1) cells were originally isolated from a hydatidiform mole at Magee-Womens Hospital. Cryogenically frozen cells from this culture were grown and transformed using human telomerase reverse transcriptase (hTERT) to immortalize the cell line. This cell line has been authenticated by short-tandem-repeat analysis by Cell Line Genetics and has tested negative for mycoplasma contamination. Human HG00733 lymphoblastoid cells were originally obtained from a female Puerto Rican child, immortalized with the Epstein–Barr Virus (EBV) and stored at the Coriell Institute for Medical Research. This cell line has been authenticated using a multiplex PCR assay with six autosomal microsatellite markers and has tested negative for mycoplasma contamination. Chimpanzee (*Pan troglodytes*, Clint, S006007) fibroblast cells were originally obtained from a male western chimpanzee named Clint (now deceased) at the Yerkes National Primate Research Center and immortalized with EBV. Orangutan (*Pongo abelii*, Susie, PR01109) fibroblast cells were originally obtained from a female Sumatran orangutan named Susie (now deceased) at the Gladys Porter Zoo, immortalized with EBV and stored at the Coriell Institute for Medical Research. Macaque (*Macaca mulatta*; AG07107) fibroblast cells were originally obtained from a female rhesus macaque of Indian origin and stored at the Coriell Institute for Medical Research. The chimpanzee, orangutan and macaque cell lines have not yet been authenticated or assessed for mycoplasma contamination to our knowledge.

### Cell culture

CHM1 cells were cultured in complete AmnioMax C-100 Basal Medium (Thermo Fisher Scientific, 17001082) supplemented with 15% AmnioMax C-100 Supplement (Thermo Fisher Scientific, 12556015) and 1% penicillin–streptomycin (Thermo Fisher Scientific, 15140122). HG00733 (*Homo sapiens*) cells were cultured in RPMI-1650 medium (Sigma-Aldrich, R8758) supplemented with 15% fetal bovine serum (FBS; Thermo Fisher Scientific, 16000-044) and 1% penicillin–streptomycin (Thermo Fisher Scientific, 15140122). Chimpanzee (*P. troglodytes*; Clint; S006007) and macaque (*Macaque mulatta*; AG07107) cells were cultured in MEM α containing ribonucleosides, deoxyribonucleosides and L-glutamine (Thermo Fisher Scientific, 12571063) supplemented with 12% FBS (Thermo Fisher Scientific, 16000-044) and 1% penicillin–streptomycin (Thermo Fisher Scientific, 15140122). Orangutan (*P. abelii*; Susie; PR01109) cells were cultured in MEM α containing ribonucleosides, deoxyribonucleosides and L-glutamine (Thermo Fisher Scientific, 12571063) supplemented with 15% FBS (Thermo Fisher Scientific, 16000-044) and 1% penicillin–streptomycin (Thermo Fisher Scientific, 15140122). All cells were cultured in a humidity-controlled environment at 37 °C under 95% $O_2$.

### DNA extraction, library preparation and sequencing

PacBio HiFi data were generated from the CHM1 and HG00733 genomes as previously described[21] with some modifications. In brief, high-molecular-weight DNA was extracted from cells using a modified Qiagen Gentra Puregene Cell Kit protocol[47]. High-molecular-weight DNA was used to generate PacBio HiFi libraries using the Template Prep Kit v1 (PacBio, 100-259-100) or SMRTbell Express Template Prep Kit v2 (PacBio, 100-938-900) and SMRTbell Enzyme Clean Up kits (PacBio, 101-746-400 and 101-932-600). Size selection was performed with SageELF (Sage Science, ELF001), and fractions sized 11 kb, 14 kb, 15 kb or 16 kb (as determined by FEMTO Pulse (Agilent, M5330AA)) were chosen for sequencing. Libraries were sequenced on the Sequel II platform with seven or eight SMRT Cells 8M (PacBio, 101-389-001) per sample using either Sequel II Sequencing Chemistry 1.0 (PacBio, 101-717-200) or 2.0 (PacBio, 101-820-200), both with 2 h pre-extension and 30 h videos, aiming for a minimum estimated coverage of 30× in PacBio HiFi reads (assuming a genome size of 3.1 Gb). Raw CHM1 data were processed using DeepConsensus[48] (v.0.2.0) with the default parameters. Raw HG00733 data were processed using the CCS algorithm (v.3.4.1) with the following parameters: --minPasses 3 --minPredictedAccuracy 0.99 --maxLength 21000 or 50000.

Ultra-long ONT data were generated from the CHM1, HG00733, chimpanzee, orangutan and macaque genomes according to a previously published protocol[49]. In brief, $3–5 \times 10^7$ cells were lysed in a buffer containing 10 mM Tris-Cl (pH 8.0), 0.1 M EDTA (pH 8.0), 0.5% (w/v) SDS and 20 µg ml$^{-1}$ RNase A (Qiagen, 19101) for 1 h at 37 °C. Then, 200 µg ml$^{-1}$ proteinase K (Qiagen, 19131) was added, and the solution was incubated at 50 °C for 2 h. DNA was purified through two rounds of 25:24:1 (v/v) phenol–chloroform–isoamyl alcohol extraction followed by ethanol precipitation. Precipitated DNA was solubilized in 10 mM Tris (pH 8.0) containing 0.02% Triton X-100 at 4 °C for 2 days. Libraries were constructed using the Ultra-Long DNA Sequencing Kit (ONT, SQK-ULK001) with modifications to the manufacturer's protocol. Specifically, around 40 µg of DNA was mixed with FRA enzyme and FDB buffer as described in the protocol and incubated for 5 min at room temperature, followed by a 5 min heat-inactivation at 75 °C. RAP enzyme was mixed with the DNA solution and incubated at room temperature for 1 h before the clean-up step. Clean-up was performed using the Nanobind UL Library Prep Kit (Circulomics, NB-900-601-01) and eluted in 225 µl EB. Then, 75 µl of library was loaded onto a primed FLO-PRO002 R9.4.1 flow cell for sequencing on the PromethION, with two nuclease washes and reloads after 24 and 48 h of sequencing.

Additional ONT data were generated from the CHM1, HG00733, chimpanzee, orangutan and macaque genomes according to a previously published protocol[21]. In brief, high-molecular-weight DNA was extracted from cells using a modified Qiagen Gentra Puregene protocol[47]. High-molecular-weight DNA was prepared into libraries with the Ligation Sequencing Kit (SQK-LSK110) from ONT and loaded onto primed FLO-PRO002 R9.4.1 flow cells for sequencing on the PromethION system, with two nuclease washes and reloads after 24 and 48 h of sequencing. All ONT data were base-called using Guppy (v.5.0.11) with the SUP model.

### Targeted sequence assembly and validation of centromeric regions

To generate complete assemblies of centromeric regions from the CHM1, HG00733, chimpanzee, orangutan and macaque genomes, we first assembled each genome from PacBio HiFi data (Supplementary Table 1) using hifiasm[24] (v.0.16.1). The resulting PacBio HiFi contigs were aligned to the T2T-CHM13 reference genome[4] (v.2.0) using minimap2[50] (v.2.24) with the following parameters: -I15G -a --eqx -x asm20 -s 5000. Fragmented centromeric contigs were subsequently scaffolded with ultra-long (>100 kb) ONT data generated from the same source genome using a method that takes advantage of SUNKs (Supplementary Fig. 1; https://github.com/arozanski97/SUNK-based-contig-scaffolding). In brief, SUNKs ($k$ = 20 bp) were identified from the CHM1 PacBio HiFi whole-genome assembly using Jellyfish (v.2.2.4) and barcoded on the CHM1 PacBio HiFi centromeric contigs as well as all ultra-long ONT reads. PacBio HiFi centromeric contigs sharing a SUNK barcode with ultra-long ONT reads were subsequently joined together to generate contiguous assemblies that traverse each centromeric region. The base accuracy of the assemblies was improved by replacing the ONT sequences with locally assembled PacBio HiFi contigs generated using HiCanu[7] (v.2.1.1).

We validated the construction of each centromere assembly using four different methods. First, we aligned native PacBio HiFi and ONT data from the same source genome to each whole-genome assembly using pbmm2 (v.1.1.0) (for PacBio HiFi data; https://github.com/PacificBiosciences/pbmm2) or Winnowmap[51] (v.1.0) (for ONT data) and assessed the assemblies for uniform read depth across the centromeric regions using IGV[52] and NucFreq[22]. We next assessed the concordance between the assemblies and raw PacBio HiFi data using VerityMap[27],

which identifies discordant $k$-mers between the two and flags them for correction. We then assessed the concordance between the assemblies and ONT data using GAVISUNK[28], which identifies concordant SUNKs between the two. Finally, we estimated the accuracy of the centromere assemblies from mapped $k$-mers ($k = 21$) using Merqury (v.1.1)[53] and publicly available Illumina data from each genome (Extended Data Table 1). We estimated the QV of the centromeric regions with the following formula:

$$-10 \times \log(1 - (1 - (\text{number of erroneous } k\text{-mers/total number of } k\text{-mers}))^{(1/k)})$$

## FISH and spectral karyotyping

To determine the karyotype of the CHM1 genome, we first prepared metaphase chromosome spreads by arresting CHM1 cells in mitosis via the addition of KaryoMAX Colcemid Solution (0.1 µg ml$^{-1}$, Thermo Fisher Scientific, 15212012) to the growth medium for 6 h. Cells were collected by centrifugation at 200$g$ for 5 min and incubated in 0.4% KCl swelling solution for 10 min. Swollen cells were pre-fixed by the addition of freshly prepared methanol:acetic acid (3:1) fixative solution (~100 µl per 10 ml total volume). Pre-fixed cells were collected by centrifugation at 200$g$ for 5 min and fixed in methanol:acetic acid (3:1) fixative solution. Spreads were dropped on a glass slide and incubated on a heating block at 65 °C overnight. Before hybridization, slides were treated with 1 mg ml$^{-1}$ RNase A (Qiagen, 19101) in 2× SSC for at least 45 min at 37 °C and then dehydrated in a 70%, 80% and 100% ethanol series for 2 min. Denaturation of spreads was performed in 70% formamide/2× SSC solution at 72 °C for 1.5 min and was immediately stopped by immersing the slides into an ethanol series pre-chilled to −20 °C.

Fluorescent probes for spectral karyotyping were generated in-house. Individual fluorescently labelled whole-chromosome paints were obtained from Applied Spectral Imaging. Paints were provided in a hybridization buffer and mixed 1:1 for indicated combinations. Labelled chromosome probes and paints were denatured by heating to 80 °C for 10 min before applying them to denatured slides. Spreads were hybridized to probes under a HybriSlip hybridization cover (Grace Bio-Labs, 716024) sealed with Cytobond (SciGene, 2020-00-1) in a humidified chamber at 37 °C for 48 h. After hybridization, the slides were washed three times in 50% formamide/2× SSC for 5 min at 45 °C, 1× SSC solution at 45 °C for 5 min twice, and at room temperature once. The slides were then rinsed with double-deionized H$_2$O, air-dried and mounted in Vectashield-containing DAPI (Vector Laboratories, H-1200-10).

For spectral karyotyping, images were acquired using LSM710 confocal microscope (Zeiss) with the 63×/1.40 NA oil-immersion objective and ZEN (v.3.7) software. Segmentation, spectral unmixing and identification of chromosomes were performed using an open-source karyotype identification via spectral separation (KISS) analysis package for Fiji[54] (v.2.13.1), freely available online (http://research.stowers.org/imagejplugins/KISS_analysis.html). A detailed description of chromosome paints, hybridization and analysis procedures was reported previously[55].

For individually painted chromosomes, $z$ stack images were acquired on the Nikon Ti-E microscope equipped with a 100× objective NA 1.45, Yokogawa CSU-W1 spinning disk and Flash 4.0 sCMOS camera with NIS-Elements AR (v.3.2) software. Image processing was performed in Fiji[54] (v.2.13.1).

## Strand-seq analysis

To assess the karyotype of the CHM1 genome, we prepared strand-seq libraries from CHM1 cells using a previously published protocol[56,57]. We sequenced the mono- and dinucleosome fractions separately, with the mononucleosomes sequenced with 75 bp, paired-end Illumina sequencing, and the dinucleosomes sequenced with 150 bp, paired-end Illumina sequencing. We demultiplexed the raw sequencing data based on library-specific barcodes and converted them to FASTQ files using

Illumina standard software. We aligned the reads in the FASTQ files to the T2T-CHM13 reference genome[4] (v.2.0) using BWA[58] (v.0.7.17-r1188), sorted the alignments using SAMtools[59] (v.1.9) and marked duplicate reads using sambamba[60] (v.1.0). We merged the BAM files for the mono- and dinucleosome fractions of each cell using SAMtools[59] (v.1.9). We used breakpointR (v.1.18)[61] to assess the quality of generated strand-seq libraries with the following parameters: windowsize = 2000000, binMethod = 'size', pairedEndReads = TRUE, min.mapq = 10, background = 0.1, minReads = 50. We filtered the libraries based on the read density, level of background reads and level of genome coverage variability[62]. In total, 48 BAM files were selected for all subsequent analysis and are publicly available. We detected changes in strand-state inheritance across all strand-seq libraries using the R package AneuFinder[63] with the following parameters: variable.width.reference = <merged BAM of all 48 strand-seq libraries>, binsizes = windowsize, use.bamsignals = FALSE, pairedEndReads = TRUE, remove.duplicate.reads = TRUE, min.mapq = 10, method = 'edivisive', strandseq = TRUE, cluster.plots = TRUE, refine.breakpoints = TRUE. We extracted a list of recurrent strand-state changes reported as sister chromatid exchange hotspots by AneuFinder. With this analysis, we identified reciprocal translocations between chromosomes 4q35.1/11q24.3 and 16q23.3/17q25.3 (see below) and established the overall copy number for each chromosome and strand-seq library.

To identify the reciprocal translocation breakpoints between chromosomes 4q35.1/11q24.3 and 16q23.3/17q25.3 in the CHM1 genome, we first aligned CHM1 PacBio HiFi reads to the T2T-CHM13 reference genome[4] (v.2.0) using pbmm2 (v.1.1.0) and used BEDtools[64] intersect (v.2.29.0) to define putative translocation regions based on AneuFinder analysis (described above). We extracted PacBio HiFi reads with supplementary alignments using SAMtools[59] (v.1.9) flag 2048. Using this method, we were able to identify the precise breakpoint of each translocation. Note that, for the reciprocal translocation between chromosomes 4q35.1/11q24.3, we report two breakpoints in each chromosome due to the presence of a ~97–98 kb deletion in the translocated homologues (Supplementary Fig. 3). The breakpoints are located at chromosome 4:187112496/chromosome 11:130542388, chromosome 4:187209555/chromosome 11:130444240, and chromosome 16:88757545/chromosome 17:81572367 (in T2T-CHM13 v.2.0).

## Sequence identity across centromeric regions

To calculate the sequence identity across the centromeric regions from CHM1, CHM13 and 56 other diverse human genomes (generated by the HPRC[10] and HGSVC[23]), we performed three analyses that take advantage of different alignment methods. In the first analysis, we performed a pairwise sequence alignment between contigs from the CHM1, CHM13 and diverse genomes using minimap2[50] (v.2.24) and the following command: minimap2 -I15G -K 8G -t {threads} -ax asm20 --secondary=no --eqx -s 2500 {ref.fasta} {query.fasta}. We chose these minimap2 parameters after testing several options and identifying optimal ones for alignment between repetitive and/or structurally divergent regions in diploid human genomes. Specifically, we chose -I15G to provide additional memory for aligning between centromeric regions (the default is 4G and sometimes throws an error because of the large number of potential alignments). We also chose -K 8G because it allows for 8 Gb of sequence to be loaded into memory at a time. This is enough for a typical human diploid genome (~6 Gb) to be loaded. If we had left it at the default (500M), only a subset of contigs would be loaded at a time, and once the shortest contigs align, we would be left with only one thread aligning the longest contig. We therefore chose to increase this parameter so that the whole assembly is aligned at one time. We also chose to use -ax asm20 as it allows for sequences that are up to 20% divergent to be aligned. This is more permissive to alternative α-satellite HOR structures and sequence compositions than the other alignment options (for example, asm5 and asm10). We also opted to use --secondary=no to prevent secondary alignments from the

same contig, thereby preventing multi-mapping and ensuring that the query would only align once to the reference. We added --eqx to allow us to parse the CIGAR string and calculate the mean sequence identity of the alignments. Finally, we selected -s 2500 as the minimal peak dynamic programming alignment score. The default setting for this parameter is 40, and we tested that one as well as 1000, 2500 and 5000. We found that with -s 40 and -s 1000, spurious alignments occurred from other centromeres, and with -s 5000, accurate alignments from centromeres were filtered out. We therefore chose -s 2500 to allow for diverse α-satellite HOR structures to align without some alignments being filtered out. After generating the alignments, we filtered them using SAMtools[59] (v.1.9) flag 4, which keeps primary and partial alignments. We subsequently partitioned the alignments into 10 kb non-overlapping windows in the reference genome (either CHM1 or CHM13) and calculated the mean sequence identity between the pairwise alignments in each window with the following formula: (number of matches)/(number of matches + number of mismatches + number of insertion events + number of deletion events). We then averaged the sequence identity across the 10 kb windows within the α-satellite HOR array(s), monomeric/diverged α-satellites, other satellites and non-satellites for each chromosome to determine the mean sequence identity in each region.

In the second analysis, we first fragmented the centromeric contigs from each genome assembly into 10 kb fragments with seqtk (v.1.3; https://github.com/lh3/seqtk) and subsequently aligned them to the reference genome (either CHM1 or CHM13) using minimap2[50] (v.2.24) and the following command: minimap2 -I15G -K 8G -t {threads} -ax asm20 --secondary=no --eqx -s 40 {ref.fasta} {query.fasta}. We filtered the alignments using SAMtools[59] (v.1.9) flag 4, which keeps primary and partial alignments. In this method, multiple 10 kb fragments are allowed to align to the same region in the reference genome, but each 10 kb fragment is only allowed to align once. We then partitioned the alignments into 10 kb non-overlapping windows in the reference genome and calculated the mean sequence identity between all alignments in each window as described above. We averaged the sequence identity across the 10 kb windows within the α-satellite HOR array(s), monomeric/diverged α-satellites, other satellites and non-satellites for each chromosome to determine the mean sequence identity in each region.

In the third analysis, we first identified the location of the α-satellite HOR array(s) in each genome assembly using RepeatMasker[65] (v.4.1.0) followed by HumAS-HMMER (https://github.com/fedorrik/HumAS-HMMER_for_AnVIL) and subsequently extracted regions enriched with 'live' α-satellite HORs (denoted with an 'L' in the HumAS-HMMER BED file). We then ran TandemAligner[66] (v.0.1) on pairs of complete centromeric HOR arrays using the following command: tandem_aligner --first {ref.fasta} --second {query.fasta} -o {output_directory}. We parsed the CIGAR string generated by TandemAligner by first binning the alignments into 10 kb non-overlapping windows and calculating the mean sequence identity in each window as described above. As TandemAligner is only optimized for tandem repeat arrays, we assessed the sequence identity only in the α-satellite HOR array(s) of each centromeric region and did not use it to assess the sequence identity in any other region.

### Better-match analysis
To determine whether the CHM1 or CHM13 centromeres are a better match to those from the 56 diverse human genomes assembled by the HPRC[10] and HGSVC[23], we performed a pairwise sequence alignment between contigs from the HPRC and HGSVC assemblies to either the CHM1 or CHM1 assembly using minimap2[50] (v.2.24) and the following command: minimap2 -I15G -K 8G -t {threads} -ax asm20 --secondary=no --eqx -s 2500 {ref.fasta} {query.fasta}. We filtered the alignments using SAMtools[59] (v.1.9) flag 4, which keeps primary, secondary and partial alignments, and then calculated an alignment score between each pair

of haplotypes, limiting our analysis to only the centromeric α-satellite HOR arrays as follows: (total number of aligned bases in the query)/ (total number of bases in the reference) × (mean sequence identity by event). The mean sequence identity by event is calculated as follows: (number of matches)/(number of matches + number of mismatches + number of insertion events + number of deletion events). The set of centromeres with a higher alignment score was determined to be a better match to that haplotype than the other set of centromeres.

### Pairwise sequence identity heat maps
To generate pairwise sequence identity heat maps of each centromeric region, we ran StainedGlass[44] (v.6.7.0) with the following parameters: window=5000 mm_f=30000 mm_s=1000. We normalized the colour scale across the StainedGlass plots by binning the percentage of sequence identities equally and recolouring the data points according to the binning. To generate heat maps that show only the variation between centromeric regions, we ran StainedGlass[44] (v.6.7.0) with the following parameters: window=5000 mm_f=60000 mm_s=30000. As above, we normalized the colour scale across the StainedGlass plots by binning the percentage of sequence identities equally and recolouring the datapoints according to the binning.

### Estimation of α-satellite HOR array length
To estimate the length of the α-satellite HOR arrays of each centromere in the CHM1, CHM13 and 56 diverse genome assemblies[10,23], we first ran RepeatMasker[65] (v.4.1.0) on the assemblies and identified contigs containing α-satellite repeats, marked by 'ALR/Alpha'. We extracted these α-satellite-containing contigs and ran HumAS-HMMER (https://github.com/fedorrik/HumAS-HMMER_for_AnVIL) on each of them. HumAS-HMMER is a tool that identifies the location of α-satellite HORs in human centromeric sequences. It uses a hidden Markov model (HMM) profile for centromeric α-satellite HOR monomers and generates a BED file with the coordinates of the α-satellite HORs and their classification. Using this BED file, we extracted contigs containing α-satellite HORs that were designated as live or active (denoted with an 'L' in the HumAS-HMMER BED file), which are those that belong to an array that consistently associates with the kinetochore in several individuals[5,67]. By contrast, dead or inactive α-satellite HORs (denoted with a 'd' in the HumAS-HMMER BED file) are those that have not been found to be associated with the kinetochore and are usually more divergent in sequence than the live or active arrays. We filtered out contigs that had incomplete α-satellite HOR arrays (such as those that did not traverse into unique sequence), thereby limiting our analysis to only complete α-satellite HOR arrays. Moreover, we assessed the integrity of each of the α-satellite HOR array-containing contigs using NucFreq[22] to ensure that they were completely and accurately assembled, filtering out those with evidence of a deletion, duplication or misjoin. Finally, we calculated the length of the α-satellite HOR arrays in the remaining contigs by taking the minimum and maximum coordinate of the 'live' α-satellite HOR arrays and plotting their lengths with GraphPad Prism (v.9.5.1).

### Sequence composition and organization of α-satellite HOR arrays
To determine the sequence composition and organization of each α-satellite HOR array in the CHM1, CHM13 and 56 diverse genome assemblies[10,23], we ran HumAS-HMMER (https://github.com/fedorrik/HumAS-HMMER_for_AnVIL) on centromeric contigs with the default parameters and parsed the resulting BED file with StV (https://github.com/fedorrik/stv). This generated a BED file with each α-satellite HOR sequence composition and its organization along the α-satellite HOR arrays. We used the stv_row.bed file to visualize the organization of the α-satellite HOR arrays with R[68] (v.1.1.383) and the ggplot2 package[66]. The α-satellite monomer and HOR classification generated with HumAS-HMMER is described in detail in the supplementary

information of a previous study[5], in which a more complete description of these annotations can be found.

## CpG methylation analysis

To determine the CpG methylation status of each CHM1 centromere, we aligned CHM1 ONT reads >30 kb in length to the CHM1 whole-genome assembly using Winnowmap[51] (v.1.0) and then assessed the CpG methylation status of the centromeric regions with Nanopolish[69] (v.0.13.3). Nanopolish distinguishes 5-methylcytosines from unmethylated cytosines via a HMM on the raw nanopore current signal. The methylation caller generates a log-likelihood value for the ratio of probability of methylated to unmethylated CpGs at a specific $k$-mer. We filtered methylation calls using the nanopore_methylation_utilities tool[70] (https://github.com/isaclee/nanopore-methylation-utilities), which uses a log-likelihood ratio of 2.5 as a threshold for calling methylation. CpG sites with log-likelihood ratios greater than 2.5 (methylated) or less than −2.5 (unmethylated) are considered to be high quality and are included in the analysis. Reads that do not have any high-quality CpG sites are filtered from the BAM for subsequent methylation analysis. Nanopore_methylation_utilities integrates methylation information into the BAM file for viewing in IGV's[52] bisulfite mode, which was used to visualize CpG methylation. To determine the size of hypomethylated region (termed the CDR[31]) in each centromere, we developed a novel tool, CDR-Finder (https://github.com/arozanski97/CDR-Finder). This tool first bins the assembly into 5 kb windows, computes the median CpG methylation frequency within windows containing α-satellite (as determined by RepeatMasker[65] (v.4.1.0), selects bins that have a lower CpG methylation frequency than the median frequency in the region, merges consecutive bins into a larger bin, filters for merged bins that are >50 kb and reports the location of these bins.

## Native CENP-A ChIP–seq and analysis

To determine the location of centromeric chromatin within the CHM1 genome, we performed two independent replicates of native CENP-A chromatin immunprecipitation–sequencing (ChIP–seq) analysis of CHM1 cells as described previously[21], with some modifications. In brief, $3–4 \times 10^{7}$ cells were collected and resuspended in 2 ml of ice-cold buffer I (0.32 M sucrose, 15 mM Tris, pH 7.5, 15 mM NaCl, 5 mM MgCl$_2$, 0.1 mM EGTA and 2× Halt Protease Inhibitor Cocktail (Thermo Fisher Scientific, 78429)). Then, 2 ml of ice-cold buffer II (0.32 M sucrose, 15 mM Tris, pH 7.5, 15 mM NaCl, 5 mM MgCl$_2$, 0.1 mM EGTA, 0.1% IGEPAL and 2× Halt Protease Inhibitor Cocktail) was added, and the samples were placed onto ice for 10 min. The resulting 4 ml of nuclei was gently layered on top of 8 ml of ice-cold buffer III (1.2 M sucrose, 60 mM KCl, 15 mM, Tris pH 7.5, 15 mM NaCl, 5 mM MgCl$_2$, 0.1 mM EGTA and 2× Halt Protease Inhibitor Cocktail (Thermo Fisher Scientific, 78429)) and centrifuged at 10,000$g$ for 20 min at 4 °C. Pelleted nuclei were resuspended in buffer A (0.34 M sucrose, 15 mM HEPES, pH 7.4, 15 mM NaCl, 60 mM KCl, 4 mM MgCl$_2$ and 2× Halt Protease Inhibitor Cocktail) to 400 ng ml$^{-1}$. Nuclei were frozen on dry ice and stored at 80 °C. MNase digestion reactions were performed on 200–300 μg chromatin, using 0.2–0.3 U μg$^{-1}$ MNase (Thermo Fisher Scientific, 88216) in buffer A supplemented with 3 mM CaCl$_2$ for 10 min at 37 °C. The reaction was quenched with 10 mM EGTA on ice and centrifuged at 500$g$ for 7 min at 4 °C. The chromatin was resuspended in 10 mM EDTA and rotated at 4 °C for 2 h. The mixture was adjusted to 500 mM NaCl, rotated for another 45 min at 4 °C and then centrifuged at maximum speed (21,100$g$) for 5 min at 4 °C, yielding digested chromatin in the supernatant. Chromatin was diluted to 100 ng ml$^{-1}$ with buffer B (20 mM Tris, pH 8.0, 5 mM EDTA, 500 mM NaCl and 0.2% Tween-20) and precleared with 100 μl 50% protein G Sepharose bead (Abcam, ab193259) slurry for 20 min at 4 °C with rotation. Precleared supernatant (10–20 μg bulk nucleosomes) was saved for further processing. To the remaining supernatant, 20 μg mouse monoclonal anti-human CENP-A antibody (3-19; Enzo, ADI-KAM-CC006-E; approximately

a 1:80 dilution) was added and rotated overnight at 4 °C. Immunocomplexes were recovered by the addition of 200 ml 50% protein G Sepharose bead slurry followed by rotation at 4 °C for 3 h. The beads were washed three times with buffer B and once with buffer B without Tween-20. For the input fraction, an equal volume of input recovery buffer (0.6 M NaCl, 20 mM EDTA, 20 mM Tris, pH 7.5 and 1% SDS) and 1 ml of RNase A (10 mg ml$^{-1}$) was added, followed by incubation for 1 h at 37 °C. Proteinase K (100 mg ml$^{-1}$, Roche) was then added, and the samples were incubated for another 3 h at 37 °C. For the ChIP fraction, 300 μl of ChIP recovery buffer (20 mM Tris, pH 7.5, 20 mM EDTA, 0.5% SDS and 500 mg ml$^{-1}$ proteinase K) was added directly to the beads and incubated for 3–4 h at 56 °C. The resulting proteinase-K-treated samples were subjected to a phenol−chloroform extraction followed by purification using the Qiagen MinElute PCR purification column. Unamplified bulk nucleosomal and ChIP DNA was analysed using an Agilent Bioanalyzer instrument and a 2100 High Sensitivity Kit.

Sequencing libraries were generated using the TruSeq ChIP Library Preparation Kit, Set A (Illumina, IP-202-1012) according to the manufacturer's instructions, with some modifications. In brief, 5–10 ng bulk nucleosomal or ChIP DNA was end-repaired and A-tailed. Illumina TruSeq adaptors were ligated, libraries were size-selected to exclude polynucleosomes using an E-Gel SizeSelect II agarose gel and the libraries were PCR-amplified using the PCR polymerase and primer cocktail provided in the kit. The resulting libraries were submitted for 150 bp, paired-end Illumina sequencing using the NextSeq 500/550 High Output Kit v2.5 (300 cycles). The resulting reads were assessed for quality using FastQC (https://github.com/s-andrews/FastQC), trimmed with Sickle (v.1.33; https://github.com/najoshi/sickle) to remove low-quality 5′- and 3′-end bases, and trimmed using Cutadapt[71] (v.1.18) to remove adapters.

Processed CENP-A ChIP and bulk nucleosomal reads were aligned to the CHM1 whole-genome assembly using BWA-MEM[72] (v.0.7.17) with the following parameters: bwa mem -k 50 -c 1000000 {index} {read1. fastq.gz} {read2.fastq.gz}. The resulting SAM files were filtered using SAMtools[59] (v.1.9) with flag score 2308 to prevent multi-mapping of reads. With this filter, reads mapping to more than one location are randomly assigned a single mapping location, thereby preventing mapping biases in highly identical regions. Alignments were normalized and filtered with deepTools[73] (v.3.4.3) bamCompare with the following parameters: bamCompare -b1 {ChIP.bam} -b2 {bulk_nucleosomal. bam} --operation ratio --binSize 1000 --minMappingQuality 1 -o {out. bw}. Alternatively, CENP-A ChIP–seq data alignments were filtered using a marker-assisted mapping strategy as described previously[5]. In brief, unique 51-mers in the CHM1 whole-genome assembly were counted and filtered with meryl[53] (v.1.3). The locations of the unique 51-mers were identified with meryl[53] (v.1.3) and then used to filter the CENP-A ChIP–seq and input alignments using BEDtools[64] intersect (v.2.29.0). Alignments were normalized and filtered with deepTools[73] (v.3.4.3) bamCompare with the following parameters: bamCompare -b1 {ChIP.bam} -b2 {bulk_nucleosomal.bam} --operation ratio --binSize 1000 -o {out.bw}.

## Estimation of the length of the kinetochore sites

To estimate the length of the CHM1 and CHM13 kinetochore sites, we first determined the CpG methylation status of each CHM1 and CHM13 centromere using the approach described above (see the 'CpG methylation analysis' section). We then mapped the CENP-A ChIP–seq data from each genome to the same source genome using the mapping parameters described above (see the 'Native CENP-A ChIP–seq and analysis' section). We next used CDR-Finder (https://github.com/ arozanski97/CDR-Finder) to identify the location of hypomethylated regions within the centromeres, and we filtered the hypomethylated regions that had less than tenfold enrichment of CENP-A ChIP–seq reads relative to the bulk nucleosomal reads. We reported the lengths of the hypomethylated regions enriched with CENP-A as determined with

CDR-Finder, and we tested for statistical significance using a two-sided Kolmogorov–Smirnov test with GraphPad Prism (v.9.5.1).

## Immuno-FISH on stretched metaphase chromosome spreads

Mechanically stretched metaphase spreads were obtained from the CHM1 cell line according to established procedures[74]. In brief, colcemid-treated cells were washed in phosphate-buffered saline (1× PBS), counted, and resuspended for 15 min in a hypotonic buffer HCM (10 mM HEPES, pH 7.3, 1 mM glycerol, 1 mM CaCl₂ and 0.8 mM MgCl₂) to achieve a final concentration of 10,000 cells per ml. Then, 0.5 ml of the cell suspension was cytocentrifuged onto glass slides at 2,000 rpm for 8 min with a Shandon Cytospin 3 and fixed in methanol at −20 °C for 15 min and in methanol:acetic acid 3:1 at −20 °C for 30 min. The slides were aged overnight at room temperature.

Immunofluorescence was performed on the stretched metaphase chromosome spreads using an in-house rabbit polyclonal CENP-C antibody as previously described with minor modifications[75]. In brief, each slide was rehydrated by immersion in 1× PBS-azide (10 mM NaPO₄, pH 7.4, 0.15 M NaCl, 1 mM EGTA and 0.01% NaN₃) for 15 min at room temperature. Chromosomes were then swollen by washing the slides (three times, 2 min each) with 1× TEEN (1 mM triethanolamine-HCl, pH 8.5, 0.2 mM NaEDTA, and 25 mM NaCl), 0.5% Triton X-100 and 0.1% BSA. The primary polyclonal antibody against the centromeric protein CENP-C was diluted 1:40 in the same solution and then added (100 µl) onto the slides. Each slide was incubated for 2 h at 37 °C. Excess of primary antibody was removed by washing the slides at room temperature (three times, 2, 5 and 3 min each) with 1× KB buffer (10 mM Tris-HCl, pH 7.7, 0.15 M NaCl and 0.1% BSA). A goat anti-rabbit IgG secondary antibody conjugated to FITC (Sigma-Aldrich, F0382) was diluted 1:40 in the same solution, and 100 µl was then added to the slides that were then incubated for 45 min at 37 °C in a dark chamber. After incubation with the secondary antibody, the slides were washed once with 1× KB for 2 min, prefixed with 4% paraformaldehyde in 1× KB for 45 min at room temperature, washed with distilled H₂O by immersion for 10 min at room temperature, and fixed with methanol and acetic acid (3:1) for 15 min. FISH was then performed using two α-satellite-containing plasmids (pZ21A and pGA16) directly labelled by nick-translation with Cy3-dUTP (Enzo, 42501) according to a standard procedure with minor modifications[76]. In brief, 300 ng of labelled probe was used for the FISH experiments; DNA denaturation was performed at 70 °C for 4 min and hybridization at 37 °C in 2× SSC, 50% (v/v) formamide, 10% (w/v) dextran sulphate, 3 µg Cot-1 DNA and 3 mg sonicated salmon sperm DNA, in a volume of 10 µl. Post-hybridization washing was performed under high stringency conditions: at 60 °C in 0.1× SSC (three times, 5 min each). Nuclei and chromosome metaphases were simultaneously DAPI-stained. Digital images were obtained using a Leica DMRXA2 epifluorescence microscope equipped with a cooled CCD camera (Princeton Instruments). DAPI, Cy3 and fluorescein fluorescence signals, detected with specific filters, were recorded separately as grayscale images. Pseudocolouring and merging of images were performed using ImageJ (v.1.53k).

## Human and NHP α-satellite SF classification and strand orientation analysis

Human and NHP α-satellite monomers are grouped into 20 distinct SF classes based on shared sequence identity and structure, which is described in detail previously[5]. The SF classes and their monomers are as follows: SF1 (J1 and J2), SF01 (J3, J4, J5 and J6), SF2 (D2, D2, FD), SF02 (D3, D4, D5, D6, D7, D8 and D9), SF3 (W1, W2, W3, W4 and W5), SF4 (Ga), SF5 (R1 and R2), SF6 (Ha), SF7 (Ka), SF8 (Oa and Na), SF9 (Ca), SF10 (Ba), SF11 (Ja), SF12 (Aa), SF13 (Ia), SF14 (La), SF15 (Fa), SF16 (Ea), SF17 (Qa), SF18 (Pa and Ta). To determine the α-satellite SF content and strand orientation of human and NHP centromeres, we ran HumAS-HMMER (https://github.com/fedorrik/HumAS-HMMER_for_AnVIL) on centromeric contigs with the following command: hmmer-run_SF.sh {path_to_directory_with_fasta} AS-SFs-hmmer3.0.290621.hmm {number_of_threads}. This generated a BED file with the SF classification and strand orientation of each α-satellite monomer, which we visualized with R[68] (v.1.1.383) using the ggplot2 package[66]. In cases in which an inversion was detected, we ran StringDecomposer[77], a tool that detects and reports changes in orientation of tandem repeats, using the default parameters to confirm the presence of reoriented α-satellite monomers at the breakpoints. Finally, we validated the presence of the inversion by aligning native ultra-long ONT reads to the assemblies as described above and confirming even coverage across the breakpoints as well as the presence of inverted α-satellite monomers in the aligned reads.

We uploaded the α-satellite SF and strand orientation tracks generated by HumAS-HMMER for each centromere assembly to the UCSC Human Genome Browser. For the CHM1 centromeres, we uploaded two additional tracks: one showing each α-satellite monomer belonging to known human HORs (ASat-HOR track) and another showing structural variation in human HORs (StV track). All tracks were built and colour-coded as described previously[5] and are publicly available online (https://genome.ucsc.edu/s/fedorrik/chm1_cen (CHM1); https://genome.ucsc.edu/s/fedorrik/T2T_dev (CHM13); https://genome.ucsc.edu/s/fedorrik/cen_primates (chimpanzee, orangutan, and macaque)). Note that the SF annotation coverage in macaque is sometimes discontinuous (some monomers are not annotated due to significant divergence of macaque dimers from their progenitor Ka class monomers). However, most monomers are identified as Ka, which indicates SF7. In orangutan centromeres, most monomers are identified as R1 and R2, which indicates SF5. In chimpanzee and human autosome and X chromosome centromeres, active arrays are formed by J1 and J2 (SF1), D1, FD and D2 (SF2), and W1–W5 (SF3) monomers. The only exception uncovered in this paper is the centromere of chimpanzee chromosome 5, which appears to be formed by R1 and R2 (SF5), with some monomers identified as J4 and Ga. The former belongs to SF01, which represents the generation of α-satellite intermediate between the progenitor SF5 and the more derived SF1, and J4 is particularly close to the R1 monomer. Moreover, the other SF01 monomers, such as J3, J5 and J6, are absent in the array, which indicates that it is not genuine SF01. Thus, the J4 monomer in chimpanzee centromere 5 should be considered variant R1. Similarly, occasional Ga monomers belong to SF4, which is the direct progenitor of SF5, and Ga is very close to R2. Ga monomers dispersed in the SF5 array are therefore just misclassed R2 monomers. The whole chimpanzee chromosome 5 α-satellite HOR array should therefore be classified as SF5, despite the abovementioned contaminations.

## Human and NHP phylogenetic analysis

Humans, chimpanzees, orangutans and macaques diverged over a period of at least 25 million years, with chimpanzees diverging approximately 6 million years ago[29], orangutans 12–16 million years ago[29] and macaques ~25 million years ago[78]. Despite these divergence times, all primates retain α-satellite repeats, which permit the phylogenetic analysis of these regions and an estimation of their evolutionary trajectory. To assess the phylogenetic relationship between α-satellite repeats in human and NHP genomes, we first masked every non-α-satellite repeat in the CHM1, CHM13, HG00733, chimpanzee, orangutan and macaque centromere assemblies using RepeatMasker[65] (v.4.1.0). We then subjected the masked assemblies to StringDecomposer[77] using α-satellite monomers derived from the T2T-CHM13 reference genome[4] (v.2.0). This tool identifies the location of α-satellite monomers in the assemblies, and we used this to extract the α-satellite monomers from the HOR/dimeric array and monomeric regions into multi-FASTA files. We randomly selected 100 and 50 α-satellite monomers from the HOR/dimeric array and monomeric regions, respectively, and aligned them with MAFFT[79,80] (v.7.453). We used IQ-TREE[81] (v.2.1.2) to reconstruct the maximum-likelihood phylogeny with model selection and 1,000 bootstraps. The resulting tree file was visualized in iTOL[82].

To estimate sequence divergence along the pericentromeric regions, we first mapped each NHP centromere assembly to the CHM13 centromere assembly using minimap2[50] (v.2.17-r941) with the following parameters: -ax asm20 --eqx -Y -t 8 -r 500000. We then generated a BED file of 10 kb windows located within the CHM13 centromere assembly. We used the BED file to subset the BAM file, which was subsequently converted into a set of FASTA files. FASTA files contained at least 5 kb of sequence from one or more NHP centromere assemblies mapping to orthologous chromosomes. Pairs of human and NHP sequences were realigned using MAFFT[79,80] (v.7.453) with the following command: mafft --maxiterate 1000 --localpair. Next, we calculated the SNV density and Ti/Tv ratios from these alignments, limiting our analysis to only those regions with one-to-one unambiguous mapping and excluding segmental duplications and satellite repeats (Supplementary Table 10). As a control, we also calculated the SNV density and Ti/Tv ratios from 500 uniquely mapping regions across the genomes (Supplementary Table 11). We estimated the sequence divergence using the Tamura-Nei substitution model[83], which accounts for recurrent mutations and differences between transversions and transitions as well as within transitions. The mutation rate per segment was estimated using Kimura's model of neutral evolution[84]. In brief, we modelled the estimated divergence ($D$) as a result of between-species substitutions and within-species polymorphisms, that is:

$$D = 2\mu t + 4N_e\mu$$

where $N_e$ is the ancestral human effective population size, $t$ is the divergence time for a given human–NHP pair and $\mu$ is the mutation rate. We assumed a generation time of [20, 29] years and the following divergence times: human–macaque = $[23 \times 10^6, 25 \times 10^6]$ years, human–orangutan = $[12 \times 10^6, 14 \times 10^6]$ years, human–chimpanzee = $[4 \times 10^6, 6 \times 10^6]$ years. To convert the genetic unit to a physical unit, our computation also assumes $N_e = 10,000$ and uniformly drawn values for the generation and divergence times.

## Human-specific phylogenetic analysis

To determine the phylogenetic relationship and divergence times between centromeric regions from chromosomes 5, 7 and 10–14 in the CHM1, CHM13 and 56 other diverse human genomes (sequenced and assembled by the HPRC[10] and HGSVC[23]), we first identified contigs with complete and accurately assembled centromeric α-satellite HOR arrays, as determined by RepeatMasker[65] (v.4.1.0) and NucFreq[22] analysis. We then aligned each of these contigs to the T2T-CHM13 reference genome[4] (v.2.0) using minimap2[50] (v.2.24). We also aligned the chimpanzee whole-genome assembly to the T2T-CHM13 reference genome[4] (v.2.0) to serve as an outgroup in our analysis. We identified 20 kb regions in the flanking monomeric α-satellite or unique regions on the p- or q-arms and ensured that the region we had selected had only a single alignment from each haplotype to the reference genome. We next aligned these regions to each other using MAFFT[79,80] (v.7.453) with the following command: mafft --auto --thread {num_of_threads} {multi-fasta.fasta}. We used IQ-TREE[81] (v.2.1.2) to reconstruct the maximum-likelihood phylogeny with model selection and 1,000 bootstraps. The resulting tree file was visualized in iTOL[82]. Timing estimates were calculated by applying a molecular clock based on the branch-length distance to individual nodes and assuming a divergence time between human and chimpanzee of 6 million years ago. Clusters of α-satellite HOR arrays with a single monophyletic origin were assessed for gains and losses of α-satellite base pairs, monomers, HORs and distinct structural changes manually.

## Polymorphic TE analysis

To detect polymorphic TEs between the CHM1 and CHM13 centromeric regions, we first ran RepeatMasker[65] (v.4.1.0) on the CHM1 and CHM13 centromeric regions. We then masked all satellite repeats within these regions using BEDtools[64] maskfasta (v.2.29.0). We aligned the masked CHM1 fasta to the masked CHM13 fasta using minimap2[50] and the following command: minimap2 -t {threads} --eqx -c -x asm20 --secondary=no {ref.fasta} {query.fasta}. Using the resulting PAF, we extracted the regions with structural variants that were >50 bp long. We next intersected these regions with the RepeatMasker annotation file to identify those variants that overlapped SINE, LINE or LTR repeat classes by >75%. We considered the following LINE and SINE subgroups: LINE/CR1, LINE/L1, LINE/L1-Tx1, LINE/L2, LINE/Penelope, LINE/RTE-BovB, LINE/RTE-X, SINE/5S-Deu-L2, SINE/Alu, SINE/MIR, SINE/tRNA, SINE/tRNA-Deu, SINE/tRNA-RTE. We then determined the variation in length of these regions between the two centromeric regions, and we plotted their position and length using R[68] (v.1.1.383) and the ggplot2 package[66].

## Reporting summary

Further information on research design is available in the Nature Portfolio Reporting Summary linked to this article.

## Data availability

All sequencing data generated and/or used in this study are publicly available and listed in Extended Data Table 1 with their BioProject ID, accession number (if available) and/or URL. The following BioProject accessions were used: CHM1 whole-genome assembly with complete centromeres (PRJNA975207); CHM1 PacBio HiFi data (PRJNA726974); CHM1 ONT data (PRJNA869061); CHM1 Illumina data (PRJNA246220); CHM1 strand-seq alignments (https://doi.org/10.5281/zenodo.7959305)[85]; CHM1 CENP-A ChIP–seq data (PRJNA975217); T2T-CHM13 (v.2.0) whole-genome assembly (PRJNA559484); CHM13 PacBio HiFi data (PRJNA530776); CHM13 ONT data (PRJNA559484); HG00733 PacBio HiFi data (PRJNA975575 and PRJEB36100); HG00733 ONT data (PRJNA975575, PRJNA686388 and PRJEB37264); HPRC whole-genome assemblies (https://projects.ensembl.org/hprc/); HGSVC whole-genome assemblies (https://www.internationalgenome.org/data-portal/data-collection/hgsvc2); and NHP (chimpanzee (Clint; S006007), orangutan (Susie; PR01109), and macaque (AG07107)) PacBio HiFi and ONT data (PRJNA659034). The original karyotyping imaging data for the CHM1 cell line are available from the Stowers Original Data Repository (http://www.stowers.org/research/publications/libpb-2457).

## Code availability

Custom code for the SUNK-based assembly of centromeric regions is available at GitHub (https://github.com/arozanski97/SUNK-based-contig-scaffolding)[86]. Custom code to detect hypomethylated regions within centromeric regions, termed CDRs[31], is available at GitHub (https://github.com/arozanski97/CDR-Finder)[87]. All other code is publicly available.

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

**Acknowledgements** We thank the staff at Google for basecalling CHM1 PacBio circular consensus sequencing data with DeepConsensus; P. M. Lansdorp and D. C. J. Spierings for generating and sharing the CHM1 strand-seq data; P. Hsieh for assistance with phylogenetic analyses; V. Slon for providing space and resources for the α-satellite SF and strand orientation analysis with funding from the Genetic Society of America and the Gruber foundation; Z. Zhao for comments and suggestions on figures; and T. Brown for assistance in editing this manuscript. This research was supported in part by funding from the National Institutes of Health (NIH) National Human Genome Research Institute (NHGRI) R01 HG010169 (to E.E.E.); the National Institute of General Medical Sciences (NIGMS) K99 GM147352 (to G.A.L.); the National Cancer Institute (NCI) R01 CA266339 (to J.L.G.); the Intramural Research Program of the National Human Genome Research Institute (NHGRI) at NIH (to M.R., S.K., S.N. and A.M.P.); Shanghai Jiao Tong University 2030 Program WH510363001-7 (to Y.M.); and the Center for Integration in Science of the Ministry of Aliyah, Israel (to I.A.A.). This work used the computational resources of the NIH HPC Biowulf cluster (https://hpc.nih.gov). E.E.E. is an investigator of the Howard Hughes Medical Institute.

**Author contributions** G.A.L. and E.E.E. conceived the project. G.A.L., J.K.L., K.H. and K.M.M. generated sequencing data. G.A.L. and A.N.R. analysed sequencing data and performed quality-control analyses. G.A.L. and A.N.R. generated and validated the CHM1 centromere assemblies. M.R., S.K., S.N. and A.M.P. generated the Verkko CHM1 centromere assemblies. G.A.L., A.N.R., F.R., V.A.S. and I.A.A. analysed the CHM1 centromere assemblies. T.P. performed spectral karyotyping and FISH experiments. C.R.C. performed immuno-FISH experiments. D.P. performed strand-seq and polymorphic TE analyses. G.A.L., A.N.R., D.Y. and Y.M. performed phylogenetic analyses. J.L.G., A.M.P., M.V., I.A.A. and E.E.E. supervised experiments and analyses. G.A.L., T.P., C.R.C., D.P., I.A.A. and E.E.E. developed figures. G.A.L., I.A.A. and E.E.E. drafted the manuscript. All of the authors read and approved the manuscript.

**Competing interests** S.N. is an employee of Oxford Nanopore Technologies. S.K. has received travel funds to speak at events hosted by Oxford Nanopore Technologies. E.E.E. is a scientific advisory board member of Variant Bio. The other authors declare no competing interests.

**Additional information**
**Correspondence and requests for materials** should be addressed to Evan E. Eichler.

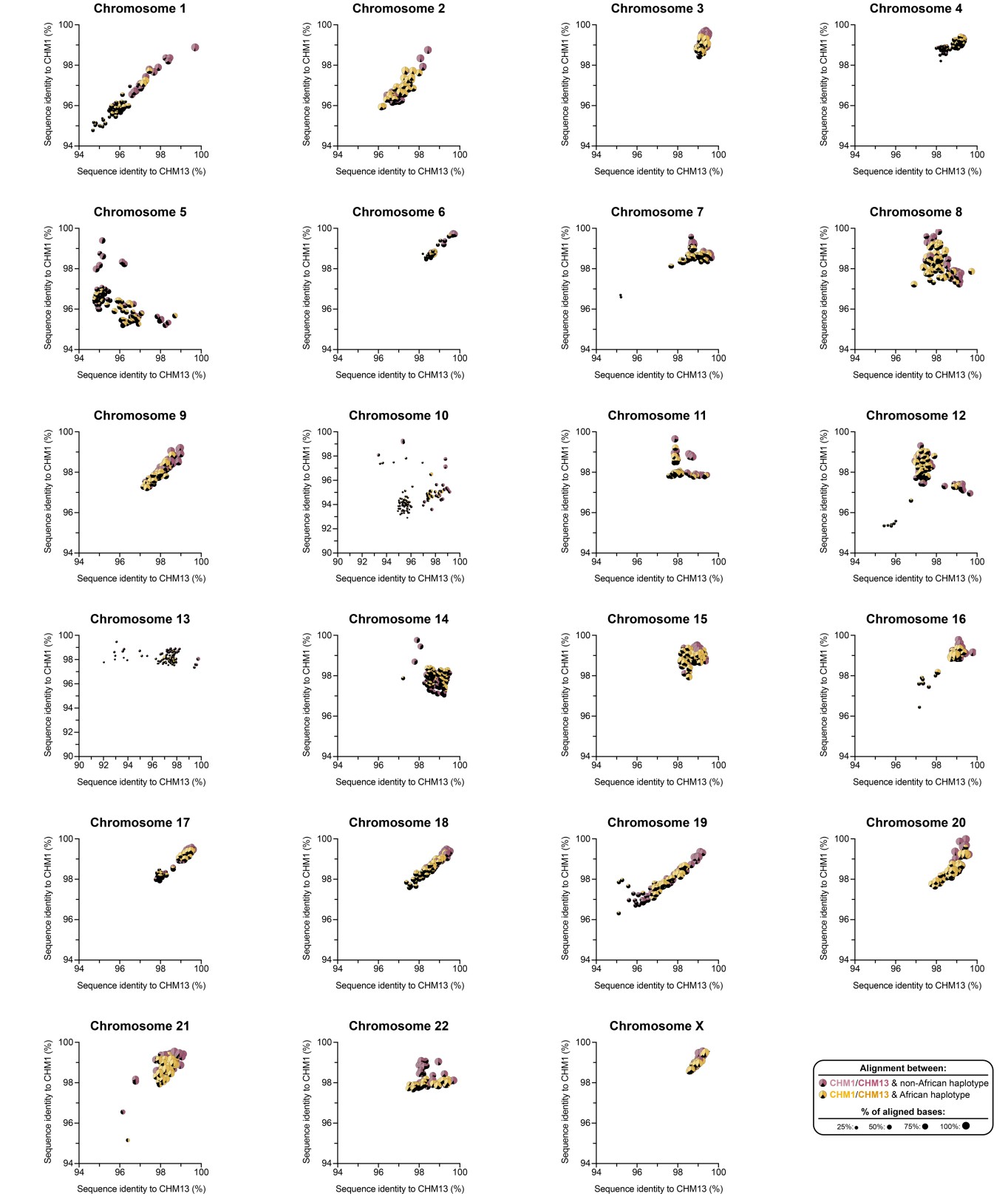

**Extended Data Fig. 1 | Variation in the sequence and structure of centromeric α-satellite higher-order repeat (HOR) arrays among 56 diverse human genomes.** Plots showing the percent sequence identity between centromeric α-satellite HOR arrays from CHM1 (y-axis), CHM13 (x-axis), and 56 other diverse human genomes [generated by the Human Pangenome Reference Consortium (HPRC)[10] and Human Genome Structural Variation Consortium (HGSVC)[23]]. Each data point shows the percent of aligned bases from each human haplotype to either the CHM1 (left) or CHM13 (right) α-satellite HOR array(s). The percent of unaligned bases are shown in black. The size of each data point corresponds to the total percent of aligned bases among the CHM1 and CHM13 centromeric α-satellite HOR arrays. Precise quantification of the sequence identity and proportion of aligned versus unaligned sequences is provided in Supplementary Table 6. Enlarged versions of these plots are shown in Supplementary Figs. 14, 15.

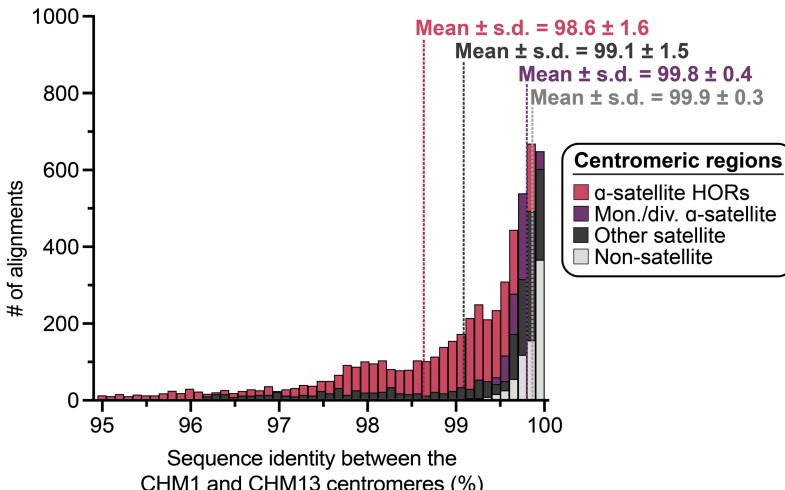

**Extended Data Fig. 2 | Sequence identities between the CHM1 and CHM13 centromeric regions.** Histogram showing the distribution of sequence identities from complete contig alignments between centromeric regions in the CHM1 and CHM13 genomes. The α-satellite HOR, monomeric/divergent α-satellite, other satellite, and non-satellite portions were assessed separately and reveal a much larger distribution in sequence identities for the α-satellite HORs. The mean and standard deviation (s.d.) are indicated.

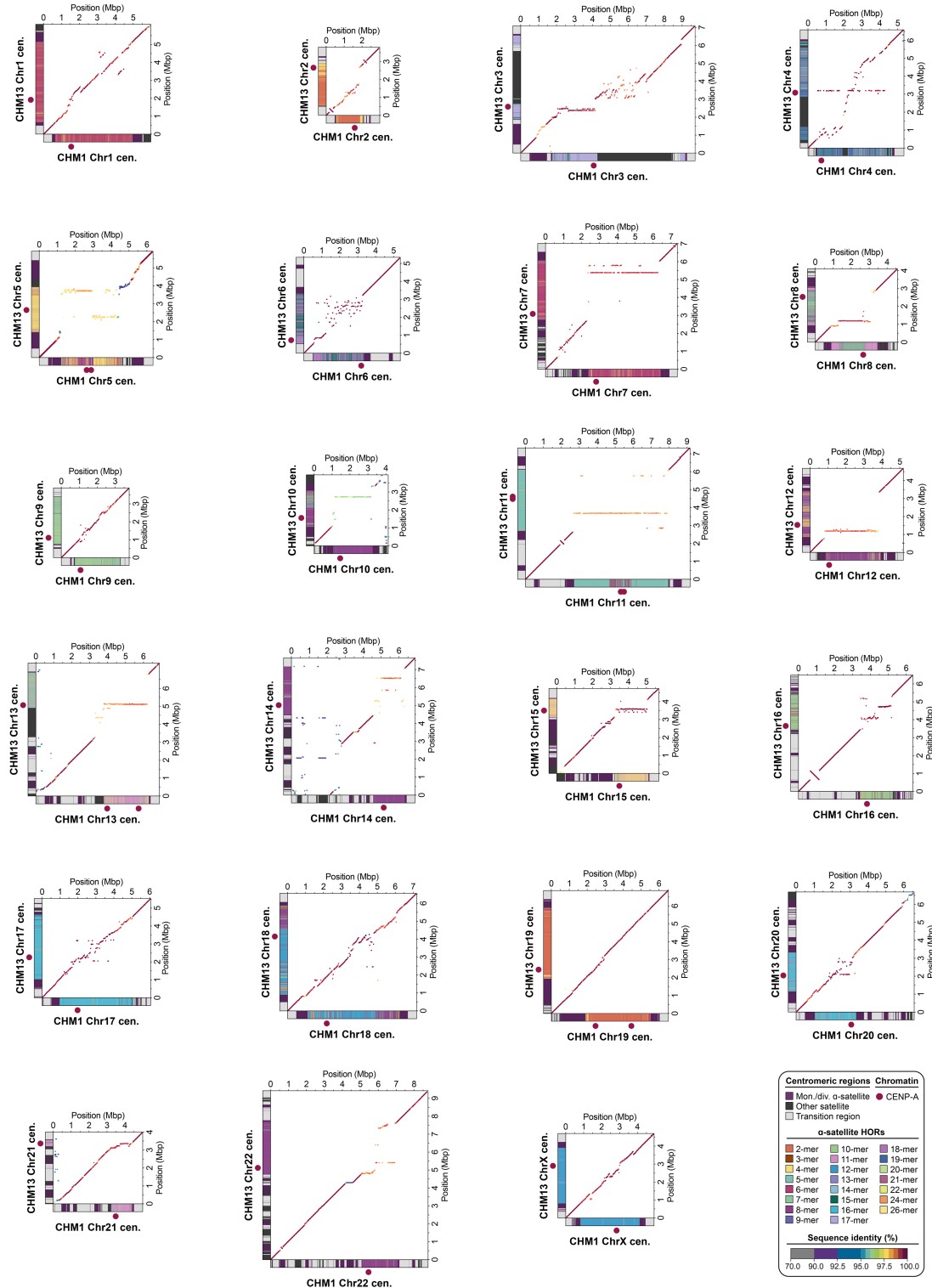

**Extended Data Fig. 3 | Comparison of the CHM1 and CHM13 centromeric regions.** Dot plots showing the percent sequence identity between the CHM1 and CHM13 centromeric regions. Plots were generated with StainedGlass[44]. Enlarged versions of these plots are shown in Supplementary Figs. 16, 17.

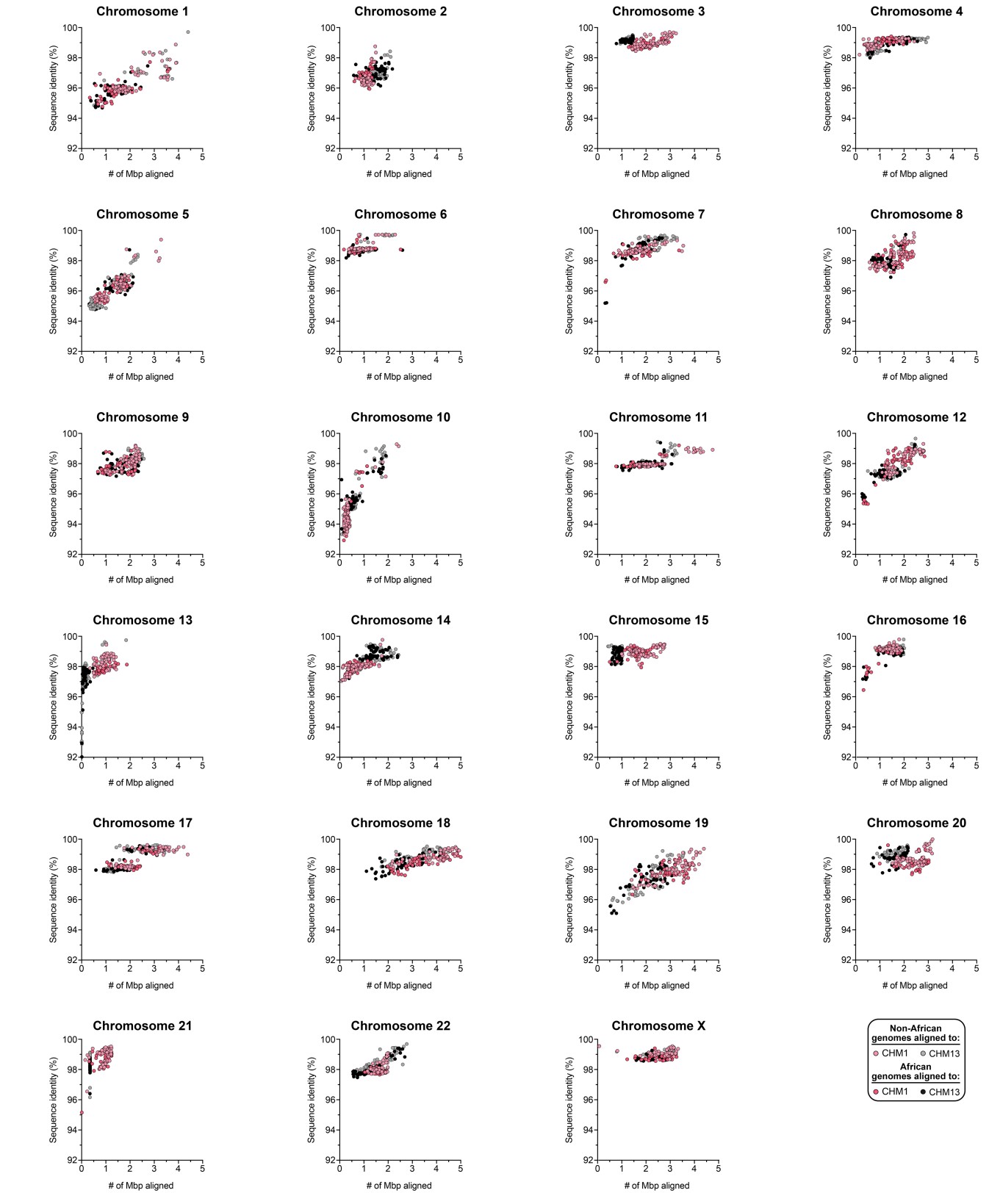

**Extended Data Fig. 4 | Comparison of CHM1 and CHM13 centromeric α-satellite HOR arrays to those from 56 diverse human genomes.** Plots showing the percent sequence identity and number of megabase pairs (Mbp) aligned for 56 diverse human genomes (112 haplotypes), generated by the HPRC[10] and HGSVC[23], mapped to the CHM1 and CHM13 centromeric regions.

Note that each data point represents a haplotype with 1:1 best mapping, although many of the centromeres are not yet complete in the HPRC and HGSVC assemblies. Enlarged versions of these plots are shown in Supplementary Figs. 18, 19.

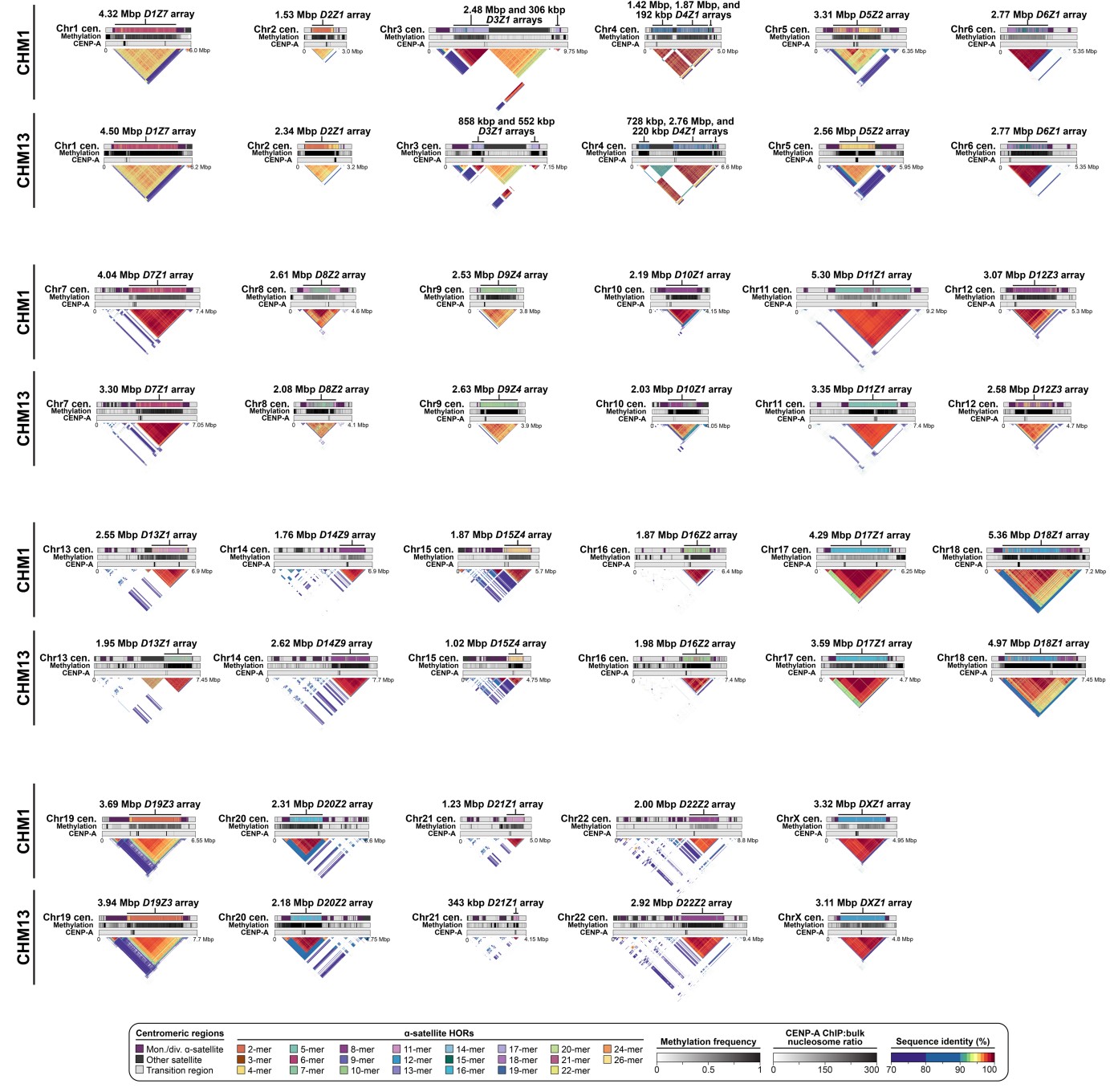

**Extended Data Fig. 5 | Comparison of the genetic, epigenetic, and evolutionary landscapes between the CHM1 and CHM13 centromeric regions.** Plots showing the sequence organization (top track), CpG methylation frequency (second track), CENP-A nucleosome enrichment (third track), and evolutionary layers (bottom triangle) for each CHM1 and CHM13 centromeric region. Enlarged versions of these plots are shown in Supplementary Figs. 45–67.

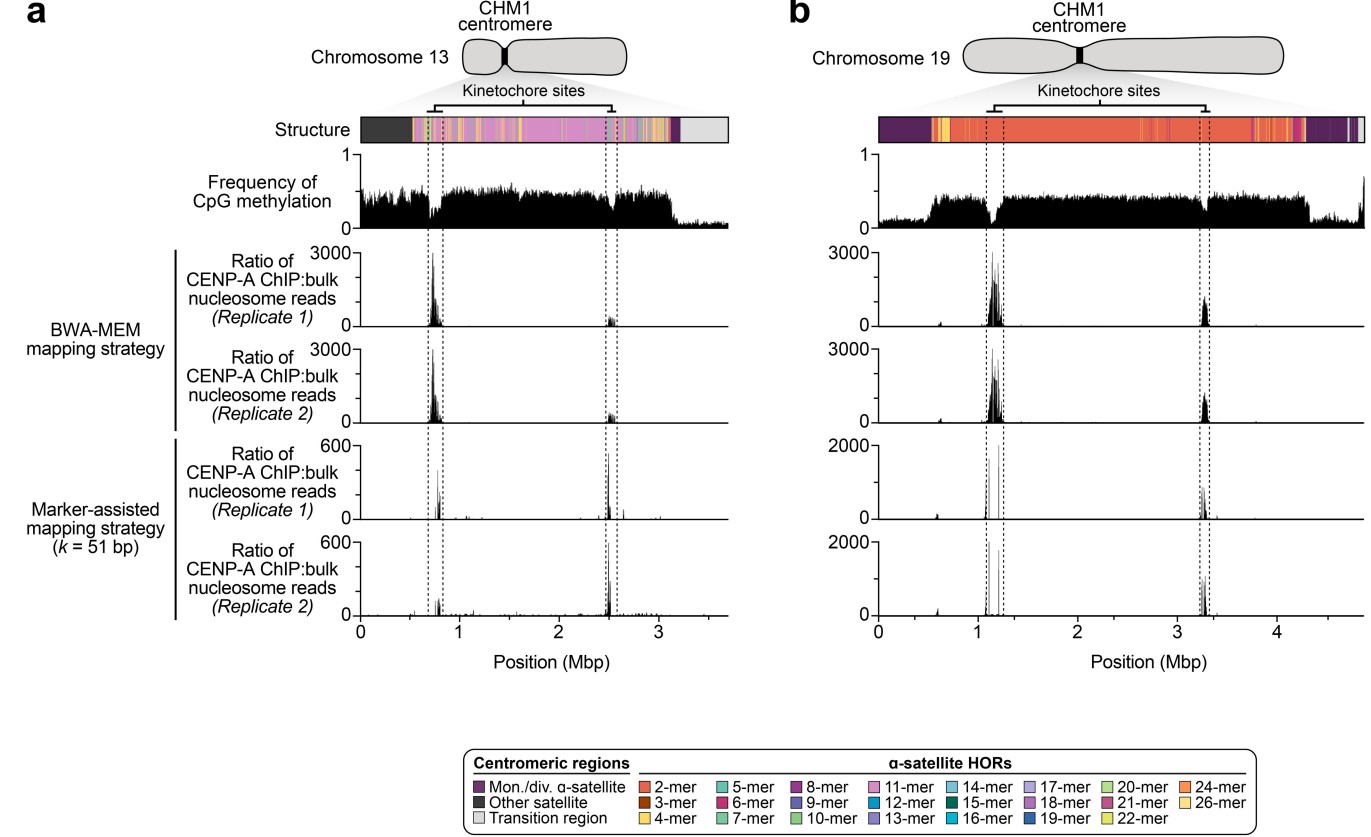

**Extended Data Fig. 6 | CHM1 chromosome 13 and 19 centromeres have two regions enriched with CENP-A chromatin within hypomethylated α-satellite DNA. a,b**) Two strategies for mapping CHM1 CENP-A ChIP-seq data (**Methods**) reveal similar patterns of CENP-A chromatin enrichment, with two regions enriched with CENP-A that coincide with hypomethylated α-satellite DNA within the CHM1 **a**) chromosome 13 and **b**) chromosome 19 α-satellite HOR arrays.

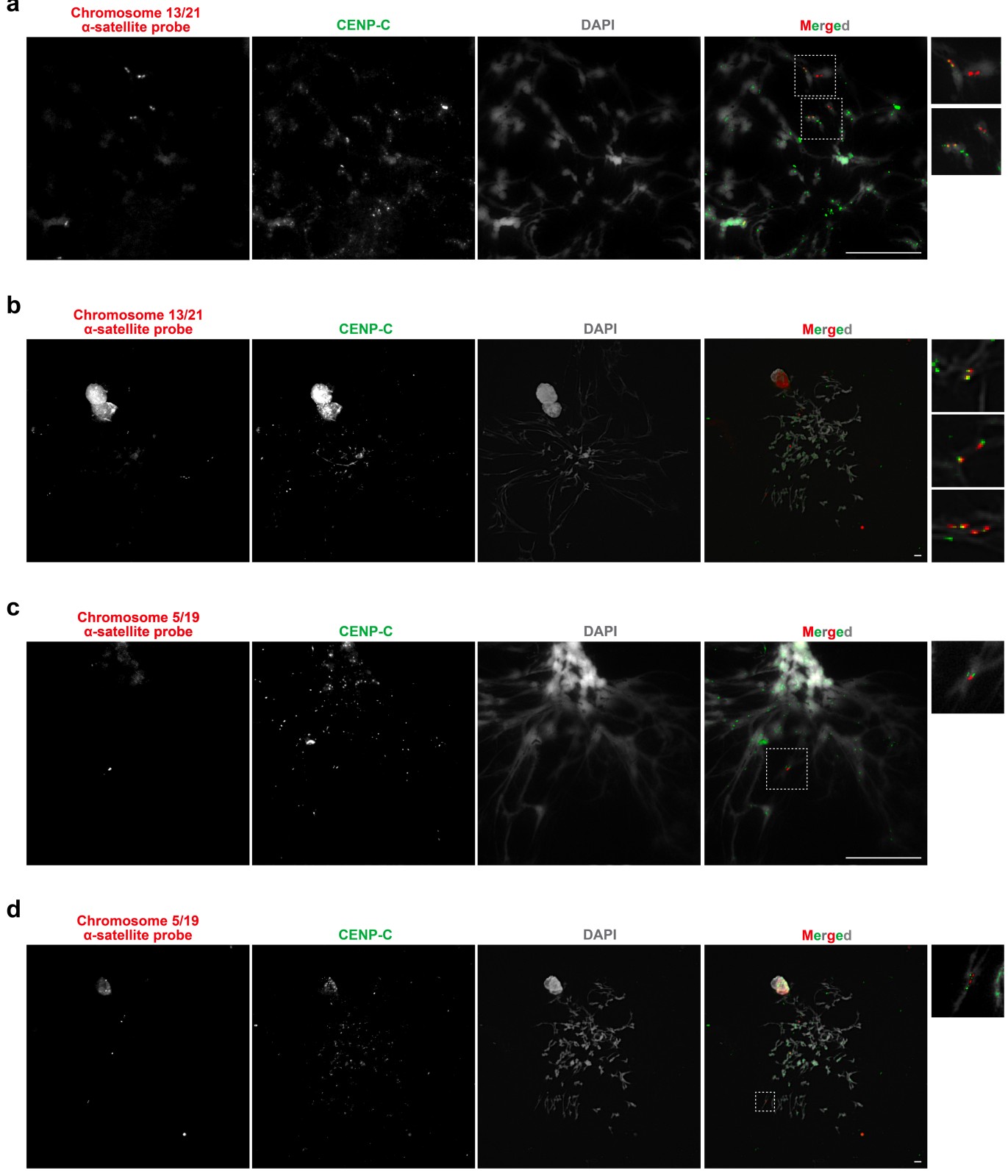

**Extended Data Fig. 7 | The CHM1 chromosome 13 centromere likely has one kinetochore site, while the CHM1 chromosome 19 centromere has two kinetochore sites. a-d)** Immuno-FISH staining of stretched metaphase chromosome spreads from CHM1 cells with a fluorescent antibody against CENP-C (an inner-kinetochore protein; green) as well as a fluorescent chromosome 13/21 α-satellite DNA probe (**a**,**b**; red) or a fluorescent chromosome 5/19 α-satellite DNA probe (**c**,**d**; red). We find that there is a single CENP-C signal that coincides with the chromosome 13/21 α-satellite probe for each chromosome 13 sister chromatid, indicating that this chromosome likely has one kinetochore (**a**,**b**). Conversely, we find that there are two CENP-C signals that coincide with a single chromosome 5/19 α-satellite probe signal for each sister chromatid, indicating there are likely two kinetochores on this chromosome (**c**,**d**). Each experiment was performed three times with similar results. n = 32 and 34 metaphase chromosome spreads were analysed for chromosomes 13 and 19, respectively. Insets are magnified 1.7-fold (panels a and c) or 3.9-fold (panels b and d). Bar, 10 μm.

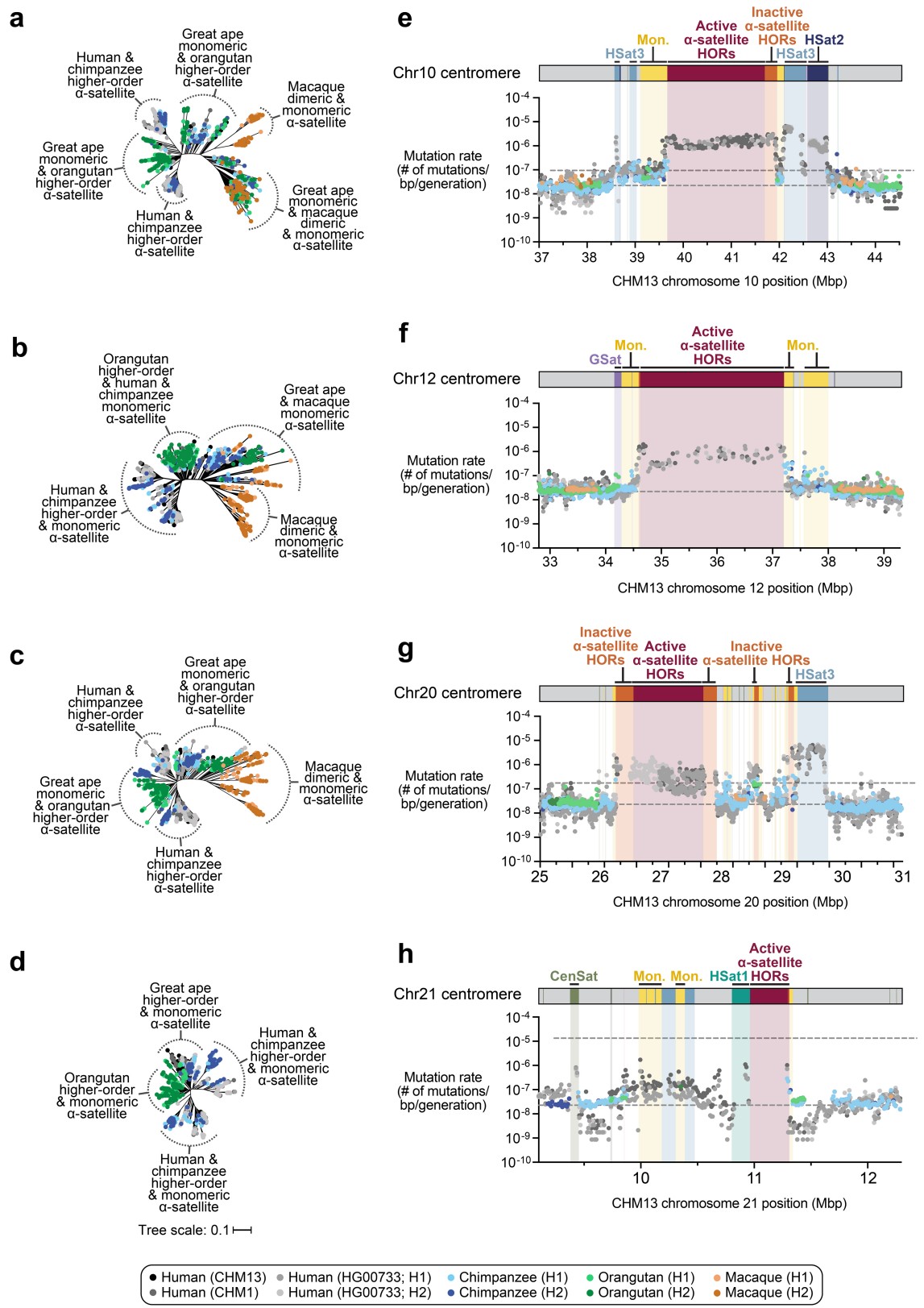

**Extended Data Fig. 8 | Centromeres evolve with different evolutionary trajectories and mutation rates. a-d)** Phylogenetic trees of α-satellite monomers derived from the human, chimpanzee, orangutan, and macaque chromosome **a)** 10, **b)** 12, **c)** 20, and **d)** 21 centromeric regions. **e-h)** Plot showing the mutation rate of the chromosome **e)** 10, **f)** 12, **g)** 20, and **h)** 21 centromeric regions. Individual data points from 10-kbp pairwise sequence alignments are shown. We note that the regions corresponding to the active α-satellite HORs have only approximate mutation rates based on human–human comparisons, Due to unequal rates of mutation and the emergence of new α-satellite HORs, interspecies comparisons are not possible in these regions.

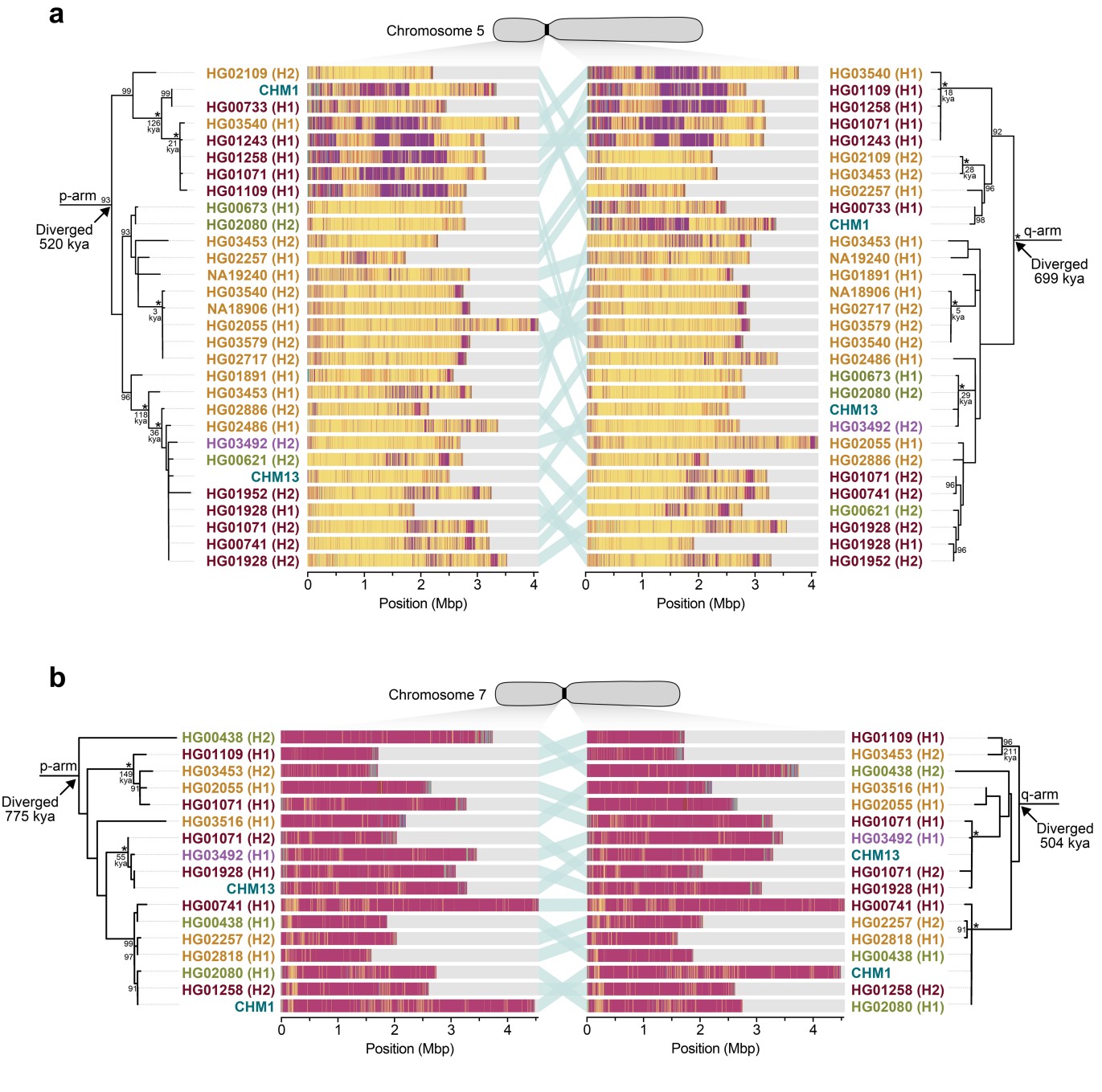

**Extended Data Fig. 9 | Phylogenetic reconstruction of human chromosome 5 and 7 centromeric haplotypes. a,b)** Phylogenetic trees showing the evolutionary relationship and estimated divergence times of completely and accurately assembled **a)** *D5Z2* α-satellite HOR arrays and **b)** *D7Z1* α-satellite HOR arrays from CHM1, CHM13, and diverse human samples (generated by the HPRC[10] and HGSVC[23]). The trees were generated from 20-kbp segments in the monomeric α-satellite or unique sequence regions on the p- (left) and q- (right) arms. Asterisks indicate nodes with 100% bootstrap support, and nodes with 90–99% bootstrap support are indicated numerically. Nodes without an asterisk or number have bootstrap support <90%. The haplotypes from the p- and the q-arm trees are linked with a light teal bar, as schematized in panel a. We note that most differences in the order of the haplotypes occur at the terminal branches where the order of sequence taxa can be readily reshuffled to establish near-complete concordance. Thus, there are no significant changes in the overall topologies of the phylogenetic trees.

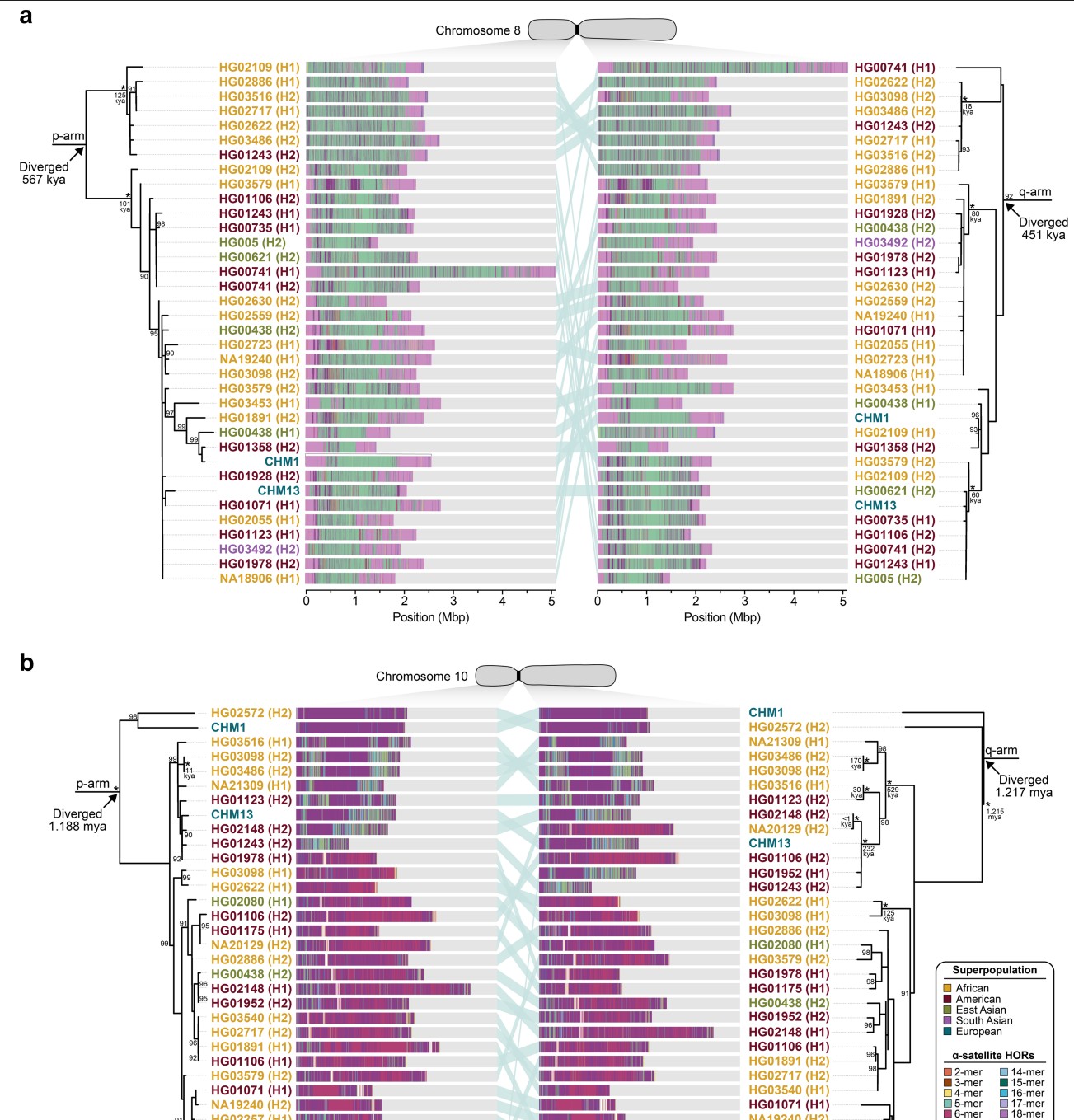

**Extended Data Fig. 10 | Phylogenetic reconstruction of human chromosome 8 and 10 centromeric haplotypes. a,b**) Phylogenetic trees showing the evolutionary relationship and estimated divergence times of completely and accurately assembled **a**) *D8Z2* α-satellite HOR arrays and **b**) *D10Z1* α-satellite HOR arrays from CHM1, CHM13, and diverse human samples (generated by the HPRC[10] and HGSVC[23]). The trees were generated from 20-kbp segments in the monomeric α-satellite or unique sequence regions on the p- (left) and q- (right) arms. Asterisks indicate nodes with 100% bootstrap support, and nodes with 90–99% bootstrap support are indicated numerically. Nodes without an asterisk or number have bootstrap support <90%. The haplotypes from the p- and the q-arm trees are linked with a light teal bar, as schematized in panel a. We note that most differences in the order of the haplotypes occur at the terminal branches where the order of sequence taxa can be readily reshuffled to establish near-complete concordance. Thus, there are no significant changes in the overall topologies of the phylogenetic trees.

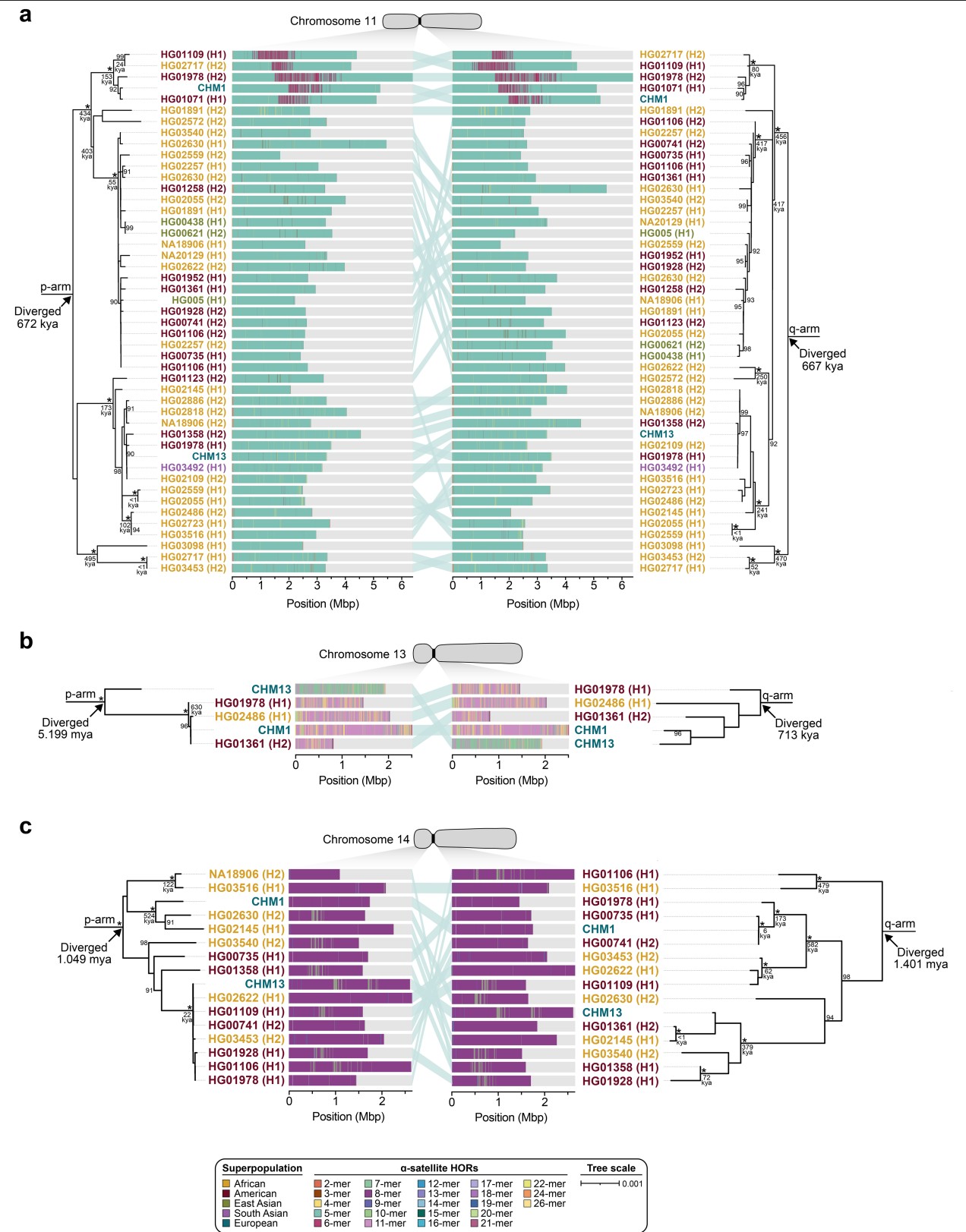

**Extended Data Fig. 11** | See next page for caption.

**Extended Data Fig. 11 | Phylogenetic reconstruction of human chromosome 11, 13, and 14 centromeric haplotypes. a-c)** Phylogenetic trees showing the evolutionary relationship and estimated divergence times of completely and accurately assembled **a)** *D11Z1* α-satellite HOR arrays, **b)** *D13Z2* α-satellite HOR arrays, and **b)** *D14Z9* α-satellite HOR arrays from CHM1, CHM13, and diverse human samples (generated by the HPRC[10] and HGSVC[23]). The trees were generated from 20-kbp segments in the monomeric α-satellite or unique sequence regions on the p- (left) and q- (right) arms. Asterisks indicate nodes with 100% bootstrap support, and nodes with 90–99% bootstrap support are indicated numerically. Nodes without an asterisk or number have bootstrap support <90%. The haplotypes from the p- and the q-arm trees are linked with a light teal bar, as schematized in panel a. We note that most differences in the order of the haplotypes occur at the terminal branches where the order of sequence taxa can be readily reshuffled to establish near-complete concordance. Thus, there are no significant changes in the overall topologies of the phylogenetic trees. We note, however, in the case of the chromosome 13 p-arm (panel b), the CHM13 divergence time is exceptional (5.2 mya) compared to all other regions of the genome. The basis for this is unknown, but it may reflect ectopic exchange of the p-arm of human acrocentric chromosomes, leading to non-homologous exchange among five human chromosomes[88].

# Extended Data Table 1 | Datasets generated and/or used in this study

| Species | Sample | Data type | BioProject | Accession # | URL | Reference |
|---|---|---|---|---|---|---|
| | | | | **Datasets generated in this study** | | |
| Human (*Homo sapiens*) | CHM1 | Whole-genome assembly with complete centromeres | PRJNA975207 | JASMMY000000000 | -- | This study |
| | | PacBio HiFi | PRJNA726974 | SRX10759865-SRX10759866 | https://s3-us-west-2.amazonaws.com/human-pangenomics/index.ht | This study |
| | | ONT | PRJNA869061 | SRX17054101-SRX17054194 | -- | This study |
| | | Strand-seq | -- | -- | https://doi.org/10.5281/zenodo.7959305 | This study |
| | | CENP-A ChIP-seq | PRJNA975217 | SRR24675260-SRR24675263 | -- | This study |
| | HG00733 | PacBio HiFi | PRJNA975575 | SRR24707288-SRR24707292 | -- | This study |
| | | ONT | PRJNA975575 | SRR24715152-SRR24715211 | -- | This study |
| Chimpanzee (*Pan troglodytes*) | Clint; S006007 | ONT | PRJNA659034 | SRR24707265-SRR24707284 | -- | This study |
| Orangutan (*Pongo abelii*) | Susie; PR01109 | ONT | PRJNA659034 | SRR24706432-SRR24706459 | -- | This study |
| Macaque (*Macaca mulatta*) | AG07107 | ONT | PRJNA659034 | SRR24709228-SRR24709293 | -- | This study |
| | | | | **Previously published datasets** | | |
| Human (*Homo sapiens*) | CHM1 | Illumina | PRJNA246220 | SRR1514950-SRR1514952 | -- | Chaisson *et al.*, *Nature*, 2015 |
| | | Whole-genome assembly | PRJNA559484 | GCA_009914755.4 | -- | Nurk *et al.*, *Science*, 2022 |
| | CHM13 | PacBio HiFi | PRJNA530776 | SRR11292120-SRR11292123 | -- | Nurk *et al.*, *Genome Res*, 2020 |
| | | ONT | PRJNA559484 | SRR10035479-SRR10035573, SRR12564439-SRR12564459, SRR12618224-SRR12618325 | -- | Logsdon *et al.*, *Nature*, 2021; Miga *et al.*, *Nature*, 2020 |
| | HG00733 | PacBio HiFi | PRJEB36100 | ERR3822935, ERR3861382-ERR3861387 | -- | Logsdon *et al.*, *Nature*, 2021; Vollger *et al.*, *Nature*, 2023 |
| | | ONT | PRJEB36388 | SRR13356541-SRR13356542, SRR13362355-SRR13362359, SRR13362450-SRR13362451 | -- | Logsdon *et al.*, *Nature*, 2021 |
| | | | PRJEB37264 | ERR3988477-ERR3988509 | -- | Shafin *et al.*, *Nature Biotechnol*, 2020 |
| | HPRC (various) | Whole-genome assembly | -- | -- | https://projects.ensembl.org/hprc/ | Liao, W.-W. *et al.*, *Nature*, 2023 |
| | HGSVC (various) | Whole-genome assembly | -- | -- | https://www.internationalgenome.org/data-portal/data-collection/hgsvc2 | Ebert *et al.*, *Science*, 2021 |
| Chimpanzee (*Pan troglodytes*) | Clint; S006007 | PacBio HiFi | PRJNA659034 | SRR12517369-SRR12517374, SRR12517378, SRR12517389-SRR12517390 | -- | Logsdon *et al.*, *Nature*, 2021 |
| | | ONT | PRJNA659034 | SRR12552556-SRR12552568 | -- | Logsdon *et al.*, *Nature*, 2021 |
| Orangutan (*Pongo abelii*) | Susie; PR01109 | PacBio HiFi | PRJNA659034 | SRR12517385-SRR12517387 | -- | Logsdon *et al.*, *Nature*, 2021 |
| | | ONT | PRJNA659034 | SRR12551266-SRR12551275 | -- | Logsdon *et al.*, *Nature*, 2021 |
| Macaque (*Macaca mulatta*) | AG07107 | PacBio HiFi | PRJNA659034 | SRR12517375-SRR12517377, SRR12517379-SRR12517381 | -- | Logsdon *et al.*, *Nature*, 2021 |
| | | ONT | PRJNA659034 | SRR12517382-SRR12517385 | -- | Logsdon *et al.*, *Nature*, 2021 |

See accompanying Excel file. ONT, Oxford Nanopore Technologies; PacBio, Pacific Biosciences; HiFi, high-fidelity; ChIP, chromatin immunoprecipitation.

# Reporting Summary

## Statistics

For all statistical analyses, confirm that the following items are present in the figure legend, table legend, main text, or Methods section.

| n/a | Confirmed | |
|---|---|---|
| ☐ | ☒ | The exact sample size (*n*) for each experimental group/condition, given as a discrete number and unit of measurement |
| ☐ | ☒ | A statement on whether measurements were taken from distinct samples or whether the same sample was measured repeatedly |
| ☐ | ☒ | The statistical test(s) used AND whether they are one- or two-sided *Only common tests should be described solely by name; describe more complex techniques in the Methods section.* |
| ☒ | ☐ | A description of all covariates tested |
| ☐ | ☒ | A description of any assumptions or corrections, such as tests of normality and adjustment for multiple comparisons |
| ☐ | ☒ | A full description of the statistical parameters including central tendency (e.g. means) or other basic estimates (e.g. regression coefficient) AND variation (e.g. standard deviation) or associated estimates of uncertainty (e.g. confidence intervals) |
| ☐ | ☒ | For null hypothesis testing, the test statistic (e.g. *F*, *t*, *r*) with confidence intervals, effect sizes, degrees of freedom and *P* value noted *Give P values as exact values whenever suitable.* |
| ☒ | ☐ | For Bayesian analysis, information on the choice of priors and Markov chain Monte Carlo settings |
| ☒ | ☐ | For hierarchical and complex designs, identification of the appropriate level for tests and full reporting of outcomes |
| ☒ | ☐ | Estimates of effect sizes (e.g. Cohen's *d*, Pearson's *r*), indicating how they were calculated |

*Our web collection on statistics for biologists contains articles on many of the points above.*

## Software and code

Policy information about availability of computer code

| | |
|---|---|
| Data collection | The software used to collect sequencing data are Pacific Bioscience Sequel II Instrument Control SW (v7.0, 7.1, and 8.0) and Oxford Nanopore Technologies PromethION software (v21.02.17 - 23.04.5). The software used to collect image data are ZEN (v3.7) and NIS-Elements AR (v3.2). |
| Data analysis | Custom code for the SUNK-based sequence assembly of centromeric regions is publicly available at https://github.com/arozanski97/SUNK-based-contig-scaffolding. Custom code to detect hypomethylated regions within centromeric regions, termed "centromere dip regions" (CDRs), is publicly available at https://github.com/arozanski97/CDR-Finder. Other publicly available software used in this study include DeepConsensus (v0.2.0), PacBio circular consensus sequencing software (v3.4.1), hifiasm (v0.16.1), HiCanu (v2.1.1), minimap2 (v2.17-r941 and v2.24), Jellyfish (v2.2.4), pbmm2 (v1.1.0), Winnowmap (v1.0), Merqury (v1.1), BWA-MEM (v0.7.17), sambamba (v1.0), SAMtools (v1.9), breakpointR (v1.18), BEDtools (v2.29.0), deepTools (v3.4.3), seqtk (v1.3), TandemAligner (v0.1), meryl (v1.3), StringDecomposer (no version specified), StainedGlass (v6.7.0), Nanopolish (v0.13.3), RepeatMasker (v4.1.0), Sickle (v1.33), Cutadapt (v1.18), MAFFT (v7.453), IQ-TREE (v2.1.2), Fiji (v2.13.1), ImageJ (v1.53k), KISS ImageJ plug-in (v1), Prism (v9.5.1), and R (v1.1.383). |

For manuscripts utilizing custom algorithms or software that are central to the research but not yet described in published literature, software must be made available to editors and reviewers. We strongly encourage code deposition in a community repository (e.g. GitHub). See the Nature Portfolio guidelines for submitting code & software for further information.

## Data

Policy information about <u>availability of data</u>

All manuscripts must include a <u>data availability statement</u>. This statement should provide the following information, where applicable:

- Accession codes, unique identifiers, or web links for publicly available datasets
- A description of any restrictions on data availability
- For clinical datasets or third party data, please ensure that the statement adheres to our <u>policy</u>

All sequencing data generated and/or used in this study are publicly available and listed in Extended Data Table 1 with their BioProject ID, accession # (if available), and/or URL. For convenience, we also list the BioProject IDs and/or URLs here: CHM1 whole-genome assembly with complete centromeres (PRJNA975207); CHM1 PacBio HiFi data (PRJNA726974); CHM1 ONT data (PRJNA869061); CHM1 Illumina data (PRJNA246220); CHM1 Strand-Seq alignments (https://doi.org/10.5281/ zenodo.7959305); CHM1 CENP-A ChIP-seq data (PRJNA975217); T2T-CHM13 (v2.0) whole-genome assembly (PRJNA559484); CHM13 PacBio HiFi data (PRJNA530776); CHM13 ONT data (PRJNA559484); HG00733 PacBio HiFi data (PRJNA975575 and PRJEB36100); HG00733 ONT data (PRJNA975575, PRJNA686388, and PRJEB37264); HPRC whole-genome assemblies (https://projects.ensembl.org/hprc/); HGSVC whole-genome assemblies (https://www.internationalgenome.org/ data-portal/data-collection/hgsvc2); and NHP [chimpanzee (Clint; S006007), orangutan (Susie; PR01109), and macaque (AG07107)] PacBio HiFi and ONT data (PRJNA659034). The original karyotyping imaging data for the CHM1 cell line is available from the Stowers Original Data Repository (http://www.stowers.org/ research/publications/libpb-2457).

## Research involving human participants, their data, or biological material

Policy information about studies with <u>human participants or human data</u>. See also policy information about <u>sex, gender (identity/presentation), and sexual orientation</u> and <u>race, ethnicity and racism</u>.

| | |
|---|---|
| Reporting on sex and gender | N/A |
| Reporting on race, ethnicity, or other socially relevant groupings | We report analyses of publicly available human genome sequencing data generated by the 1000 Genomes Project (https:// www.internationalgenome.org/home) and their associated genetic ancestry information, as established and described by the 1000 Genomes Project (https://www.internationalgenome.org/category/population/). |
| Population characteristics | See above. |
| Recruitment | See above. |
| Ethics oversight | See above. |

Note that full information on the approval of the study protocol must also be provided in the manuscript.

# Field-specific reporting

Please select the one below that is the best fit for your research. If you are not sure, read the appropriate sections before making your selection.

☒ Life sciences       ☐ Behavioural & social sciences       ☐ Ecological, evolutionary & environmental sciences

For a reference copy of the document with all sections, see nature.com/documents/nr-reporting-summary-flat.pdf

# Life sciences study design

All studies must disclose on these points even when the disclosure is negative.

| | |
|---|---|
| Sample size | We generated complete sequence assemblies of each centromere in the human CHM1hTERT genome (n=23) as well as a subset of orthologous centromeres in a second human (HG00733; n=12), chimpanzee (Pan troglodytes; Clint; S006007; n=11), orangutan (Pongo abelii; Susie; PR01109; n=10), and macaque (Macaca mulatta; AG07107; n=10). We also analyzed whole-genome assemblies from diverse humans generated by the Human Pangenome Reference Consortium (HPRC) and Human Genome Structural Variation Consortium (HGSVC; n=56 genomes; n=112 haplotypes; n=580 completely assembled centromeres; and n=2,049 incompletely assembled centromeres). For phylogenetic tree construction of centromeric regions, we used 150 data points from each genome. For centromeric mutation rate computation, we used hundreds to thousands of data points from each genome. |
| Data exclusions | No data were excluded. |
| Replication | Computational experiments are deterministic and are, therefore, reproducible. Each wet-lab experiment was performed at least two independent times. |
| Randomization | Randomization is not applicable to this study because we did not perform any experiments where there are treatment and control groups that would necessitate randomization between the subjects. |
| Blinding | Blinding is not applicable to this study because we did not perform any experiments where there are treatment and control groups that would necessitate blinding. |

# Reporting for specific materials, systems and methods

We require information from authors about some types of materials, experimental systems and methods used in many studies. Here, indicate whether each material, system or method listed is relevant to your study. If you are not sure if a list item applies to your research, read the appropriate section before selecting a response.

## Materials & experimental systems

| n/a | Involved in the study |
|-----|----------------------|
| ☐ | ☒ Antibodies |
| ☐ | ☒ Eukaryotic cell lines |
| ☒ | ☐ Palaeontology and archaeology |
| ☒ | ☐ Animals and other organisms |
| ☒ | ☐ Clinical data |
| ☒ | ☐ Dual use research of concern |
| ☒ | ☐ Plants |

## Methods

| n/a | Involved in the study |
|-----|----------------------|
| ☐ | ☒ ChIP-seq |
| ☒ | ☐ Flow cytometry |
| ☒ | ☐ MRI-based neuroimaging |

## Antibodies

| | |
|---|---|
| Antibodies used | We used a mouse monoclonal anti-human CENP-A antibody (clone 3-19; Enzo, ADI-KAM-CC006-E) in the ChIP-seq experiments. We used a rabbit polyclonal anti-human CENP-C antibody (made in house) and a goat anti-rabbit IgG antibody conjugated to FITC (Sigma F0382) in the immuno-FISH experiments. |
| Validation | The anti-human CENP-A antibody was generated against a synthetic peptide consisting of amino acids 3-19 of human CENP-A, and mutation of this epitope in human cells prevents antibody binding (Logsdon et al., JCB, 2015). |

## Eukaryotic cell lines

Policy information about cell lines and Sex and Gender in Research

| | |
|---|---|
| Cell line source(s) | The human CHM1hTERT cell line was a gift from Urvashi Surti (Pittsburgh, PA). The human HG00733 lymphoblastoid cell line was obtained from the Coriell Institute for Medical Research (Camden, NJ). Chimpanzee (Pan troglodytes; Clint; S006007) fibroblast cells were obtained from a male western chimpanzee named Clint (now deceased) at the Yerkes National Primate Research Center (Atlanta, GA) and immortalized with EBV. Orangutan (Pongo abelii; Susie; PR01109) fibroblast cells were obtained from a female Sumatran orangutan named Susie (now deceased) at the Gladys Porter Zoo (Brownsville, TX), immortalized with EBV, and stored at the Coriell Institute for Medical Research (Camden, NJ). Macaque (Macaca mulatta; AG07107) fibroblast cells were originally obtained from a female rhesus macaque of Indian origin and stored at the Coriell Institute for Medical Research (Camden, NJ). |
| Authentication | The human CHM1hTERT cell line was authenticated via STR analysis by Cell Line Genetics (Madison, WI). The human HG00733 cell line is part of the NHGRI Sample Repository for Human Genetic Research at the Coriell Institute for Medical Research (Camden, NJ) and was authenticated using a multiplex PCR assay with six autosomal microsatellite markers. The chimpanzee, orangutan, and macaque cell lines have not yet been authenticated to our knowledge. |
| Mycoplasma contamination | The human CHM1hTERT and HG00733 cell lines are negative for mycoplasma contamination. The chimpanzee, orangutan, and macaque cell lines have not yet been tested for mycoplasma contamination to our knowledge. |
| Commonly misidentified lines (See ICLAC register) | No commonly misidentified cell lines were used in this study. |

## Plants

| | |
|---|---|
| Seed stocks | *Report on the source of all seed stocks or other plant material used. If applicable, state the seed stock centre and catalogue number. If plant specimens were collected from the field, describe the collection location, date and sampling procedures.* |
| Novel plant genotypes | *Describe the methods by which all novel plant genotypes were produced. This includes those generated by transgenic approaches, gene editing, chemical/radiation-based mutagenesis and hybridization. For transgenic lines, describe the transformation method, the number of independent lines analyzed and the generation upon which experiments were performed. For gene-edited lines, describe the editor used, the endogenous sequence targeted for editing, the targeting guide RNA sequence (if applicable) and how the editor was applied.* |
| Authentication | *Describe any authentication procedures for each seed stock used or novel genotype generated. Describe any experiments used to assess the effect of a mutation and, where applicable, how potential secondary effects (e.g. second site T-DNA insertions, mosiacism, off-target gene editing) were examined.* |

# ChIP-seq

## Data deposition

☒ Confirm that both raw and final processed data have been deposited in a public database such as GEO.

☒ Confirm that you have deposited or provided access to graph files (e.g. BED files) for the called peaks.

| | |
|---|---|
| **Data access links**<br>*May remain private before publication.* | https://www.ncbi.nlm.nih.gov/sra/?term=SRR24675260<br>https://www.ncbi.nlm.nih.gov/sra/?term=SRR24675261<br>https://www.ncbi.nlm.nih.gov/sra/?term=SRR24675262<br>https://www.ncbi.nlm.nih.gov/sra/?term=SRR24675263 |
| **Files in database submission** | CHM1_CA_ChIP_1_S3_R1_001.fastq.gz<br>CHM1_CA_ChIP_1_S3_R2_001.fastq.gz<br>CHM1_CA_ChIP_2_S4_R1_001.fastq.gz<br>CHM1_CA_ChIP_2_S4_R2_001.fastq.gz<br>CHM1_Input_1_S1_R1_001.fastq.gz<br>CHM1_Input_1_S1_R2_001.fastq.gz<br>CHM1_Input_1_S2_R1_001.fastq.gz<br>CHM1_Input_1_S2_R2_001.fastq.gz |
| **Genome browser session**<br>(e.g. UCSC) | No longer applicable. |

## Methodology

| | |
|---|---|
| **Replicates** | Two independent replicates of CENP-A ChIP-seq (with chromatin input as a control) were performed on CHM1hTERT cells and were in agreement with each other. |
| **Sequencing depth** | The total number of reads generated from each CHM1hTERT CENP-A ChIP-seq experiment is as follows:<br><br>CHM1hTERT CENP-A ChIP (Replicate 1): 113,284,073 paired-end, 150x150-bp reads<br>CHM1hTERT CENP-A ChIP (Replicate 2): 82,612,743 paired-end, 150x150-bp reads<br>CHM1hTERT Input (Replicate 1): 81,452,960 paired-end, 150x150-bp reads<br>CHM1hTERT Input (Replicate 2): 90,430,891 paired-end, 150x150-bp reads |
| **Antibodies** | We used a mouse monoclonal anti-human CENP-A antibody (clone 3-19; Enzo, ADI-KAM-CC006-E) to enrich for CENP-A-containing chromatin in the CHM1hTERT cell line. |
| **Peak calling parameters** | We aligned the CHM1hTERT CENP-A ChIP and input sequencing data to the CHM1hTERT whole-genome assembly generated in this study using BWA-MEM (v0.7.17) with the following parameters: bwa mem -k 50 -c 1000000 {index} {read1.fastq.gz} {read2.fastq.gz}. The resulting SAM files were filtered using SAMtools (v1.9) with flag score 2308 to prevent multi-mapping of reads. With this filter, reads mapping to more than one location are randomly assigned a single mapping location, thereby preventing mapping biases in highly identical regions. Alignments were normalized with deepTools (v3.4.3) bamCompare with the following parameters: bamCompare -b1 {ChIP.bam} -b2 {bulk_nucleosomal.bam} --operation ratio --binSize 1000 minMappingQuality 1 -o {out.bw}. |
| **Data quality** | The CHM1hTERT CENP-A ChIP and input sequencing data were assessed for quality using FastQC (https://github.com/s-andrews/FastQC), trimmed with Sickle (v1.33; https://github.com/najoshi/sickle) to remove low-quality 5' and 3' end bases, and trimmed with Cutadapt (v1.18) to remove adapters. |
| **Software** | BWA-MEM (v0.7.17) was used to align the CHM1hTERT CENP-A ChIP and input sequencing data to the CHM1hTERT whole-genome assembly. SAMtools (v1.9) was used to remove multi-mapped reads, and deepTools (v3.4.3) bamCompare was used to normalize and filter CENP-A ChIP data relative to input data to calculate fold enrichment. |

