## [Peer Review File · Nature]

Manuscript Title: The variation and evolution of complete human centromeres

Reviewer Comments & Author Rebuttals

Reviewer Reports on the Initial Version:

Referee expertise:

Referee #1: centromeres

Referee #2: centromeres

Referee #3: evolutionary genomics

Referees' comments:

Referee #1 (Remarks to the Author):

Centromeres are essential for chromosome segregation and are frequently composed of complex tandem repeat arrays. Until recently these centromeric arrays could not be reliably assembled. The advent of long read sequencing makes this now possible and the authors have been at the forefront of assembling human and primate centromere arrays, including the recently released CHM13 human T2T genome.

In this work, these approaches are used to assemble a second human T2T genome CHM1 and a comparative genomics approach taken to infer new insights into the evolution of human centromere regions. A combination of deep HiFi and ONT sequencing are performed, combined with assembly and analytical methods, previously applied to CHM13. A very high accuracy CHM1 assembly (QV>60) is obtained and detailed validation is performed. Further, the authors compare to recent HPRC genomes, which are not centromere-complete – despite this, useful comparisons are made with the gold-standard CHM13 and CHM1 assemblies. Finally, comparison is made to three primate species to highlight inter-species centromere evolution. Through this analysis, the authors derive new insights in the genetic and epigenetic organisation of human and primate centromeres, with implications for how they evolve. These findings will be of broad interest to the genomics and genetic communities, as well as specific implications for human genome evolution.

Although the work is very strong and novel, I have several questions and suggestions for improvement.

It would be useful to understand more about the history of the CHM lines used for CHM13 and CHM1. Many of the centromeres are quite similar between these genomes, but with evidence of recent gain (or loss) of satellite arrays since divergence. Were these lines derived from the same, or a related, individual? Based on the initial comparisons to the pangenome, it seems that CHM1 and CHM13 are likely derived from at least individuals of the same demographic group? Do the authors think the satellite polymorphisms that distinguish CHM1 and CHM13 centromeres were already present in the individuals that were the source of the cell lines, or during CHM propagation?

On lines 120-121 - when comparing to the HPRC/HGSVC assemblies, it is stated that 20.9% of haplotypes match to CHM1, and that 46.9% of these match better to CHM1 than CHM13, and that this is 'confirming biological relevance of the CHM1 centromeres'. I didn't follow why this is the case? What is the definition of 'match better'? As CHM1 and CHM13 are overall quite similar, I

would not expect them to match very different haplotypes in the HPRC sample?

In the methods, three approaches to analyse sequence identity within and between centromeres are described. First, a minimap2 approach is taken using. Could the authors provide an explanation for how the minimap2 settings were selected? How sensitive are the conclusions of this analysis to varying minimap2 alignment parameters? Second, the regions were split into 10 kb fragments and aligned. Third, HORs were identified using RepeatMasker and Hum-AS-HMMER. A wide range of methods to analyse human alpha-satellites have been reported (eg HORmon, centroFlye, Alpha-CENTAURI, HiCAT, CentromereArictect), in addition to approaches that don't rely on prior knowledge of satellites (eg TRASH). How sensitive are the paper's conclusions to use of different methods? Why is the approach selected most suitable? Further information on monomer definition and HOR classing would be useful.

Later in Methods line 795 onwards, a StringDecomposer approach is taken for monomer identification – why is this used instead of Hum-AS-HMMER? Lines 811-813 – what is the definition of 'orthologous' here? I think its inherently very difficult to identify the 'same' repeat here? It seems very problematic here to attempt to say which sites are the same/orthologous, when both mutation and recombination are acting? Does the putative satellite recombination process need to be incorporated into the mutation estimations?

Line 678 – how is 'live' being used here, a clearer definition of what this classifier means would be helpful. It would be useful if a brief description of Hum-AS-HMMER could be provided and why its suitable.

Lines 152-153 – centromeres 19 and X are stated to be most concordant between CHM13 and CHM1 – is this a trend previously noted in human centromere studies, or just by virtue of this specific pairwise comparison? With the preliminary HPRC comparison, 1, 5, 10, 12, 13 and 19 are stated to be the most variable. Similarly, is there precedence for those observations in the large literature on human satellite diversity?

Line 189 – what is the definition of 'distinctly different structures'? Also in this paragraph – 'layers' are referred to – what is the definition of layers in this context?

The authors extend previous observations by showing that DNA methylation is depleted in regions of CENP-A occupancy. Have the authors attempted to detect non-CG methylation throughout the centromere arrays? Eg DeepSignal may be suitable to derive this information whereas nanopolish only has the capability to detect CG methylation? This is relevant as there is an increasing appreciation of non-CG methylation in human tissues, eg it is high in the brain, and has been connected recently to Rett's syndrome by the Bird laboratory. For these reasons I would like to know methylation status of non-CG sites in CHM1 and CHM13 centromeres. If it is present, it would be interesting to know whether it also is depleted in CENPA enriched locations?

It would also be interesting to compare DNA methylation states in regions of duplicated or expanded HORs. Do the expanded HORs share similar methylation states?

Line 220 – it is stated that CHM1 CENPA regions are significantly shorter than CHM13 – please provide a statistic for this comparison.

The relative distance from HOR-to-monomeric transition zones to the position of sites of CENPA enrichment is analysed, which are shown to be in distinct locations between CHM1 and CHM13. This is interesting and suggests migration or 'creep' of the kinetochore location over time – the authors may be interested in this paper <https://www.ncbi.nlm.nih.gov/pmc/articles/PMC5480163/> where maize CENH3 location is analysed in inbred lineages and related species and evidence for 'creeping' provided. In the case where the authors observe two CENPA peaks in their ChIP-seq data this is likely to represent two cell populations, as dicentrics are notoriously unstable (although

the proximity of the peaks within the alpha satellite arrays may permit this I guess). Could this be looked at using FISH and immunostaining perhaps? To test whether two cell CENPA states are present in the sampled cell population?

Line 738 – which species CENPA was this antibody raised against?

As the human centromeres are associated with LINEs, are there LINE or other transposons that are polymorphic between CHM1 and CHM13 either inside, or close to, the alpha satellite arrays. If there are such transposons, it would be good to indicate their positions in Fig 1 or Fig 2a. Equally, in Fig 2c, I would be interested to know how centromeric TE content varied as a function of this analysis across the chromosomes.

The authors extend analysis to comparison of primate assemblies. How heterozygous were the primate centromeres - could the authors comment on the challenges of phasing any heterozygous centromere arrays? Why were only 6 chromosomes specifically focused on here?

I think it is worth noting somewhere the time of divergence for the 4 species analysed. To what extent do the more divergent genomes show greatest divergence at the centromere level? Or are they all equally un-alike?

Line 284 – I believe centromere dimeric satellite units have been reported in monkeys previously and could be cited here?

Line 286 – 'suprachromosomal family' as a term should be more clearly defined on first usage.

Line 288 – what is the significance of SF5? Is this a repeat family observed in humans? Are these alpha satellite families that predate the split of humans and chimps? When comparing SF families between species, can evidence of concerted evolution be detected? Am I correct in understanding that all alpha satellites, across the genomes analysed, share a common ancestry, but have since diverged following speciation, although some chromosomes have diverged less than others? The methods on line 770 onwards for SF analysis do not clearly explain the approach to the uninitiated. Line 781 – what is 'Ka' and why does it indicate 'SF7', and similarly for the other designations in the next few lines?

Line 320 – please define the use of 'monomeric' more clearly – does this mean not in a HOR? This is very interesting, indicating that mutation rate is higher in areas of HOR formation? This might suggest a centromeric recombination process that is typified by relatively lower fidelity? The authors attempt to compare mutation rates between HORs and flanking regions and estimate higher rates. This is potentially very interesting, but could this be confounded by recombination processes? What is spectrum of transition and transversion classes and how does this relate to spectrums defined in other human cell types / contexts? To what extent are these estimates of mutation rate based on assumptions that orthologous 'loci/repeats' are compared and are such assumptions meaningful in the centromere? In this case the CHM1 and CHM13 cells appear to have some common ancestry, but it may be a long time since they have diverged, and they have been propagated as cell lines. I would like to see a more explicit rationale for how mutation rate is being estimated, and potential confounding effects discussed, including the difficulty in identifying the 'same' repeat and the action of recombination simultaneously to mutation in the history of these two cell lines.

Lines 342-344. An assumption of limited or no recombination across the satellite arrays is reasonable if the authors are referring to the documented absence of meiotic crossovers based on segregation of flanking markers (i.e. cenhaps). However, as the authors show, the alpha satellite arrays are polymorphic between CHM1 and CHM13, and more widely - this indicates most likely some form of DSB formation and homologous repair, including non-allelic forms, that are capable of generating these changes. Otherwise, how do the authors propose that the array polymorphisms arise? The patterns of internal centromere satellite polymorphism, embedded in

cenhaps, was also recently noted in the plant *Arabidopsis thaliana* (Włodzimierz et al 2023 Nature), suggesting these modes of centromere evolution are conserved as far as plants and humans, which I think would be worth adding to the discussion.

A more explicit model of recombination may also be valuable to consider in relation to mutation estimates, as the observed polymorphisms will be influenced by both mutation and recombination rates. Indeed, Profs Alexandrov and Miga have previously published a kinetochore associated recombination machine (KARM) relating to this, which it might be illuminating to discuss in light of CHM1 vs CHM13. Although I note that in Fig3c, the new HORs are more widely distributed than the kinetochore site, although, perhaps they are being 'pushed out' from the CENPA centre where KARM would putatively be located? Or the kinetochore used to be in different locations to that observed here?

Figure 2a – It would be interesting to also see StainedGlass plots for these comparisons.

Line 386 – how is 'center' defined here? Is it the physical centre of the alpha-satellite array? Is it in relation to CENPA?

Could the authors comment on HSat array variation between CHM1 and CHM13? Are the Hsat arrays evident in the pang genome assemblies?

In Figure 1 the repeats are coloured by n-mer composition. Are all say 9-mers the same, or is it possible to have independent 9-mers on different chromosomes?

Line 412. This is an interesting discussion on different evolutionary rates between different centromeres. Can differential selection be ruled out, instead or, or in addition to, varying mutation and/or recombination rates?

Lines 422 – how karyotypically different are these genomes? Do they have the same number of chromosomes? What has happened to centromeres when they have been gain or lost since speciation?

Line 432 – what do the authors hypothesize causes the higher apparent mutation rate in centromere 5?

In Figure 5 – it would be interesting to also have some stainedglass analysis comparing across species, eg Chr5 compared in the same stainedglass plot for all 5 assemblies?

Lines 72-73 – please provide reference(s) for this statement. Similarly at the end of sentence ending line 76.

Lines 144-145 – how are 'euchromatin' and 'heterochromatin' being defined in this statement – a more precise definition would be beneficial.

Referee #2 (Remarks to the Author):

The study is primarily interesting because they fully assembled centromeres of a second human genome and compared them with CHM13 and other primates, as well as insights into the genetic variation and evolutionary trajectories of human centromeres. Extending previous findings reported by Altemose et al. 2022, they show that centromeric alpha-satellite arrays are highly dynamic in humans by comparing with additional 56 human genomes. The mutation rate analysis of centromeres and its use as a proxy to estimate the separation times of human centromeric haplotypes is very interesting and a pioneer, despite being likely underestimated. In summary, the study significantly contributes to the understanding of centromere biology by providing

comprehensive and accurate centromere sequences, characterizing genetic variation among human centromeres, and shedding light on the evolutionary dynamics of centromeres across primate species. The results underscore the complexity and heterogeneity of centromeres, emphasizing the need for further research to elucidate their functional significance and contribution to genome stability. The figures are well-prepared and self-explaining.

The analysis of genetic variation among human centromeres revealed significant heterogeneity in sequence identity and structure. The comparison between CHM1 and CHM13 centromeres demonstrated that sequence identity increases from heterochromatin to euchromatin regions. The α -satellite higher-order repeat (HOR) arrays, which are characteristic of centromeres, exhibited varying degrees of sequence identity between the two haplotypes. Furthermore, when comparing centromeres from diverse human genomes, the researchers found considerable variation in α -satellite HORs, indicating the emergence of new HOR structures in some haplotypes but not others. This highlights the extensive single-nucleotide and structural diversity of human centromeres.

The length and organization of α -satellite HOR arrays were found to vary among centromeres. CHM1 arrays were, on average, larger than their CHM13 counterparts, and specific chromosomes showed significant differences in size and structure. The analysis also revealed the presence of evolutionary layers within the α -satellite HOR arrays, indicating complex dynamics of HOR expansions and contractions. The comparison with 56 incompletely assembled reference genomes further confirmed the chromosome-specific nature of centromere variations and highlighted the unique patterns of α -satellite HOR organization and size for different chromosomes.

The study also investigated epigenetic differences between CHM1 and CHM13 centromeres, particularly focusing on the kinetochore regions. The kinetochore is a proteinaceous complex critical for chromosome segregation, and it resides within the hypomethylated centromere dip region (CDR) in humans. The researchers observed differences in the size and position of kinetochore sites between the two haplotypes, with several sites located hundreds of kilobases apart. The examination of underlying sequences revealed both conserved and divergent regions associated with kinetochores, suggesting complex evolutionary dynamics.

To gain insights into the evolutionary history of centromeres across primate lineages, the researchers sequenced and assembled orthologous centromeres from four primate species. The analysis of centromeres in chimpanzee, orangutan, and macaque genomes allowed for the reconstruction of evolutionary trajectories over millions of years. The comparison of centromeres revealed species-specific differences in α -satellite HOR organization and highlighted the presence of evolutionary layers within the arrays. The findings suggest different mutation rates and diverse evolutionary trajectories among primate centromeres.

The methods were performed adequately as mentioned for:

1. Well-described pipeline for complete assembly of centromeres and subsequent analysis
2. ChIP-seq properly performed with two replicates

Below are my comments:

Major concerns:

The potential "discovery" of two kinetochores on chromosome 13 and chromosome 19 centromeres in the CHM1 genomes is due to the presence of two hypomethylated regions enriched with CENP-A chromatin.

The authors discuss this in terms of "likely represents two populations of cells, which may have arisen due to a somatic mutation, resulting in differing epigenetic landscapes". However, it cannot

be excluded they actually are present in the same cell, as it has been shown before that centromeric sites that are not too far apart from each other can still stably function as a monocentromere (https://www.nature.com/articles/ng1198_227; <https://link.springer.com/article/10.1007/s10577-012-9302-3>,).

On this topic, I noticed that Altemose et al. (2022) have used a so-called "Marker-assisted mapping strategy" that can localize CENP-A enrichment specifically to one of two large macro-repeat structures with recent HOR expansions. I wonder if the different mapping strategy used in the present study is responsible for the two centromere peaks found in Chr13 and chr19 as they were not found in Altemose et al., but rather on Chr4, which in contrast not found here...

Page 5 Lines 218-220: I wonder if "... Although CHM1 centromeric α -satellite HOR arrays are typically larger, the majority of CHM1 kinetochore sites (18 out of 23) are smaller than their CHM13 counterparts, with an average size of 178 versus 214 kbp...", it is or partially due to the differential mapping strategies applied.

The authors refer to their ChIPseq mapping as follows: "The resulting SAM files were filtered using SAMtools (v1.9) 764 with flag score 2308 to prevent multi-mapping of reads. With this filter, reads mapping to more than one location are randomly assigned a single mapping location, thereby preventing mapping biases in highly identical regions."

This mapping strategy can be commonly applied to mapping to centromeric repeats assuming a more homogenous centromere organization. However, in such cases of two centromere peaks appearing the author should perform more strict mapping strategies.

Minor concerns:

The authors mention an average increase of 1.3-fold in HOR of CHM1 compared to CHM13, but I miss how many CHM13 were in fact larger than in CHM1.

Altemose et al. (2022) have detected smaller regions of CENP-A enrichment outside of the primary CDR, with some overlapping a minor, secondary CDR (chr 4, chr16, and chr22) or no CDR at all (chr18). It would be nice to have a comparison of these chromosomes here.

Page 6 Line 258 – Please correct Mbp for Gbp

Figure 5 bottom legend – Dark green dots for Orangutan should be H2 and not H1, right?

Limitations of the study that could be better addressed:

I am not really aware of possible limitations on that, but despite all the efforts in sequencing and assembly of the CHM1 centromeres, it would have been better to have chosen a natural human sample than rather a cell line with somatic rearrangements. Despite all the authors' efforts in checking the integrity of the CHM1 cell lines, the somatic rearrangements found can certainly have an influence on the results obtained.

ChIPseq for the other primates' centromeres. The study would have been more complete if they had carried out ChIPseq for the primates as well. This would give a clear picture, at a deeper resolution, about the evolution of centromeres in the group.

André Marques

Referee #3 (Remarks to the Author):

Centromeres have been largely excluded from genomic studies due to their extremely repetitive and complex sequence compositions, and the first fully resolved sequences of human centromeres have only recently become available. The authors of this manuscript present a second completely resolved set of human centromere sequences, together with completely resolved sequences for 6 chromosomes in 3 non-human primate species. These data enable the first high-resolution assessment of sequence variation in centromeres between human individuals by comparing the completely resolved complements in addition to partial complements from a diversity panel of other human genomes, and an assessment of the recent evolutionary history of primate centromeres. The authors detect large-scale variability in human centromeric sequences, including array sizes, array composition, and kinetochore positioning, as well as multiple new alpha-satellite HORs and a nearly complete turnover of alpha-satellite HORs in primates.

The study is well presented, and most technical aspects are, in my opinion, sound. It represents a significant addition to our understanding of human centromere biology and the results are likely to be of substantial interest to the human genetics and comparative genomics community. I do however have several questions and comments I hope the authors could address (not in order of importance).

I strongly encourage the authors to reconsider some of their coloring schemes, particularly those used to classify alpha-sat HOR-mers. The current scheme uses repeated or very similar divergent colors for different parts of what is a continuous scale. In Fig 1. it is not possible to distinguish between 6/11/17/18/24-mers or 5/14/15-mers, among others. If there is a requirement to use this scale, it is not evident from the text.

Several plots in the extended data are too cramped to make the visualizations useful. I understand the wish to include all chromosomes on the same page, but the subpanels end up being too small to be readable, despite their importance for the manuscript's conclusions. Examples include Ext. Figs 4,5,7,11.

Page 3, Line 107: The authors use several lines of evidence to ensure that potential somatic alterations in CHM1 do not generate issues with their downstream analyses. They show that a significant fraction of cells contains an aberrant karyotype and several genomic rearrangements, which is not evident from the main text. Given that at least for the tetraploid CHM1 cells chromosomal segregation did at some point not function correctly and the observed rearrangements contain deletions of tumor suppressors, it seems this might be an important caveat regarding the representative nature of the assemblies. It is furthermore not clear why the same degree of caution was not necessary for the NHP cell lines, which are at least in part EBV immortalized and thus potentially subject to the same issues.

P3 L109 / Ext. Fig. 4: The figure legend claims "uniform read depth, indicating a lack of large structural errors" and states two exceptions. This statement is insufficiently backed up by the figures, and gauging by eye there are several more, although the figure is difficult to see due to its small size. Examples include chr3 Mb 4-8.5 (PB), chr3 Mb ~2.5-3 (ONT), chr1 Mb ~5.5-6 (PB), chr8 Mb ~2.3-2.8 (ONT), among several other peaks and dips. Could the authors please produce read-depth histograms for these regions and specify what size cutoff they consider 'large-scale'? I would also be interested in reading their thoughts for the underlying reasons, particularly for the event on chr3. All comments equally apply to the NHP data presented in Ext Figs 15-16, which show similar patterns.

P3 L126 / Ext. Data Table 3: There are substantial differences between, but also within the same alignment strategy when switching query and reference for both identity and alignability. As several important subsequent results are based on this, could the authors please justify their choice of alignment strategy? Why was the tandem-repeat-aware strategy dropped? Please also clarify how the sequence identity was calculated, as the choice of denominator can influence these

values significantly.

P4 149: I'm confused by this sentence. Are the authors suggesting the numbers presented in the preceding sentences do not reflect allelic variation?

Fig 2a: Please increase the dot sizes, it is currently not possible to distinguish the underlying colors in many cases.

P4 L159 / Fig2b-c / Ext. Fig 7 / Ext. Fig 10: Do the observed patterns of divergence & diversity on the centromeres recapitulate what is known about the population divergence of these 56 samples based on other markers? While the current analyses of human diversity observing higher sequencing identity for either CHM1 or CHM13 and diversity of array lengths underline the extreme variation, it feels like a missed opportunity to better understand the sources and stratification of variability (particularly beyond the subdivision into "African" and "non-African" haplotypes). For Fig 2b / Ext. Fig 10 could the authors please include information on the proportion of the query that aligns?

Ext. Fig7 is not readable at the current size, do these fractions makes sense considering what is known about the ancestries for CHM1/CHM13?

For Fig. 2c. have the authors explored sources of length variation? Given the extreme differences, is there any evidence that this might encompass not only germline but also somatic variation?

P5 L174: Could the authors please comment/hypothesize on the underlying reasons for mostly larger centromeres (and often significantly so) in CHM1 vs CHM13?

P6 L253: Please clarify the criteria used to select the specific chromosomes. Does this set encapsulates the previously described patterns of diversity observed across human centromeres?

P7 L292: Could the authors please provide more details on how this inversion was detected, as there is no mention in the methods? Is this based solely on the orientation of the satellite sequence, and were there any orthogonal analyses?

P8 L324: The presented model assumes that mutations accumulate at an equal rate on both branches of each human-NHP pair. The authors need to more carefully consider whether this assumption is justified, especially given the extreme variability they observe. It also would be helpful to see the distributions of mu stratified by species and sequence category, as these differences are difficult to gauge from the scatterplots only.

P8 L345: Please include a distance metric to understand what "remarkably similar" corresponds to, and visually connect corresponding haplotypes in Fig. 7b-c. Do nodes without annotated bootstrap support values (assuming that is what the number corresponds to) have full support? What do the asterisks correspond to?

P8 L334: I'm surprised by the sparsity of comments on the divergence dating for different haplotypes the authors present in Fig. 7b-c and Ext Figs 18-20, many of which exceed the coalescent times for AMH. Am I correctly interpreting that the authors suggest that e.g. on the chr13 p-arm the CHM13 haplotype diverged from other ones 5.2 Mya (<1Mya after the chimp divergence used as calibration) ago and has been maintained ever since? If so, these results are very surprising even for the relatively young dates. Can the authors please clarify how exactly these dates were calculated, as this is not evident from the methods (beyond using chimp to calibrate)? If a simple molecular clock scaled to the human-chimp divergence was used, the authors need to consider whether this is adequate in the face of the extreme variability of sequence, structure, and mutation rates they claim. Are the authors surprised about the tree topology given what we know about the history of the underlying superpopulations, and if not why?

P8 L334: The ML tree estimates are sensitive to the alignment quality and correctly established orthology, which is challenging in the face of repetitive regions. Can the authors please clarify the chosen model and present statistics on missing data (gaps) for individuals and alignment columns? Were these alignments filtered in any way prior to the tree inference? The same comment applies to the results presented in Fig. 6a-c and corresponding Ext. Figs. Was Tamura-Nei chosen after testing its adequacy for this data?

Author Rebuttals to Initial Comments:

Authors' responses to referees' comments:

Referee #1 (Remarks to the Author):

Centromeres are essential for chromosome segregation and are frequently composed of complex tandem repeat arrays. Until recently these centromeric arrays could not be reliably assembled. The advent of long read sequencing makes this now possible and the authors have been at the forefront of assembling human and primate centromere arrays, including the recently released CHM13 human T2T genome.

In this work, these approaches are used to assemble a second human T2T genome CHM1 and a comparative genomics approach taken to infer new insights into the evolution of human centromere regions. A combination of deep HiFi and ONT sequencing are performed, combined with assembly and analytical methods, previously applied to CHM13. A very high accuracy CHM1 assembly (QV>60) is obtained and detailed validation is performed. Further, the authors compare to recent HPRC genomes, which are not centromere-complete – despite this, useful comparisons are made with the gold-standard CHM13 and CHM1 assemblies. Finally, comparison is made to three primate species to highlight inter-species centromere evolution. Through this analysis, the authors derive new insights in the genetic and epigenetic organisation of human and primate centromeres, with implications for how they evolve. These findings will be of broad interest to the genomics and genetic communities, as well as specific implications for human genome evolution.

Although the work is very strong and novel, I have several questions and suggestions for improvement.

It would be useful to understand more about the history of the CHM lines used for CHM13 and CHM1. Many of the centromeres are quite similar between these genomes, but with evidence of recent gain (or loss) of satellite arrays since divergence. Were these lines derived from the same, or a related, individual? Based on the initial comparisons to the pangenome, it seems that CHM1 and CHM13 are likely derived from at least individuals of the same demographic group? Do the authors think the satellite polymorphisms that distinguish CHM1 and CHM13 centromeres were already present in the individuals that were the source of the cell lines, or during CHM propagation?

This is a great question. CHM1 and CHM13 were derived from unrelated individuals; however, both appear to be largely from the same European demographic group based on ancestry information and PCA (Nurk et al., *Science*, 2022, and Vollger et al., *Nature*, 2023). The tissue from both of them were obtained decades ago (1981 for CHM1 and prior to 2001 for CHM13), and the cell lines were established via hTERT transformation in the early 2000s (2001 for CHM1 and 2001-2002 for CHM13). Both have gone through extensive passaging (CHM1 more so than CHM13). However, almost all of the α -satellite HOR structures and polymorphisms observed in CHM1 and CHM13 have been identified in other HPRC and HGVC samples and, thus, we believe that both standards are very biologically relevant and were present in the original sperm that sourced the cell lines. We have included this information in **Supplementary Note 1**, which now reads as follows (changes underlined):

“Supplementary Note 1. Isolation, immortalization, and karyotype analysis of the CHM1 cell line. CHM1hTERT (abbr. CHM1) cells were originally isolated from a hydatidiform mole at Magee-Womens Hospital (Pittsburgh, PA) in 1981 and subsequently immortalized via transformation with human telomerase reverse transcriptase (hTERT) in 2001. Analysis of the CHM1 cell line has shown that it is

primarily of European origin (Vollger et al., *Nature*, 2023), similar to the CHM13 cell line that was collected and established around the same time (Nurk et al., *Science*, 2022). To determine the karyotype of the CHM1 cell line, we used three orthogonal methods: Giemsa staining, spectral karyotyping, and single-cell sequencing of template DNA strands (Strand-seq; **Extended Data Figs. 2,3**)...

On lines 120-121 - when comparing to the HPRC/HGSVC assemblies, it is stated that 20.9% of haplotypes match to CHM1, and that 46.9% of these match better to CHM1 than CHM13, and that this is 'confirming biological relevance of the CHM1 centromeres'. I didn't follow why this is the case? What is the definition of 'match better'? As CHM1 and CHM13 are overall quite similar, I would not expect them to match very different haplotypes in the HPRC sample?

When we align the HPRC/HGSVC assemblies to the CHM1 assembly, we find that 20.9% of the assemblies match with $\geq 99\%$ sequence identity to the CHM1 α -satellite HOR arrays, indicating that the sequences and structures present in the CHM1 arrays are also found in other human genomes. If, for example, we had observed much lower sequence identity when aligning the HPRC/HGSVC assemblies to the CHM1 assembly ($< 95\%$), this would indicate that the CHM1 α -satellite HOR arrays are not representative of those in other genomes and would not be of much biological relevance. However, because we don't observe this, we conclude that the CHM1 centromeres are biologically relevant.

When we compare the length and sequence identity of the alignments of the HPRC/HGSVC assemblies to the CHM1 and CHM13 centromeres, we find that $\sim 47\%$ of the assemblies are a "better match" to the CHM1 centromeres. The phrase "better match" is defined as having a higher alignment score to one set of centromeres over the other. We calculate the alignment score as follows:

$$\frac{(\text{total \# of aligned bases in the query})}{(\text{total \# of bases in the reference}) * (\text{mean sequence identity by event})}$$

where the mean sequence identity by event is:

$$\frac{(\text{\# of matches})}{(\text{\# of matches} + \text{\# of mismatches} + \text{\# of insertion events} + \text{\# of deletion events})}$$

To make this clearer, we have added a new paragraph to the Methods that describes how we calculate the alignment score in order to identify which set of centromeres is a "better match" to those from the HPRC/HGSVC assemblies, which we now cite in the main text. This section reads as follows:

“**Better match**” analysis

To determine whether the CHM1 or CHM13 centromeres are a better match to those from the 56 diverse human genomes assembled by the HPRC⁷ and HGSVC²¹, we performed a pairwise sequence alignment between contigs from the HPRC and HGSVC assemblies to either the CHM1 or CHM13 assembly using minimap2⁴⁷ (v2.24) and the following command: `minimap2 -I 15G -K 8G -t {threads} -ax asm20 --secondary=no --eqx -s 2500 {ref.fasta} {query.fasta}`. We filtered the alignments using SAMtools⁵⁶ (v1.9) flag 4, which keeps primary, secondary, and partial alignments and then calculated an alignment score between each pair of haplotypes, limiting our analysis to only the centromeric α -satellite HOR arrays, as follows: $(\text{total \# of aligned bases in the query}) / (\text{total \# of bases in the reference}) * (\text{mean sequence identity by$

event) . The mean sequence identity by event is calculated as follows: $(\# \text{ of matches}) / (\# \text{ of matches} + \# \text{ of mismatches} + \# \text{ of insertion events} + \# \text{ of deletion events})$. The set of centromeres with a higher alignment score was determined to be a better match to that haplotype than the other set of centromeres.”

In the methods, three approaches to analyse sequence identity within and between centromeres are described. First, a minimap2 approach is taken using. Could the authors provide an explanation for how the minimap2 settings were selected? How sensitive are the conclusions of this analysis to varying minimap2 alignment parameters? Second, the regions were split into 10 kb fragments and aligned. Third, HORs were identified using RepeatMasker and Hum-AS-HMMER. A wide range of methods to analyse human alpha-satellites have been reported (eg HORmon, centroFlye, Alpha-CENTAURI, HiCAT, CentromereArictect), in addition to approaches that don't rely on prior knowledge of satellites (eg TRASH). How sensitive are the paper's conclusions to use of different methods? Why is the approach selected most suitable? Further information on monomer definition and HOR classing would be useful.

Because the alignment of α -satellite HOR arrays is particularly challenging, we chose three very different approaches in an effort to determine if, in fact, the general observations and conclusions could be reached independently of the precise algorithm applied. The first alignment approach (minimap2) was the most stringent because it forced the query to align to the reference, but we allow many-to-one alignments in order to accommodate tandem repeat expansions in the query. The second approach, 10-kbp fragments, was more flexible because it essentially allowed the best mapping positions not to be contiguous—allowing α -satellite blocks to map to their best location in the HOR array. The third approach, TandemAligner, is a repeat-aware aligner that takes advantage of rare k -mer substrings in both the reference and query to properly align repetitive regions. While the precise % sequence identity computed differed, the relative magnitude of the effect was generally quite consistent and proportional, supporting the general observations that we report in the paper.

We chose the following minimap2 parameters after testing several options and identifying optimal ones for alignment between repetitive and/or structurally divergent regions in diploid human genomes: `-I 15G -K 8G -ax asm20 --secondary=no --eqx -s 2500`. Specifically, we chose `-I 15G` to provide additional memory for aligning between centromeric regions (the default is 4G and sometimes throws an error because of the large number of potential alignments). We also chose `-K 8G` because it allows for 8 Gbp of sequence to be loaded into memory at a time. This is enough for a typical human diploid genome (~6 Gbp) to be loaded. If we had left it at the default (500M), only a subset of contigs would be loaded at a time, and once the shortest contigs align, we would be left with only one thread aligning the longest contig. Therefore, we chose to increase this parameter so that the whole assembly is aligned at one time. We also chose to use `-ax asm20` because it allows for sequences that are up to 20% divergent to be aligned. This is more permissive to alternative α -satellite HOR structures and sequence compositions than the other alignment options (e.g., `asm5` and `asm10`). We also opted to use `--secondary=no` to prevent secondary alignments from the same contig, thereby preventing multi-mapping and ensuring that the query would only align once to the reference. We added `--eqx` to allow us to parse the CIGAR string and calculate the mean sequence identity of the alignments. Finally, we selected `-s 2500` as the minimal peak dynamic programming alignment score. The default setting for this parameter is 40, and we tested that one as well as 1000, 2500, and 5000. We found that with `-s 40` and `-s 1000`, spurious alignments occurred from other centromeres, and with `-s 5000`, accurate

alignments from centromeres were filtered out. Therefore, we chose `-s 2500` to allow for diverse α -satellite HOR structures to align without some alignments being filtered out. We have added this more detailed explanation to the Methods section, which now reads as follows (changes underlined):

“In the first analysis, we performed a pairwise sequence alignment between contigs from the CHM1, CHM13, and diverse genomes using minimap2⁴⁷ (v2.24) and the following command: `minimap2 -I 15G -K 8G -t {threads} -ax asm20 --secondary=no --eqx -s 2500 {ref.fasta} {query.fasta}`. We chose these minimap2 parameters after testing several options and identifying optimal ones for alignment between repetitive and/or structurally divergent regions in diploid human genomes. Specifically, we chose `-I 15G` to provide additional memory for aligning between centromeric regions (the default is 4G and sometimes throws an error because of the large number of potential alignments). We also chose `-K 8G` because it allows for 8 Gbp of sequence to be loaded into memory at a time. This is enough for a typical human diploid genome (~6 Gbp) to be loaded. If we had left it at the default (500M), only a subset of contigs would be loaded at a time, and once the shortest contigs align, we would be left with only one thread aligning the longest contig. Therefore, we chose to increase this parameter so that the whole assembly is aligned at one time. We also chose to use `-ax asm20` because it allows for sequences that are up to 20% divergent to be aligned. This is more permissive to alternative α -satellite HOR structures and sequence compositions than the other alignment options (e.g., `asm5` and `asm10`). We also opted to use `--secondary=no` to prevent secondary alignments from the same contig, thereby preventing multi-mapping and ensuring that the query would only align once to the reference. We added `--eqx` to allow us to parse the CIGAR string and calculate the mean sequence identity of the alignments. Finally, we selected `-s 2500` as the minimal peak dynamic programming alignment score. The default setting for this parameter is 40, and we tested that one as well as 1000, 2500, and 5000. We found that with `-s 40` and `-s 1000`, spurious alignments occurred from other centromeres, and with `-s 5000`, accurate alignments from centromeres were filtered out. Therefore, we chose `-s 2500` to allow for diverse α -satellite HOR structures to align without some alignments being filtered out. After generating the alignments, we ...”

While there are many tools designed to annotate centromeric regions (such as HORmon, Alpha-CENTAURI, HiCAT, or CentromereArchitect), all of these tools require a list of α -satellite monomers as input, which could potentially limit or bias our results to only those monomers that have already been identified in other studies. Therefore, we chose to use a different tool, HumAS-HMMER, because it is able to identify new α -satellite HOR structures that have never been observed before as well as identify split α -satellite monomers that are derived from at least two different monomers. Both of these cases were observed in the CHM1 centromeres. Additionally, using HumAS-HMMER allowed us to directly compare the CHM1 centromere annotation to that of the CHM13 centromeres, which were reported in Altemose et al., *Science*, 2022, so it was also chosen for consistency and ease of comparison.

Another tool that has been developed for centromere assembly is centroFlye (Bzikadze & Pevzner, *Nat Biotechnol*, 2020). centroFlye uses long, error-prone reads (such as ONT reads) for centromere assembly, which results in assemblies with a similar base accuracy as an ONT read (~93-99%). Our method uses PacBio HiFi reads to generate an initial assembly, which is then scaffolded with ultra-long ONT reads. Because PacBio HiFi reads have a base accuracy of >99% and the contigs that are generated from the corrected HiFi reads have a base accuracy >99.99%, our approach results in assemblies that are higher in base accuracy than those typically produced by centroFlye.

Similarly, we chose not to implement TRASH (Wlodzimierz, et al., *Bioinformatics*, 2023) in our analyses because TRASH was mainly designed to assess tandem repeats, and our study focuses on the comparison between α -satellite repeats as well flanking sequences in the p- and q-arms (which typically include monomeric α -satellite, other satellites, and unique sequences). The flanking sequences of each centromere serve as a control and a phylogenetic anchor in our analyses and allow us to assess sequence divergence in the α -satellite HOR arrays relative to the neighboring sequences in the p- and q-arms. Thus, we chose to use a more sequence-agnostic approach for alignments and analyses across entire centromeric regions.

The α -satellite monomer and HOR classifications are extensively described in the supplemental materials of Altemose et al., *Science*, 2022 (pages 10-16), and we now cite this article in the Methods when describing the α -satellite nomenclature and classification. This section now reads as follows (changes underlined):

“Sequence composition and organization of α -satellite HOR arrays

To determine the sequence composition and organization of each α -satellite HOR array in the CHM1, CHM13, and 56 diverse genome assemblies^{7,21}, we ran HumAS-HMMER (<https://github.com/fedorrik/HumAS-HMMER> for AnVIL) on centromeric contigs with the default parameters and parsed the resulting BED file with StV (<https://github.com/fedorrik/stv>). This generated a BED file with each α -satellite HOR sequence composition and its organization along the α -satellite HOR arrays. We used the stv_row.bed file to visualize the organization of the α -satellite HOR arrays with R⁶⁵ (v1.1.383) and the ggplot2 package⁶⁶. The α -satellite monomer and HOR classification generated with HumAS-HMMER is described in detail in the supplemental material of Ref. 4, and we refer our readers to that publication for a more complete description of these annotations.”

Later in Methods line 795 onwards, a StringDecomposer approach is taken for monomer identification – why is this used instead of Hum-AS-HMMER? Lines 811-813 – what is the definition of ‘orthologous’ here? I think its inherently very difficult to identify the ‘same’ repeat here? It seems very problematic here to attempt to say which sites are the same/orthologous, when both mutation and recombination are acting? Does the putative satellite recombination process need to be incorporated into the mutation estimations?

We used StringDecomposer to identify monomers in both human and nonhuman primate centromeres because it is the only tool designed to be compatible with all primates species studied here, while HumAS-HMMER is only compatible with human (see our response above). StringDecomposer is very efficient at identifying α -satellite monomers when given a single consensus α -satellite sequence; therefore, it was used to identify the α -satellite monomers in primate centromere assemblies for downstream phylogenetic analyses.

Thank you for drawing our attention to our ambiguous usage of “orthologous”. You are correct that the orthology for the α -satellite repeats cannot be unambiguously determined, so we define “orthologous” as sequences originating from the same orthologous chromosomes. In the text, we meant to refer to the pericentromeric regions themselves as orthologous because the chromosomes they reside on are >95% identical in sequence, while the α -satellite repeats have diverged and mutated independently across the different lineages. We have revised this section so that the word “orthologous” describes the chromosomes themselves, not the 5 kbp sequences in the pericentromeric regions (changes underlined):

“FASTA files contained at least 5 kbp of sequence from one or more NHP centromere assemblies mapping to orthologous chromosomes. Pairs of human and NHP sequences were realigned using MAFFT^{78,79} (v7.453) and the following command: `mafft --maxiterate 1000 --localpair.`”

Incorporation of recombination and other mutational processes would be ideal, though it should be noted that we find no evidence of standard allelic recombination. In fact, the topology of the p- and q-arm trees are essentially identical, consistent with suppression of recombination, and the non-allelic or illegitimate recombination mechanisms are not well-enough understood to incorporate in a biologically meaningful way.

Line 678 – how is ‘live’ being used here, a clearer definition of what this classifier means would be helpful. It would be useful if a brief description of Hum-AS-HMMER could be provided and why its suitable.

“Live” or “active” α -satellite HORs are those that belong to an array that consistently associates with the kinetochore in multiple individuals, as defined by McNulty et al., *Chromosome Res*, 2018, and Altemose et al., *Science*, 2022. This is in contrast to “dead” or “inactive” α -satellite HORs, which have not been found to associate with the kinetochore and are typically more divergent in sequence than the “live” or “active” HORs.

To clarify this in the text, we have revised the Methods section as follows (changes underlined):

“Estimation of α -satellite HOR array length

To estimate the length of the α -satellite HOR arrays of each centromere in the CHM1, CHM13, and 56 diverse genome assemblies^{7,21}, we first ran RepeatMasker⁶² (v4.1.0) on the assemblies and identified contigs containing α -satellite repeats, marked by “ALR/Alpha”. We extracted these α -satellite-containing contigs and subsequently ran HumAS-HMMER (https://github.com/fedorrik/HumAS-HMMER_for_AnVIL) on each of them. HumAS-HMMER is a tool that identifies the location of α -satellite HORs in human centromeric sequences. It uses a hidden Markov model (HMM) profile for centromeric α -satellite HOR monomers and generates a BED file with the coordinates of the α -satellite HORs and their classification. Using this BED file, we extracted contigs containing α -satellite HORs that were designated as “live” or “active” (denoted with an “L” in the HumAS-HMMER BED file), which are those that belong to an array that consistently associates with the kinetochore in several individuals^{4,56}. In contrast, “dead” or “inactive” α -satellite HORs (denoted with a “d” in the HumAS-HMMER BED file), are those that have not been found to be associated with the kinetochore and are usually more divergent in sequence than the “live” or “active” arrays. We filtered out contigs that had incomplete α -satellite HOR arrays (e.g., those that did not traverse into unique sequence), thereby limiting our analysis to only complete α -satellite HOR arrays. Additionally, we assessed the integrity of each of the α -satellite HOR array-containing contigs with NucFreq²⁰ to ensure that they were completely and accurately assembled, filtering out those with evidence of a deletion, duplication, or misjoin in sequence. Finally, we calculated the length of the α -satellite HOR arrays in the remaining contigs by taking the minimum and maximum coordinate of the “live” or “active” α -satellite HOR arrays and plotting their lengths with Graphpad Prism (v9.5.1).”

Lines 152-153 – centromeres 19 and X are stated to be most concordant between CHM13 and CHM1 – is this a trend previously noted in human centromere studies, or just by virtue of this specific pairwise

comparison? With the preliminary HPRC comparison, 1, 5, 10, 12, 13 and 19 are stated to be the most variable. Similarly, is there precedence for those observations in the large literature on human satellite diversity?

Extended Data Fig. 7 shows the variability of centromere sequence and structure across 56 genomes (112 haplotypes). While the chromosome X centromere is more conserved across diverse genomes, the chromosome 19 centromere is only similar among a subset of genomes (including CHM1 and CHM13).

To our knowledge, only two human centromeres have been previously studied in depth in order to understand their diversity. The first is the chromosome 17 centromere, which has been studied by Beth Sullivan's group and shown to have variation in the length of the three α -satellite HOR arrays that exist in tandem on this chromosome as well as the position of the kinetochore (Aldrup-MacDonald et al., *Genome Res*, 2016; Maloney et al., *PNAS*, 2012). The other is chromosome X, which was extensively studied by Hunt Willard's group and has been the focus of more recent studies by the T2T Consortium, where it was shown to have relatively low sequence and structural variation among a subset of human genomes (Altemose et al., *Science*, 2022). We were able to find one paper that investigates the variation of the chromosome 19 centromere (Crossen, *Clinical Genetics*, 1975), and this paper states that among a population of cells from a single individual, the "centromeric banding pattern of chromosome 19 exhibited considerable variation". However, it did not compare the centromeric banding of this chromosome among multiple individuals, so it is still unclear how variable the centromeric region is among the human population.

Line 189 – what is the definition of 'distinctly different structures'? Also in this paragraph – 'layers' are referred to – what is the definition of layers in this context?

'Distinctly different structures' refers to a different organization of α -satellite HORs in the array and/or the presence of new α -satellite HOR variants that change the structure of the array. To make this clearer, we have changed the phrase to 'distinctly different α -satellite HOR array structures' in order to emphasize that the array has different structures between haplotypes. This section now reads as follows (changes underlined):

"Comparison of the CHM1 and CHM13 centromeres identifies eight with distinctly different α -satellite HOR array structures (chromosomes 5, 7, 8, and 10-14; **Figs. 3b,c, Extended Data Fig. 12**)."

A 'layer' here refers to an 'evolutionary layer', or a stretch of sequence that has evolved separately from neighboring sequences and for which there is usually evidence of it being displaced over evolutionary time (i.e., there are pockets of sequence homology bracketing the most recently evolved layer). We first used this term when describing the CHM13 chromosome 8 centromere (Logsdon et al., *Nature*, 2021), which has five distinct evolutionary layers. We used this term here again to draw attention to the stretch of sequences in the center of the arrays that have higher sequence identity with each other than the rest of the centromere, as well as the surrounding sequences, which form their own evolutionary layers. In **Fig. 3b,c**, we also use the term "evolutionary layers" to show the mirror symmetry organization of the centromere, with an evolutionarily younger layer in the core (with higher sequence identity within itself) and more divergent sequences on the periphery. To make this clearer, we have changed the word "layer" to "evolutionary layer" in the main text and edited the legend of **Fig. 3** to state the definition of "evolutionary layer". This legend now reads as follows (changes underlined):

“Similarly, the CHM1 chromosome 11 *D11Z1* α -satellite HOR array contains a 6-monomer HOR variant that is much more abundant than in the CHM13 array and comprises a new evolutionary layer, or a stretch of sequence that has evolved separately from neighboring sequences and has pockets of homology flanking it (Layer 4; indicated with an arrow), although this 1.21 Mbp segment is more highly identical to the flanking sequence. The inset shows each of the new evolutionary layers with a higher stringency of sequence identity, as well as the relative position of the kinetochore.”

The authors extend previous observations by showing that DNA methylation is depleted in regions of CENP-A occupancy. Have the authors attempted to detect non-CG methylation throughout the centromere arrays? Eg DeepSignal may be suitable to derive this information whereas nanopore only has the capability to detect CG methylation? This is relevant as there is an increasing appreciation of non-CG methylation in human tissues, eg it is high in the brain, and has been connected recently to Rett’s syndrome by the Bird laboratory. For these reasons I would like to know methylation status of non-CG sites in CHM1 and CHM13 centromeres. If it is present, it would be interesting to know whether it also is depleted in CENPA enriched locations?

We attempted to install and run DeepSignal, but it currently only has a trained model for CpG methylation detection (see: <https://github.com/PengNi/deepsignal2/tree/master#trained-models>). We discussed this with the developers of DeepSignal, who confirmed that it is unable to detect non-CpG methylation motifs at this time. Instead, we ran Remora, which is able to detect 5hmC on ONT data. Unfortunately, the results were still inconclusive. While an interesting consideration, we believe the tools for detecting such non-CpG methylation patterns are still not yet fully mature.

It would also be interesting to compare DNA methylation states in regions of duplicated or expanded HORs. Do the expanded HORs share similar methylation states?

To address this question, we focused on two centromeres that have recently expanded α -satellite HORs (chromosomes 5 and 11) to determine if the same DNA methylation pattern is observed on each expanded HOR. We found that the expanded α -satellite HORs in the CHM1 chromosome 5 and 11 centromeres have similar, but not identical, 5mC patterns (see **Extended Data Fig. 25** below). For the chromosome 5 centromere, we find that ~66.6% of recently expanded 8-mers have the same 5mC pattern, while the remaining ~33.3% differ (usually a gain or loss of one 5mC on an α -satellite repeat within the HOR; **Extended Data Fig. 25a-c**). Similarly, we find that ~56.3% of 6-mer HORs in the CHM1 chromosome 11 centromere have one predominant 5mC pattern, while the remaining ~44.7% of 6-mer HORs show another pattern. These two patterns differ by three 5mCs, all within the third and sixth α -satellite monomers of the HOR (**Extended Data Fig. 25d-f**). Thus, we find that, despite expansion or duplication of identical α -satellite HORs within a centromeric array, the HORs can have different CpG methylation patterns.

Extended Data Figure 25. Expanded α -satellite HORs in the CHM1 chromosome 5 and 11 centromeres have divergent CpG methylation patterns. a,d) CpG methylation patterns on recently expanded α -satellite HORs in the core of the CHM1 a) chromosome 5 centromere and d) chromosome 11 centromere. b,e) CpG methylation patterns on individual α -satellite monomers from the HORs within the core of the CHM1 b) chromosome 5 centromere or e) chromosome 11 centromere. c,f) Unique CpG methylation patterns and their frequencies within the recently expanded α -satellite HORs within the CHM1 c) chromosome 5 centromere and f) chromosome 11 centromere.

Line 220 – it is stated that CHM1 CENPA regions are significantly shorter than CHM13 – please provide a statistic for this comparison.

This section states that “...the majority of CHM1 kinetochore sites (18 out of 23) are smaller than their CHM13 counterparts, with an average size of 178 versus 214 kbp, respectively.” To test the significance of this, we performed a Kolmogorov-Smirnov test, which compares the distribution of the kinetochore site lengths. and found that this difference is not statistically significant ($p = 0.194$). We have modified **Fig. 4a** to indicate that this difference is not significant (see below) and revised the legend to reflect this as well (changes underlined):

“Figure 4. Variation in the site of the kinetochore among two sets of human centromeres. a) Plot comparing the length of the kinetochore site, marked by hypomethylated DNA and CENP-A-containing chromatin, between the CHM1 and CHM13 centromeres. . . n.s., not significant, as determined with a two-sided Kolmogorov-Smirnov test.”

We have also included the details of this statistical test in a new section in the Methods, which now reads as follows:

“Estimation of the length of the kinetochore sites

To estimate the length of the CHM1 and CHM13 kinetochore sites, we first determined the CpG methylation status of each CHM1 and CHM13 centromere using the approach described above (see **CpG methylation analysis**). We, then, mapped the CENP-A ChIP-seq data from each genome to the same source genome using the mapping parameters described above (see **Native CENP-A ChIP-seq and analysis**). Next, we used CDR-Finder (<https://github.com/arozanski97/CDR-Finder>) to identify the location of hypomethylated regions within the centromeres, and we filtered the hypomethylated regions that had less than 10-fold enrichment of CENP-A ChIP-seq reads relative to the bulk nucleosomal reads. We reported the lengths of the hypomethylated regions enriched with CENP-A as determined with CDR-Finder, and we tested for statistical significance using a two-sided Kolmogorov-Smirnov test with GraphPad Prism (v9.5.1).”

The relative distance from HOR-to-monomeric transition zones to the position of sites of CENPA enrichment is analysed, which are shown to be in distinct locations between CHM1 and CHM13. This is

interesting and suggests migration or 'creep' of the kinetochore location over time – the authors may be interested in this paper <https://www.ncbi.nlm.nih.gov/pmc/articles/PMC5480163/> where maize CENH3 location is analysed in inbred lineages and related species and evidence for 'creeping' provided. In the case where the authors observe two CENPA peaks in their ChIP-seq data this is likely to represent two cell populations, as dicentrics are notoriously unstable (although the proximity of the peaks within the alpha satellite arrays may permit this I guess). Could this be looked at using FISH and immunostaining perhaps? To test whether two cell CENPA states are present in the sampled cell population?

We agree that it is possible that there may be a single population of cells with two kinetochores separated by 1-2 Mbp of sequence, or, conversely, two populations of cells with different kinetochore locations. To distinguish between these two, we first assess our CHM1 ultra-long ONT data to determine if there are any reads that span both hypomethylated regions (which mark the site of the kinetochore) in the chromosome 13 and 19 centromeres. However, because of the distance between the two hypomethylated regions (~1.7 and ~2.0 Mbp for chr13 and chr19, respectively), we were unable to identify any reads that spanned both regions.

We, therefore, reached out to our collaborators, Mario Ventura and Claudia Catacchio at the University of Bari, Italy, who are experts in performing immuno-FISH on metaphase chromosome spreads. They performed this procedure on stretched CHM1 metaphase chromosomes using a fluorescent DNA probe specific to either the chromosome 13/21 α -satellite (plasmid pZ21A; Archidiacono et al., *Genomics*, 1985) or chromosome 5/19 α -satellite (plasmid pGA16; Hulsebos et al., *Cytogenet Cell Genet.*, 1988) as well as an antibody specific to CENP-C (an inner-kinetochore protein). They assessed over 30 metaphase chromosome spreads for both chromosomes 13 and 19 (n=32 and 34, respectively) and found that the chromosome 13 centromere has a single CENP-C signal associated with the α -satellite DNA, while chromosome 19 has two CENP-C signals associated with the α -satellite DNA (see figure, below). This suggests that both hypotheses are possible: the chromosome 13 centromere has two populations of cells with a single kinetochore, while the chromosome 19 centromere has one population of cells with two kinetochores (forming a dicentric chromosome). We have added this finding to the main text and included this figure as **Extended Data Fig. 16**. The main text now reads as follows (changes underlined):

“In the case of chromosomes 13 and 19, the two distinct kinetochores are located more than 1 Mbp apart from each other (Fig. 4c,d). To test whether these two kinetochores represent two distinct cell populations or, alternatively, an early-stage somatic mutational event resulting in two kinetochores within the α -satellite HOR array, we performed immunostaining combined with fluorescent *in situ* hybridization (immuno-FISH) on stretched CHM1 metaphase chromosome spreads. We found that the chromosome 13 centromere has a single kinetochore, marked by the inner-kinetochore protein CENP-C, within the *D13Z2* α -satellite HOR array, while the chromosome 19 centromere has two kinetochores within the *D19Z3* α -satellite HOR array (Extended Data Fig. 16). Assessment of the underlying sequence and structure of the chromosome 13 *D13Z2* α -satellite HOR array reveals a 631 kbp deletion in approximately half of CHM1 cells (Extended Data Fig. 4, Supplementary Note 2), which may have contributed to the repositioning of the kinetochore in a subpopulation of cells, whereas the chromosome 19 centromere has no such deletion and may have had two kinetochores present from the first few cell divisions. Centromeres with two kinetochores (known as dicentrics) have been previously observed in humans and other species and have been shown to be viable, even with inter-kinetochore distances of up to 12 Mbp^{30,31}.”

Extended Data Figure 16. The CHM1 chromosome 13 centromere likely has one kinetochore site, while the CHM1 chromosome 19 centromere has two kinetochore sites. a-d) Immunofluorescence staining of stretched metaphase chromosome spreads from CHM1 cells with a fluorescent antibody against CENP-C (an inner-kinetochore protein; green) as well as a fluorescent chromosome 13/21 α -satellite DNA probe (**a,b**; red) or a fluorescent chromosome 5/19 α -satellite DNA probe (**c,d**; red). We find that there is a single CENP-C signal that coincides with the chromosome 13/21 α -satellite probe for each chromosome 13 sister chromatid, indicating that this chromosome likely has one kinetochore (**a,b**). Conversely, we find that there are two CENP-C signals that coincided with a single chromosome

5/19 α -satellite probe signal for each sister chromatid, indicating there is likely two kinetochores on this chromosome (**c,d**). $n=32$ and 34 metaphase chromosome spreads for chromosomes 13 and 19, respectively. Insets are magnified 1.7-fold (panels a and c) or 3.9-fold (panels b and d).

Line 738 – which species CENPA was this antibody raised against?

The CENP-A antibody was raised against human (specifically, aa 3-19 of the human CENP-A protein). We have included this in the sentence, and it now reads as such (changes underlined):

“To the remaining supernatant, 20 μ g mouse monoclonal anti-human CENP-A antibody (Enzo, ADI-KAM-CC006-E) was added and rotated overnight at 4°C.”

As the human centromeres are associated with LINEs, are there LINE or other transposons that are polymorphic between CHM1 and CHM13 either inside, or close to, the alpha satellite arrays. If there are such transposons, it would be good to indicate their positions in Fig 1 or Fig 2a. Equally, in Fig 2c, I would be interested to know how centromeric TE content varied as a function of this analysis across the chromosomes.

We performed this analysis and identified 92 polymorphic TEs between the CHM1 and CHM13 centromeric regions. All of the polymorphic TEs are located outside of the α -satellite HOR arrays, with most residing within unique sequences in the p- and q-arms. Interestingly, these polymorphic TEs are only found on 13/23 chromosomes (chromosomes 2, 6, 7, 8, 11, 13, 14, 15, 16, 17, 20, 21, and 22). We plotted these polymorphic TEs on **Fig. 1** as requested by the reviewer, coloring them by type (LINE, SINE, or LTR; panel a). Additionally, we plotted the number of polymorphic TEs in the CHM1 centromeric regions relative to CHM13 and vice versa (panels b,c). Finally, we calculated the lengths of the TE polymorphisms and plotted them in panel d. We have included this as new **Extended Data Fig. 24** (below), which we now cite in the legend of **Fig. 1**. Additionally, we provide a new section in the Methods on this analysis, which reads as follows:

“Polymorphic TE analysis

To detect polymorphic TEs between the CHM1 and CHM13 centromeric regions, we first ran RepeatMasker⁶² (v4.1.0) on the CHM1 and CHM13 centromeric regions. Then, we masked all satellite repeats within these regions using BEDtools⁶¹ maskfasta (v2.29.0). We aligned the masked CHM1 fasta to the masked CHM13 fasta using minimap2⁴⁷ and the following command: `minimap2 -t {threads} --eqx -c -x asm20 --secondary=no {ref.fasta} {query.fasta}`. Using the resulting PAF, we extracted the regions with structural variants that were >50 bp long. Then, we intersected these regions with the RepeatMasker annotation file to identify those variants that overlapped SINE, LINE, or LTR repeat classes by >75%. We considered the following LINE and SINE subgroups: LINE/CR1, LINE/L1, LINE/L1-Tx1, LINE/L2, LINE/Penelope, LINE/RTE-BovB, LINE/RTE-X, SINE/5S-Deu-L2, SINE/Alu, SINE/MIR, SINE/tRNA, SINE/tRNA-Deu, SINE/tRNA-RTE. We then determined the variation in length of these regions between the two centromeric regions, and we plotted their position and length using R⁶⁵ (v1.1.383) and the ggplot2 package⁶⁶.”

Extended Data Figure 24. Polymorphic TEs within the CHM1 and CHM13 centromeric regions. a) Map of the CHM1 and CHM13 centromeric regions, showing the location of 92 total LINEs (blue), SINEs (green), and LTRs (purple) relative to the α -satellite HOR array(s) and kinetochores(s). The TEs are shown as colorful lines next to the centromeric structures. **b,c)** Number of polymorphic LINE, SINE, and LTR insertions for the **b)** CHM1 centromeric regions and **c)** CHM13 centromeric regions. **d)** Length of the polymorphic TEs in the CHM1 and CHM13 centromeric regions.

The authors extend analysis to comparison of primate assemblies. How heterozygous were the primate centromeres - could the authors comment on the challenges of phasing any heterozygous centromere arrays? Why were only 6 chromosomes specifically focused on here?

The higher heterozygosity of the nonhuman primate genomes facilitated assembly and, in several cases, both haplotypes of a given chromosome were completely and accurately assembled and validated. Others, however, took time to methodically assess and resolve using other assembly methods. We selected these six centromeres because, in humans, they corresponded to distinct patterns and evolutionary trajectories. We are working on resolving all centromeres in multiple nonhuman primates; however, this is a particularly time-consuming process that will take at least another year to complete.

I think it is worth noting somewhere the time of divergence for the 4 species analysed. To what extent do the more divergent genomes show greatest divergence at the centromere level? Or are they all equally un-alike?

The estimated time of divergence of the nonhuman primate species is approximately 6 mya for chimpanzee, 12-16 mya for orangutan, and ~25 mya for macaque (as determined by molecular dating; Besenbacher et al., *Nat Ecol Evol*, 2019 and Glazko et al., *Mol Biol Evol*, 2003). In general, there is less sequence homology (i.e., greater sequence divergence) the larger the evolutionary difference; however, even for the most closely related species (human and chimpanzee), very little of the α -satellite HOR can be aligned, suggesting nearly complete evolutionary turnover since speciation. We are hopeful that sequencing more closely related species (e.g., bonobo and chimpanzee or Sumatran and Bornean orangutan) or even subspecies (e.g., Western and Central chimpanzee) will provide more insight into the evolutionary processes shaping the α -satellite HORs.

We have added a statement on the divergence times in the Methods, and it now reads as follows (changes underlined):

“Human and NHP phylogenetic analysis

Humans, chimpanzees, orangutans, and macaques diverged over a period of at least 25 million years, with chimpanzees diverging approximately 6 million years ago (mya)⁷⁶, orangutans 12-16 mya⁷⁶, and macaques ~25 mya⁷⁷. Despite these divergence times, all primates retain α -satellite repeats, which permit the phylogenetic analysis of these regions and an estimation of their evolutionary trajectory.”

Line 284 – I believe centromere dimeric satellite units have been reported in monkeys previously and could be cited here?

That is correct, and we now cite Pike et al., *J Mol Evol*, 1986 and Alkan et al., *PLOS Comp Biol*, 2007 in this sentence. It now reads as follows (changes underlined):

“Unlike apes, which possess complex HOR structures, macaque centromeric arrays are composed of dimeric α -satellite units^{16,34} that are 93-97% identical across all centromeres.”

Line 286 – ‘suprachromosomal family’ as a term should be more clearly defined on first usage. Line 288 – what is the significance of SF5? Is this a repeat family observed in humans? Are these alpha satellite families that predate the split of humans and chimps? When comparing SF families between species, can evidence of concerted evolution be detected? Am I correct in understanding that all alpha satellites, across the genomes analysed, share a common ancestry, but have since diverged following speciation, although some chromosomes have diverged less than others? The methods on line 770 onwards for

SF analysis do not clearly explain the approach to the uninitiated. Line 781 – what is ‘Ka’ and why does it indicate ‘SF7’, and similarly for the other designations in the next few lines?

We now define the term ‘suprachromosomal family’ in the main text and cite the original paper that coined the term (Alexandrov et al., *Chromosoma*, 1988; changes underlined):

“Assessment of the α -satellite suprachromosomal families (SFs), which are groups of α -satellite HORs that have a defined linear order of monomers and share sequence homology³⁵, among each primate centromere revealed four unexpected findings.”

Additionally, we now describe the different SF classes and their monomer designations in the Methods to clarify our analysis (changes underlined):

“Human and NHP α -satellite suprachromosomal family (SF) classification and strand orientation analysis

Human and NHP α -satellite monomers are grouped into 20 distinct SF classes based on shared sequence identity and structure, which is described in detail in Ref. 4. The SF classes and their monomers are as follows: SF1 (J1 and J2), SF01 (J3, J4, J5, and J6), SF2 (D2, D2, FD), SF02 (D3, D4, D5, D6, D7, D8, and D9), SF3 (W1, W2, W3, W4, and W5), SF4 (Ga), SF5 (R1 and R2), SF6 (Ha), SF7 (Ka), SF8 (Oa and Na), SF9 (Ca), SF10 (Ba), SF11 (Ja), SF12 (Aa), SF13 (Ia), SF14 (La), SF15 (Fa), SF16 (Ea), SF17 (Qa), SF18 (Pa and Ta). To determine the α -satellite SF content and strand orientation of human and NHP centromeres...”

With regard to the question about the SF5 repeats, SF5 is found in human centromeres, but it is never the dominant SF in human α -satellite HOR arrays. Instead, human α -satellite HOR arrays are typically composed of SFs 1-4 and 01. When we found that the chimpanzee chromosome 5 α -satellite HOR array was mainly composed of SF5 repeats, it was very surprising to us because it has never been found to be the dominant SF in any centromere among humans, chimpanzees, or gorillas. This suggests to us that human and chimpanzee lineages have evolved independently post-speciation and have had complete α -satellite turnover in some centromeres, such as chromosome 5, where the SF structure is completely altered between human and chimpanzee.

With regard to the question about α -satellite ancestry, all α -satellite DNA is thought to have a monophyletic origin that emerged after divergence of the prosimian lineages from the catarrhine and haplorhine ancestor. As a result, all simians have α -satellite DNA, and all prosimians and more distant primates have a different centromeric repeat.

Line 320 – please define the use of ‘monomeric’ more clearly – does this mean not in a HOR? This is very interesting, indicating that mutation rate is higher in areas of HOR formation? This might suggest a centromeric recombination process that is typified by relatively lower fidelity? The authors attempt to compare mutation rates between HORs and flanking regions and estimate higher rates. This is potentially very interesting, but could this be confounded by recombination processes? What is spectrum of transition and transversion classes and how does this relate to spectrums defined in other human cell types / contexts? To what extent are these estimates of mutation rate based on assumptions that orthologous ‘loci/repeats’ are compared and are such assumptions meaningful in the centromere? In this case the CHM1 and CHM13 cells appear to have some common ancestry, but it may be a long time since they have diverged, and they have been propagated as cell lines. I would like

to see a more explicit rationale for how mutation rate is being estimated, and potential confounding effects discussed, including the difficulty in identifying the ‘same’ repeat and the action of recombination simultaneously to mutation in the history of these two cell lines.

Monomeric α -satellite are single α -satellite monomers that are not organized into HORs. They typically exist in blocks and are usually located in the regions flanking the α -satellite HOR array. To make this clearer in the text, we have added the following definition (changes underlined):

“Because our analyses showed that the monomeric α -satellite sequences, or α -satellite monomers that are not organized into a HOR but typically flank the α -satellite HOR array, mutate less quickly and can be readily aligned among human and nonhuman apes...”

Our phylogenetic analyses would argue that standard homologous recombination processes are limited or suppressed, but intrachromosomal recombination or different repair mechanisms may, in fact, be contributing to the emergence of new α -satellite HORs. The monomeric α -satellite regions serve as effective anchor points because they typically are able to map uniquely to orthologous regions among the human and nonhuman primate species, with unique or particular segmental duplication blocks occurring distally. This is demonstrated in the StainedGlass plots below (pages 25-30), which reveal conservation of these regions (purple boxes/rectangles) among ape species despite nearly complete turnover of α -satellite HOR repeats. This conservation holds until the divergence of Asian and African great apes ~18 million years ago. Because the monomeric α -satellite regions are relatively modest in size (a few 100 kbp on average; **Extended Data Table 7**) and align uniquely between species, we assume that they are the most appropriate anchor points for the phylogenetic analysis.

We revisited the Ti/Tv rates as suggested and found that the Ti/Tv is approximately 1.69, based on the analysis of 9.2 Mbp of monomeric/diverged α -satellite sequence aligning 1:1 between the CHM1 and CHM13 genomes (see **Supplementary Table 1**, below). This is very similar to what we recently reported for segmental duplications (Vollger et al., *Nature*, 2023), which suggests an elevated mutation rate of 30-40% and potential for GC-biased gene conversion. Given this, we don’t believe the fact that CHM1 and CHM13 are passaged cell lines is of concern for this analysis. This is because the phylogenetic analyses and mutation rate estimates involved the use of nearly 60 different sequenced haplotypes from a variety of human genomes. If CHM1 and CHM13 were exceptional because of somatic mutation, we would expect branch length distortions in the ML trees, and this is simply not observed. As a control to these experiments, we also assessed 500 unique regions (corresponding to 64.95 Mbp) across the CHM1/CHM13 genomes and observed an expected normal density of SNVs and expected Ti/Tv ratio of 2.16 (see excerpt of **Supplementary Table 2**, below). We have included these tables in the **Supplementary Information** and have added the following details to the Methods:

“Pairs of human and NHP sequences were realigned using MAFFT^{72,73} (v7.453) and the following command: `mafft --maxiterate 1000 --localpair`. Next, we calculated the SNV density and Ti/Tv ratios from these alignments, limiting our analysis to only those regions with one-to-one unambiguous mapping and excluding segmental duplications and satellite repeats (Supplementary Table 1). As a control, we also calculated the SNV density and Ti/Tv ratios from 500 uniquely mapping regions across the genomes (**Supplementary Table 2**). We estimated the sequence divergence . . .”

Supplementary Table 1. SNP density and Ti/Tv ratios for 70 monomeric/diverged α -satellite regions across the CHM13 genome.

Coordinate	# of SNPs	# of kbp	SNP density (# of SNPs per	Ti/Tv
chr1:126701150-12682412	16	123.0	0.13	1.14
chr3:91711558-91738002	19	26.4	0.72	1.25
chr3:90339372-90804701	246	465.3	0.53	1.55
chr3:96415026-96498140	5	83.1	0.06	3.00
chr5:46752951-46830042	224	77.1	2.91	2.06
chr5:50026895-50121284	312	94.4	3.31	1.30
chr5:50455749-50532291	292	76.5	3.81	0.99
chr5:50597029-50664972	360	67.9	5.30	1.41
chr5:46161727-46684449	1,206	522.7	2.31	1.80
chr5:50968285-51082874	308	114.6	2.69	1.87
chr6:61058390-61194504	19	136.1	0.14	0.64
chr6:61195997-61413504	30	217.5	0.14	3.14
chr6:62446529-62665581	485	219.1	2.21	2.25
chr6:58243294-58286706	4	43.4	0.09	0.50
chr7:59398571-59610435	441	211.9	2.08	1.68
chr7:59816928-59834266	32	17.3	1.85	2.44
chr7:60044088-60414372	631	370.3	1.70	1.60
chr7:58123963-58276138	272	152.2	1.79	1.82
chr7:63714499-63765558	104	51.1	2.04	1.45
chr7:63766621-64188435	579	421.8	1.37	1.58
chr8:46325080-46389509	98	64.4	1.52	1.31
chr8:46423659-46466919	40	43.3	0.92	2.33
chr8:46473674-46475518	3	1.8	1.63	1.00
chr8:46531195-46551285	12	20.1	0.60	2.67
chr8:46669235-46703931	36	34.7	1.04	1.92
chr8:46722257-46751419	36	29.2	1.23	2.60
chr8:46769937-46779244	13	9.3	1.40	11.00
chr8:46827220-46920436	147	93.2	1.58	1.98
chr8:43846456-44215832	422	369.4	1.14	1.73
chr8:44217695-44243546	40	25.9	1.55	0.86
chr10:41926237-41933458	14	7.2	1.94	0.56
chr10:41935290-42061843	520	126.6	4.11	1.32
chr10:39068402-39633793	2,226	565.4	3.94	1.57
chr11:50542870-50584783	97	41.9	2.31	1.74
chr11:50600214-51023358	1,046	423.1	2.47	1.77
chr11:48819609-49047353	469	227.7	2.06	1.89
chr11:54681429-55197721	1,129	516.3	2.19	1.55
chr11:49077628-49118925	117	41.3	2.83	2.05
chr12:37202490-37369876	342	167.4	2.04	1.60
chr12:34156787-34163766	10	7.0	1.43	2.33
chr12:37548365-37997178	968	448.8	2.16	1.88
chr12:34284892-34473230	306	188.3	1.62	1.70
chr12:34485622-34593492	242	107.9	2.24	1.65
chr14:12708411-12710759	16	2.3	6.81	0.88
chr15:14702856-14750851	81	48.0	1.69	1.89
chr15:14954293-14969218	89	14.9	5.96	1.00
chr15:15902875-15996634	29	93.8	0.31	2.50
chr15:16668822-16678794	6	10.0	0.60	0.67
chr16:34517842-34542596	6	24.8	0.24	0.67
chr16:35677618-35696235	5	18.6	0.27	3.00
chr16:35794934-35834066	11	39.1	0.28	1.50
chr16:38028998-38213677	42	184.7	0.23	1.73
chr16:38342360-38373039	5	30.7	0.16	3.00
chr16:38417827-38432871	5	15.0	0.33	1.00
chr17:22893753-22926977	5	33.2	0.15	3.00
chr17:28100210-28139193	3	39.0	0.08	0.00
chr19:25740136-25800914	8	60.8	0.13	0.75
chr19:24352656-24570766	8	218.1	0.04	2.50
chr19:29769351-30152526	22	383.2	0.06	1.33
chr20:29896766-29928185	6	31.4	0.19	4.00
chr20:30660957-30663157	13	2.2	5.91	0.50
chr20:30814141-30883872	193	69.7	2.77	1.91
chr20:26343418-26383658	4	40.2	0.10	2.00
chr20:31650517-31685375	96	34.9	2.75	1.88
chr20:31694601-31725734	71	31.1	2.28	1.50
chr20:31905998-32017135	299	111.1	2.69	1.49
chr22:16514391-16570401	629	56.0	11.23	0.82
chr22:16844531-16875107	172	30.6	5.63	1.38
chrX:60927707-61200254	28	272.5	0.10	1.08
chrX:61201711-61247454	2	45.7	0.04	1.00
chrX:57593564-57819763	23	226.2	0.10	0.83
All	15,795	9,219.9	1.71	1.69
Mean	222	129.9	1.83	1.78

Supplementary Table 2. SNP density and Ti/Tv ratios for 500 unique regions across the CHM13 genome.

Coordinate	# of SNPs	# of kbp	SNP density (# of SNPs per	Ti/Tv
chr1:10006768-10136668	122	129.9	0.94	2.90
chr1:102605319-10273521	10	129.9	0.08	2.33
chr1:109071578-10920147	20	129.9	0.15	2.80
chr1:116908110-11703801	129	129.9	0.99	2.23
chr1:160677290-16080719	164	129.9	1.26	1.04
chr1:161723356-16185325	109	129.9	0.84	1.92
chr1:161903678-16203357	2	129.9	0.02	0.00
chr1:171102778-17123267	265	129.9	2.04	2.14
chr1:173187309-17331720	146	129.9	1.12	1.92
chr1:185495893-18562579	28	129.9	0.22	3.00
chr1:186207008-18633690	179	129.9	1.38	1.74
chr1:187466528-18759642	165	129.9	1.27	1.49
chr1:19223074-19352974	62	129.9	0.48	2.26
chr1:196289803-19641970	131	129.9	1.01	2.85
chr1:197177054-19730695	130	129.9	1.00	1.83
chr1:198997286-19912718	8	129.9	0.06	1.67
chr1:20088656-20218556	83	129.9	0.64	1.73
chr1:202813419-20294331	16	129.9	0.12	2.20
chr1:208089096-20821899	92	129.9	0.71	2.68
chr1:217969126-21809902	183	129.9	1.41	1.73
chr1:222761884-22289178	227	129.9	1.75	1.94
chr1:227352892-22748279	169	129.9	1.30	2.07
chr1:230400095-23052999	167	129.9	1.29	2.86
chr1:231110400-23124030	268	129.9	2.06	3.68
chr1:24570071-24699971	39	129.9	0.30	2.00
chr1:28444982-28574882	101	129.9	0.78	1.81
chr1:3077707-3207607	83	129.9	0.64	2.42
chr1:35679346-35809246	38	129.9	0.29	1.64
chr1:41128043-41257943	44	129.9	0.34	2.39
chr1:42867829-42997729	115	129.9	0.89	2.35
chr1:49199815-49329715	61	129.9	0.47	2.00
...				
chr22:29390893-29520793	106	129.9	0.82	1.92
chr22:35986699-36116599	118	129.9	0.91	2.90
chr22:36353831-36483731	9	129.9	0.07	8.00
chr22:36551850-36681750	125	129.9	0.96	1.55
chr22:50710389-50840289	100	129.9	0.77	3.71
chrX:106572411-10670231	25	129.9	0.19	2.00
chrX:106896415-10702631	2	129.9	0.02	1.00
chrX:110031436-11016133	13	129.9	0.10	2.00
chrX:115541161-11567106	238	129.9	1.83	2.00
chrX:123315013-12344491	135	129.9	1.04	1.81
chrX:124070175-12420007	166	129.9	1.28	2.17
chrX:12967390-13097290	136	129.9	1.05	2.40
chrX:131989225-13211912	7	129.9	0.05	1.33
chrX:139033877-13916377	151	129.9	1.16	1.89
chrX:17199315-17329215	5	129.9	0.04	1.50
chrX:22758829-22888729	135	129.9	1.04	1.79
chrX:25232421-25362321	121	129.9	0.93	1.79
chrX:31370480-31500380	108	129.9	0.83	2.34
chrX:4038267-4168167	62	129.9	0.48	1.77
chrX:41459852-41589752	77	129.9	0.59	2.67
chrX:43478412-43608312	30	129.9	0.23	1.14
chrX:46684015-46813915	78	129.9	0.60	3.33
chrX:49618620-49748520	76	129.9	0.59	1.89
chrX:53701630-53831530	8	129.9	0.06	2.50
chrX:66991715-67121615	41	129.9	0.32	3.00
chrX:68876115-69006015	79	129.9	0.61	1.36
chrX:74491118-74621018	158	129.9	1.22	1.45
chrX:80157295-80287195	26	129.9	0.20	0.92
chrX:83434701-83564601	110	129.9	0.85	2.55
All	59,302	65,079.9	0.91	2.16
Mean	118	129.9	0.91	2.16

Lines 342-344. An assumption of limited or no recombination across the satellite arrays is reasonable if the authors are referring to the documented absence of meiotic crossovers based on segregation of flanking markers (i.e. cenhaps). However, as the authors show, the alpha satellite arrays are polymorphic between CHM1 and CHM13, and more widely - this indicates most likely some form of DSB formation and homologous repair, including non-allelic forms, that are capable of generating these changes. Otherwise, how do the authors propose that the array polymorphisms arise? The patterns of internal centromere satellite polymorphism, embedded in cenhaps, was also recently noted in the plant *Arabidopsis thaliana* (Wlodzimierz et al 2023 Nature), suggesting these modes of centromere evolution are conserved as far as plants and humans, which I think would be worth adding to the discussion.

Yes, DSB followed by homologous repair is one potential mechanism that could result in variation within and between α -satellite HOR arrays, and this is particularly attractive because there is corresponding loss of old α -satellite HORs as the new α -satellite HORs expand (**Fig. 7b,c**). The recent study of *Arabidopsis* does indeed note large blocks of linkage disequilibrium coupled with rapid turnover, likely as a result of unidirectional gene conversion or unequal crossover between sister chromatids coupled with transposon purging. While we see little evidence of the latter, the interdigitated spread of new α -satellite HOR (i.e., not continuous) favors a unidirectional gene conversion. We now include the Wlodzimierz citation in the Discussion:

“These changes in DNA occur most frequently in concert with gains and losses of α -satellite HOR units and do not appear to do so in a contiguous manner but, instead, are intermixed with ancestral HORs. The mechanism responsible for these changes is currently not well described, but it is hypothesized that they occur in a saltatory fashion as opposed to a constant rate of mutation, potentially as a result of meiotic drive for the newly minted HORs. Mechanisms involving DNA double-strand break formation followed by homologous or unidirectional gene conversion between sister chromatids, as has been recently suggested for centromeric DNA in *A. thaliana* (Wlodzimierz et al., *Nature*, 2023), may account for this pattern.”

A more explicit model of recombination may also be valuable to consider in relation to mutation estimates, as the observed polymorphisms will be influenced by both mutation and recombination rates. Indeed, Profs Alexandrov and Miga have previously published a kinetochore associated recombination machine (KARM) relating to this, which it might be illuminating to discuss in light of CHM1 vs CHM13. Although I note that in Fig3c, the new HORs are more widely distributed than the kinetochore site, although, perhaps they are being ‘pushed out’ from the CENPA centre where KARM would putatively be located? Or the kinetochore used to be in different locations to that observed here?

While we don’t yet have complete sequences of the majority of NHP centromeres nor have we sufficiently sampled the extent of human diversity, including those with recently emerged α -satellite HORs, we don’t find sufficient evidence for the kinetochore-associated recombination machine (KARM) model. As we noted before, some of the most identical α -satellite HORs (suggesting the most recent homogenization) do not correspond to either the CDR or CENP-A chromatin enrichment observed in humans (Logsdon et al., *Nature*, 2021). Nevertheless, we’ve added this point to the Discussion and included the Shepelev and Alexandrov & Miga citations as follows (changes underlined):

“The chromosome 5 centromere, for example, mutates at least 10-fold faster than the chromosome X centromere, with the net effect that almost 48% of the α -satellite HORs cannot be aligned to either CHM1 or CHM13 references (**Fig. 6**). This rapid evolution has led to the emergence of new, human-

specific α -satellite HORs that are unique to a subset of haplotypes. Interestingly, we find little evidence that there is a one-to-one correspondence between the sites of kinetochore attachment and areas of rapid evolutionary turnover and homogenization as predicted by kinetochore-associated recombination machine (KARM) model (Shepelev et al., *PLoS Genet*, 2009; Miga & Alexandrov, *Annu Rev Genet*, 2010). As more primate centromeres are sequenced and assembled, we expect that we will discover many more novel α -satellites that have evolved separately from the other great apes, which will provide further insight into their variation and evolution.”

Figure 2a – It would be interesting to also see StainedGlass plots for these comparisons.

The StainedGlass plots for these centromeres are included in **Extended Data Fig. 14**, along with the plots for all of the other CHM1 and CHM13 centromeres. You can see the sequence similarity between the chromosome 19 α -satellite HOR arrays as well as the divergence between the chromosome 5 α -satellite HOR arrays from CHM1 and CHM13 (featured in **Fig. 2a**).

Line 386 – how is ‘center’ defined here? Is it the physical centre of the alpha-satellite array? Is it in relation to CENPA?

The “center” of the α -satellite array is the midpoint of the youngest evolutionary layer. It typically has the highest sequence identity within the centromere and is flanked by older, more divergent α -satellites. It is typically associated with CENP-A chromatin, but not always. For example, in **Extended Data Fig. 14**, you can see that the CENP-A-enriched region is offset from the “center” of the array in CHM13 chromosome 8, yet it is positioned within it in the CHM1 counterpart.

Could the authors comment on HSat array variation between CHM1 and CHM13? Are the Hsat arrays evident in the pangenome assemblies?

The HSat arrays are particularly challenging to assemble, as they are often 50 kbp–5 Mbp long and highly identical in sequence (>99%). For this reason, we did not focus on assembling these regions and assessing their variation. However, in the process of assembling the α -satellite HOR arrays for both chromosomes 3 and 4, we did assemble the HSat arrays that reside between and adjacent to them. Comparison of the lengths of the HSat arrays reveals that they vary in size by up to 5.7-fold. Below, we show the structure, size, and sequence identity for the chromosome 3 and 4 HSat arrays in the CHM1 and CHM13 genomes. Interestingly, we find that the CHM13 chromosome 3 HSat1A array is interrupted by a short, 33.6 kbp α -satellite HOR array that does not exist in the CHM1 genome (panel b). This highlights the variation in structure that we see even across two genomes, and there is certainly even greater variation present in the pangenome assemblies, which will be assessed in a future study.

Reviewer Figure 1. Variation in HSat1A and HSat3 array lengths and structure within the chromosome 3 and 4 centromeric regions. a-d) Complete sequence assembly of the HSat1A and HSat3 arrays residing between and adjacent to the α -satellite HOR arrays on a) CHM1 chromosome 3, b) CHM13 chromosome 3, c) CHM1 chromosome 4, and d) CHM13 chromosome 4 reveals up to 5.7-fold variation in length.

In Figure 1 the repeats are coloured by n-mer composition. Are all say 9-mers the same, or is it possible to have independent 9-mers on different chromosomes?

It is possible to have different n-mer HOR compositions for a given n, and we highlighted this in **Fig. 3b**, where the CHM1 chromosome 5 *D5Z2* α -satellite HOR array has a novel 4- and 6-mer HOR that is structurally different from the other 4- and 6-mer HORs in the array. We were also able to define the composition of these α -satellite HOR variants, and we illustrated them in **Extended Data Fig. 13a**. We provide these figures below for convenience.

Fig. 3b:

Fig. 3b. Discovery of novel α -satellite HOR variants within the CHM1 chromosome 5 centromere that are missing from the CHM13 chromosome 5 centromere. The novel α -satellite HOR variants (indicated with an asterisk) are 4- and 6- α -satellite monomers in length and have a different structure from the other 4- and 6-mer α -satellite HOR variants present in both centromeres.

Extended Data Fig. 13a:

Fig. 13a. Structure of the α -satellite HOR variants present in the CHM1 chromosome 5 centromere. HORs with the same number of α -satellite monomers but different structures are indicated with a superscript letter. There are three different structures for a 10-mer HOR, six different structures for a 6-mer HOR, four different structures for a 4-mer HOR, and two difference structures for a dimer HOR.

Line 412. This is an interesting discussion on different evolutionary rates between different centromeres. Can differential selection be ruled out, instead or, or in addition to, varying mutation and/or recombination rates?

This is a good point, and selection is indeed possible and cannot be ruled out at this point. We have added a note to this effect as follows (new sentence underlined):

“This remarkable plasticity in kinetochore position despite the conserved, essential function of these regions underscores the “centromere paradox”³⁷, an unresolved conundrum regarding the contradictory phenomenon of rapidly evolving centromeric DNA and proteins despite their essential role in ensuring faithful chromosome transmission. The germline and somatic stability of both the kinetochore location and the underlying DNA sequence will need to be investigated by examining genetic and epigenetic variation in centromeres across multiple generations. In addition, because both CHM1 and CHM13 represent cell cultures subject to somatic changes during passaging, it will be important to assess variation in multiple primary tissues from the same donor. Both differential selection and accelerated mutation may be contributing to the centromere paradox.”

Lines 422 – how karyotypically different are these genomes? Do they have the same number of chromosomes? What has happened to centromeres when they have been gain or lost since speciation?

Karyotypically, humans and chimpanzees are very similar. Chimpanzees have one more chromosome ($2n=48$) due to the presence of two chromosomes (2a and 2b) that fused in the ancestral human lineage to form chromosome 2. In humans, the chromosome 2q centromere became inactivated, and almost all α -satellite was subsequently lost in this vestigial centromere, other than a remnant of ~40-50 kbp. There are nine additional pericentric inversions that distinguish human and chimpanzee—seven of which occurred on the chimpanzee lineage (i.e., human organization is ancestral). Most Giemsa bands are virtually identical between the lineages.

Line 432 – what do the authors hypothesize causes the higher apparent mutation rate in centromere 5?

We do not have a good explanation at present as to why there is a higher rate of mutation and evolutionary turnover for the chromosome 5 centromere. It may be, however, an intrinsic property of the suprachromosomal subfamily itself. Haaf and Willard, *Chromosoma*, 1997, made the observation that concerted evolution had led to largely non-orthologous centromeres between human and chimpanzee. Similarly, Archidiacono et al., *Genomics*, 1995, reported that suprachromosomal I and II α -satellite subfamilies seem to show the greatest evolutionary turnover based on FISH data. Our sequence and assembly results confirm these earlier cytogenetic observations.

In Figure 5 – it would be interesting to also have some stainedglass analysis comparing across species, eg Chr5 compared in the same stainedglass plot for all 5 assemblies?

We generated new StainedGlass plots that show the sequence identity across each of the six centromeres for all six primate genomes (CHM1, CHM13, human (HG00733), chimpanzee, orangutan, and macaque; see below). For all centromeres, the monomeric α -satellite sequences are 70-80% identical to each other across all primates, whereas the α -satellite HORs have completely turned over.

These plots also show the relative scale of the different centromeric regions across primate genomes. We have included these plots as **Supplementary Figs. 65-70** and refer to them in the legend of **Fig. 5**.

Chromosome 5 centromeres:

Supplementary Figure 65. Sequence identity map of the chromosome 5 centromeres from six human and NHPs. A sequence identity map of the chromosome 5 centromeres from CHM1, CHM13, human (HG00733), chimpanzee, orangutan, and macaque genomes (generated via StainedGlass⁵⁹) reveal 70-90% sequence identity among monomeric/diverged α -satellite flanking the α -satellite HOR array.

Chromosome 10 centromeres:

Supplementary Figure 66. Sequence identity map of the chromosome 10 centromeres from six human and NHPs. A sequence identity map of the chromosome 10 centromeres from CHM1, CHM13, human (HG00733), chimpanzee, orangutan, and macaque genomes (generated via StainedGlass⁵⁹) reveal 70-90% sequence identity among monomeric/diverged α -satellite flanking the α -satellite HOR array.

Chromosome 12 centromeres:

Supplementary Figure 67. Sequence identity map of the chromosome 12 centromeres from six human and NHPs. A sequence identity map of the chromosome 12 centromeres from CHM1, CHM13, human (HG00733), chimpanzee, orangutan, and macaque genomes (generated via StainedGlass⁵⁹) reveal 70-90% sequence identity among monomeric/diverged α -satellite flanking the α -satellite HOR array.

Chromosome 20 centromeres:

Supplementary Figure 68. Sequence identity map of the chromosome 20 centromeres from six human and NHPs. A sequence identity map of the chromosome 20 centromeres from CHM1, CHM13, human (HG00733), chimpanzee, orangutan, and macaque genomes (generated via StainedGlass⁵⁹) reveal 70-90% sequence identity among monomeric/diverged α -satellite flanking the α -satellite HOR array as well as some α -satellite HORs within the array.

Chromosome 21 centromeres:

Supplementary Figure 69. Sequence identity map of the chromosome 21 centromeres from six human and NHPs. A sequence identity map of the chromosome 21 centromeres from CHM1, CHM13, human (HG00733), chimpanzee, orangutan, and macaque genomes (generated via StainedGlass⁵⁹) reveal 70-90% sequence identity among monomeric/diverged α -satellite flanking the α -satellite HOR array.

Chromosome X centromeres:

Supplementary Figure 70. Sequence identity map of the chromosome X centromeres from six human and NHPs. A sequence identity map of the chromosome X centromeres from CHM1, CHM13, human (HG00733), chimpanzee, orangutan, and macaque genomes (generated via StainedGlass⁵⁹) reveal 70-90% sequence identity among monomeric/diverged α-satellite flanking the α-satellite HOR array.

Lines 72-73 – please provide reference(s) for this statement. Similarly at the end of sentence ending line 76.

We have added citations to the end of those sentences, and they now read as follows (changes underlined):

“Human centromeres have been shown to represent some of the most diverse and rapidly evolving regions in the genome (Archidiacono et al., *Genomics*, 1995; Cechova et al., *MBE*, 2019).”

“The bulk of human centromeric DNA is composed of tandemly repeating, ~171 bp α -satellite DNA, which are organized into higher-order repeat (HOR) units that can extend for megabase pairs (Mbp) of sequence and are particularly variable among humans due to the action of unequal crossing over, concerted evolution, and saltatory amplification (Miga and Alexandrov, *Annu Rev Genet*, 2021, and Logsdon and Eichler, *Genes*, 2022).”

Lines 144-145 – how are ‘euchromatin’ and ‘heterochromatin’ being defined in this statement – a more precise definition would be beneficial.

We refer to the original cytogenetic definition of “euchromatin” as loosely packed chromatin and “heterochromatin” as densely packed chromatin, typically stained with quinacrine and other dyes.

Referee #2 (Remarks to the Author):

The study is primarily interesting because they fully assembled centromeres of a second human genome and compared them with CHM13 and other primates, as well as insights into the genetic variation and evolutionary trajectories of human centromeres. Extending previous findings reported by Altemose et al. 2022, they show that centromeric alpha-satellite arrays are highly dynamic in humans by comparing with additional 56 human genomes. The mutation rate analysis of centromeres and its use as a proxy to estimate the separation times of human centromeric haplotypes is very interesting and a pioneer, despite being likely underestimated. In summary, the study significantly contributes to the understanding of centromere biology by providing comprehensive and accurate centromere sequences, characterizing genetic variation among human centromeres, and shedding light on the evolutionary dynamics of centromeres across primate species. The results underscore the complexity and heterogeneity of centromeres, emphasizing the need for further research to elucidate their functional significance and contribution to genome stability. The figures are well-prepared and self-explaining.

The analysis of genetic variation among human centromeres revealed significant heterogeneity in sequence identity and structure. The comparison between CHM1 and CHM13 centromeres demonstrated that sequence identity increases from heterochromatin to euchromatin regions. The α -satellite higher-order repeat (HOR) arrays, which are characteristic of centromeres, exhibited varying degrees of sequence identity between the two haplotypes. Furthermore, when comparing centromeres from diverse human genomes, the researchers found considerable variation in α -satellite HORs, indicating the emergence of new HOR structures in some haplotypes but not others. This highlights the extensive single-nucleotide and structural diversity of human centromeres.

The length and organization of α -satellite HOR arrays were found to vary among centromeres. CHM1 arrays were, on average, larger than their CHM13 counterparts, and specific chromosomes showed significant differences in size and structure. The analysis also revealed the presence of evolutionary layers within the α -satellite HOR arrays, indicating complex dynamics of HOR expansions and contractions. The comparison with 56 incompletely assembled reference genomes further confirmed the chromosome-specific nature of centromere variations and highlighted the unique patterns of α -satellite HOR organization and size for different chromosomes.

The study also investigated epigenetic differences between CHM1 and CHM13 centromeres, particularly focusing on the kinetochore regions. The kinetochore is a proteinaceous complex critical for chromosome segregation, and it resides within the hypomethylated centromere dip region (CDR) in humans. The researchers observed differences in the size and position of kinetochore sites between the two haplotypes, with several sites located hundreds of kilobases apart. The examination of underlying sequences revealed both conserved and divergent regions associated with kinetochores, suggesting complex evolutionary dynamics.

To gain insights into the evolutionary history of centromeres across primate lineages, the researchers sequenced and assembled orthologous centromeres from four primate species. The analysis of centromeres in chimpanzee, orangutan, and macaque genomes allowed for the reconstruction of evolutionary trajectories over millions of years. The comparison of centromeres revealed species-specific differences in α -satellite HOR organization and highlighted the presence of evolutionary layers

within the arrays. The findings suggest different mutation rates and diverse evolutionary trajectories among primate centromeres.

The methods were performed adequately as mentioned for:

1. Well-described pipeline for complete assembly of centromeres and subsequent analysis
2. ChIP-seq properly performed with two replicates

Below are my comments:

Major concerns:

The potential “discovery” of two kinetochores on chromosome 13 and chromosome 19 centromeres in the CHM1 genomes is due to the presence of two hypomethylated regions enriched with CENP-A chromatin.

The authors discuss this in terms of “likely represents two populations of cells, which may have arisen due to a somatic mutation, resulting in differing epigenetic landscapes”. However, it cannot be excluded they actually are present in the same cell, as it has been shown before that centromeric sites that are not too far apart from each other can still stably function as a monocentromere

(https://www.nature.com/articles/ng1198_227; <https://link.springer.com/article/10.1007/s10577-012-9302-3>).

This comment is similar to one raised by Reviewer 1, and to address this, we performed immuno-FISH on metaphase chromosome spreads from CHM1 cells to determine if the two hypomethylated regions enriched with CENP-A on the chromosome 13 and 19 centromeres represent a single population of cells with two kinetochores or two populations of cells with different kinetochore locations. To do this, we initiated a collaboration with Mario Ventura and Claudia Catacchio at the University of Bari, Italy, who are experts in performing immuno-FISH on metaphase chromosome spreads. They used a fluorescent DNA probe specific to either the chromosome 13/21 α -satellite or the chromosome 5/19 α -satellite as well as an antibody specific to CENP-C (an inner-kinetochore protein) on stretched metaphase chromosome spreads and found that the chromosome 13 centromere has a single CENP-C signal associated with the α -satellite DNA, while chromosome 19 has two CENP-C signals associated with the α -satellite DNA (see **Extended Data Fig. 16**, below). This suggests that both hypotheses are possible: the chromosome 13 centromere has two populations of cells with a single kinetochore, while the chromosome 19 centromere has one population of cells with two kinetochores. We added this finding to the main text and cited the publications above (changes underlined):

“In the case of chromosomes 13 and 19, the two distinct kinetochores are located more than 1 Mbp apart from each other (**Fig. 4c,d**). To test whether these two kinetochores represent two distinct cell populations or, alternatively, an early-stage somatic mutational event resulting in two kinetochores within the α -satellite HOR array, we performed immunostaining combined with fluorescent in situ hybridization (immuno-FISH) on stretched CHM1 metaphase chromosome spreads. We found that the chromosome 13 centromere has a single kinetochore, marked by the inner-kinetochore protein CENP-C, within the *D13Z2* α -satellite HOR array, while the chromosome 19 centromere has two kinetochores within the *D19Z3* α -satellite HOR array (**Extended Data Fig. 16**). Assessment of the underlying sequence and structure of the chromosome 13 *D13Z2* α -satellite HOR array reveals a 631 kbp deletion in approximately half of CHM1 cells (**Extended Data Fig. 4, Supplementary Note 2**), which may have contributed to the repositioning of the kinetochore in a subpopulation of cells, whereas the chromosome

19 centromere has no such deletion and may have had two kinetochores present from the first few cell divisions. Centromeres with two kinetochores (known as dicentrics) have been previously observed in humans and other species and have been shown to be viable, even with inter-kinetochore distances of up to 12 Mbp (Stimpson et al., *Chromosome Research*, 2012; Sullivan and Willard, *Nature Genetics*, 1998).”

Extended Data Figure 16. The CHM1 chromosome 13 centromere likely has one kinetochore site, while the CHM1 chromosome 19 centromere has two kinetochore sites. a-d) Immuno-FISH

staining of stretched metaphase chromosome spreads from CHM1 cells with a fluorescent antibody against CENP-C (an inner-kinetochore protein; green) as well as a fluorescent chromosome 13/21 α -satellite DNA probe (a,b; red) or a fluorescent chromosome 5/19 α -satellite DNA probe (c,d; red). We find that there is a single CENP-C signal that coincides with the chromosome 13/21 α -satellite probe for each chromosome 13 sister chromatid, indicating that this chromosome likely has one kinetochore (a,b). Conversely, we find that there are two CENP-C signals that coincided with a single chromosome 5/19 α -satellite probe signal for each sister chromatid, indicating there is likely two kinetochores on this chromosome (c,d). n=32 and 34 metaphase chromosome spreads for chromosomes 13 and 19, respectively. Insets are magnified 1.7-fold (panels a and c) or 3.9-fold (panels b and d).

On this topic, I noticed that Altemose et al. (2022) have used a so-called “Marker-assisted mapping strategy” that can localize CENP-A enrichment specifically to one of two large macro-repeat structures with recent HOR expansions. I wonder if the different mapping strategy used in the present study is responsible for the two centromere peaks found in Chr13 and chr19 as they were not found in Altemose et al., but rather on Chr4, which in contrast not found here...

We tested if using the marker-assisted mapping strategy described in Altemose et al. affects the CENP-A ChIP-seq enrichment patterns observed for the CHM1 chromosome 13 and 19 centromeres by performing the same procedure for both replicates of CENP-A ChIP-seq and filtering alignments by unique 51-mers. However, we still observed two CENP-A ChIP-seq peaks that coincide with hypomethylation on both chromosomes 13 and 19, confirming our initial findings and indicating that these peaks are present even when a different mapping strategy is applied (see below). These results are also consistent with our immuno-FISH experiments (described above), and we have now included this as **Extended Data Fig. 15**.

Extended Data Figure 15. CHM1 chromosome 13 and 19 centromeres have two regions enriched with CENP-A chromatin within hypomethylated α -satellite DNA. a,b) Two strategies for mapping CHM1 CENP-A ChIP-seq data (Methods) reveal similar patterns of CENP-A chromatin enrichment,

with two regions enriched with CENP-A that coincide with hypomethylated α -satellite DNA within the CHM1 **a)** chromosome 13 and **b)** chromosome 19 α -satellite HOR arrays.

Page 5 Lines 218-220: I wonder if "... Although CHM1 centromeric α -satellite HOR arrays are typically larger, the majority of CHM1 kinetochore sites (18 out of 23) are smaller than their CHM13 counterparts, with an average size of 178 versus 214 kbp...", it is or partially due to the differential mapping strategies applied.

We used the same CENP-A ChIP-seq mapping procedure for both CHM1 and CHM13 genomes before we calculated the lengths of the kinetochore sites. We did this to remove any biases and ensure that both sets of kinetochore lengths were calculated with the same approach. Therefore, the reported lengths are not due to differential mapping strategies, as they were determined with the same mapping strategy. We have now added a section on this in the Methods to clarify our approach:

“Estimation of the length of the kinetochore sites

To estimate the length of the CHM1 and CHM13 kinetochore sites, we first determined the CpG methylation status of each CHM1 and CHM13 centromere using the approach described above (see **CpG methylation analysis**). We, then, mapped the CENP-A ChIP-seq data from each genome to the same source genome using the mapping parameters described above (see **Native CENP-A ChIP-seq and analysis**). Next, we used CDR-Finder (<https://github.com/arozanski97/CDR-Finder>) to identify the location of hypomethylated regions within the centromeres, and we filtered the hypomethylated regions that had less than 10-fold enrichment of CENP-A ChIP-seq reads relative to the bulk nucleosomal reads. We reported the lengths of the hypomethylated regions enriched with CENP-A as determined with CDR-Finder, and we tested for statistical significance using a two-sided Kolmogorov-Smirnov test with GraphPad Prism (v9.5.1).”

Importantly, because this approach estimates the length of the kinetochore sites based on the CpG methylation profile and then filters regions that are not significantly enriched with CENP-A, the same sizes would be estimated with the marker-assisted mapping strategy, as these hypomethylated regions are still enriched with CENP-A.

The authors refer to their ChIPseq mapping as follows: “The resulting SAM files were filtered using SAMtools (v1.9) 764 with flag score 2308 to prevent multi-mapping of reads. With this filter, reads mapping to more than one location are randomly assigned a single mapping location, thereby preventing mapping biases in highly identical regions.”

This mapping strategy can be commonly applied to mapping to centromeric repeats assuming a more homogenous centromere organization. However, in such cases of two centromere peaks appearing the author should perform more strict mapping strategies.

In addition to filtering reads with FLAG score 2308, we also required a minimum MAPQ score of 1, which greatly reduces the likelihood that a read is mismapped. We find that including this minimum MAPQ requirement filters many mismapped reads and gives confident alignments, similar to those generated with the marker-assisted mapping strategy, as shown above. We clarify this in the following sentence in that paragraph (changes underlined):

“Alignments were normalized and filtered with deepTools⁶² (v3.4.3) bamCompare with the following parameters: `bamCompare -b1 {ChIP.bam} -b2 {bulk_nucleosomal.bam} --operation ratio --binSize 1000 --minMappingQuality 1 -o {out.bw}.`”

Minor concerns:

The authors mention an average increase of 1.3-fold in HOR of CHM1 compared to CHM13, but I miss how many CHM13 were in fact larger than in CHM1.

We find that 9 out of 23 α -satellite HOR arrays are larger in CHM13 than in CHM1, while 16 out of 23 arrays are larger in CHM1 than CHM13 (**Figs. 2c,3a** and **Extended Data Table 7**). This sums to 25 α -satellite HOR arrays for 23 chromosomes because chromosomes 3 and 4 have two kinetochore-forming α -satellite HOR arrays, and one from each chromosome is larger in CHM1 than in CHM13 (and vice versa). The exact sizes of the α -satellite HOR arrays are reported in **Extended Data Table 7**.

Altemose et al. (2022) have detected smaller regions of CENP-A enrichment outside of the primary CDR, with some overlapping a minor, secondary CDR (chr 4, chr16, and chr22) or no CDR at all (chr18). It would be nice to have a comparison of these chromosomes here.

We provided a map of the CpG methylation profile and CENP-A chromatin enrichment for all CHM1 and CHM13 centromeres in **Extended Data Fig. 14**, and we provide a zoom-in of these four sets of centromeres (from chromosomes 4, 16, 18, and 22) below for convenience. While we do not see any secondary enrichments of CENP-A chromatin on the chromosome 4, 18, and 22 centromeres in CHM1 (panels a, e, and g), we do see it on the chromosome 16 centromere (panel c). This enrichment also coincides with a dip in CpG methylation frequency and occurs in the transition region between diverged α -satellite HORs and the α -satellite HOR array. However, we note that this enrichment is in a different location relative to the main CENP-A chromatin site in CHM1 vs. CHM13 (it is p-arm-proximal in CHM1, whereas it is q-arm-proximal in CHM13). Each of these plots are now included as separate supplemental figures as requested by Reviewer 3, and they are now **Supplemental Figs. 43, 55, 57, and 62**. We note the secondary enrichment site in the figure legend for CHM1 chromosome 16 (**Supplemental Fig. 55**).

Reviewer Figure 2. Secondary sites of CENP-A chromatin enrichment present in the CHM13 centromeres are not observed in the CHM1 centromeres, except for on chromosome 16. a-h) Comparison of the structure, CpG methylation status, CENP-A chromatin enrichment pattern, and sequence identity for four centromeres (chromosomes 4, 16, 18, and 22) that have secondary sites of CENP-A enrichment in the CHM13 genome (Altemose et al., *Science*, 2022). We find that only the chromosome 16 centromere has a secondary site of CENP-A enrichment in the CHM1 genome, and it is located in the transition region between divergent α -satellite HORs and the D16Z2 HOR array. This location is different from the location of CENP-A enrichment observed in the CHM13 chromosome 16, which resides on highly identical α -satellite HORs.

Page 6 Line 258 – Please correct Mbp for Gbp

Thank you for catching this and bringing it to our attention. We have now made this change.

Figure 5 bottom legend – Dark green dots for Orangutan should be H2 and not H1, right?

Yes, thanks again. We have fixed this in the figure.

Limitations of the study that could be better addressed:

I am not really aware of possible limitations on that, but despite all the efforts in sequencing and assembly of the CHM1 centromeres, it would have been better to have chosen a natural human sample than rather a cell line with somatic rearrangements. Despite all the authors' efforts in checking the integrity of the CHM1 cell lines, the somatic rearrangements found can certainly have an influence on the results obtained.

We agree. While both CHM1 and CHM13 had the benefit of being haploid and, therefore, facilitating complete centromere characterization, both sources have been subject to multiple passages and do not represent true diploid primary material. We have made a note of this limitation in the Discussion (changes underlined):

“The germline and somatic stability of both the kinetochore location and the underlying DNA sequence will need to be investigated by examining genetic and epigenetic variation in centromeres across multiple generations. In addition, because both CHM1 and CHM13 represent cell cultures subject to somatic changes during passaging, it will be important to assess variation in multiple primary tissues from the same donor.”

ChIPseq for the other primates' centromeres. The study would have been more complete if they had carried out ChIPseq for the primates as well. This would give a clear picture, at a deeper resolution, about the evolution of centromeres in the group.

André Marques

While it is a significant undertaking to perform ChIP-seq experiments and examine all the centromeres and haplotypes for every primate under study here, we did select one ape for more detailed investigation. Specifically, we performed CENP-A ChIP-seq on the chimpanzee cell line in duplicate and mapped the data to the assemblies using two strategies (the approach we originally described, as well as the marker-assisted mapping strategy from Altemose et al., *Science*, 2022). Additionally, we determined the CpG methylation profile for each of the six sets of chimpanzee centromeres (from chromosomes 5, 10, 12, 20, 21, and X). Below, we show the CENP-A enrichment profiles and CpG methylation frequency for each of these centromeres (**Reviewer Figs. 3-5**). While we observed a clear dip in CpG methylation for each centromere (which typically indicates the likely site of the kinetochore), we found that the CENP-A chromatin domain does not coincide with this region. This was the case for nearly every centromere except for chromosome 12, where the CENP-A chromatin domain partially overlapped with the hypomethylated region (see **Reviewer Fig. 4a,b** below). For some centromeres

(e.g., chromosomes 10 and X), there was a clear CENP-A chromatin domain that was nearly 1000-fold enriched relative to bulk nucleosomal DNA yet did not coincide with the hypomethylated region. These results suggest three possibilities: 1) CENP-A has moved over time between the initial sequencing of the cell line and the CENP-A ChIP-seq experiment (approximately 2-3 weeks of cell culture); 2) mapping of short reads to diploid centromeres has additional challenges and will require either new mapping strategies or implementation of a long-read CENP-A mapping approach (e.g., DiMeLo-Seq); or 3) CENP-A does not coincide with CpG hypomethylation at the chimpanzee centromeres, indicating this is a true, biological result.

We note that the CENP-A antibody was optimized for use in human, and while there are now a few papers describing a correspondence between CENP-A ChIP-seq and CpG hypomethylation, it has not been extensively tested under different conditions. It is possible that the CENP-A chromatin peaks do not coincide with the hypomethylated region in chimpanzee, as it does in humans, as a result of epitope differences or modifications of the histone tail that are specific to NHPs. Because of this ambiguity, we plan to investigate this more deeply in a follow-up publication and take multiple complementary approaches to defining the site of the CENP-A chromatin domain and kinetochore across all primates studied here. We don't believe the comparative analysis of the CENP-A ChIP-seq experiment is central to the main message of the paper and will require many more months or years to resolve.

Reviewer Figure 3. CENP-A chromatin does not coincide with CpG hypomethylation in the chimpanzee centromeres from chromosomes 5 and 10. a-d) CpG methylation frequency and CENP-A enrichment profiles for both haplotypes from the **a,b)** chromosome 5 centromere and **c,d)** chromosome 10 centromere. The CpG methylation status was determined via Nanopolish using ONT reads >30 kbp long. The CENP-A chromatin profiles were mapped with two strategies: the original strategy detailed in our initial submission and a marker-assisted strategy with a *k*-mer size of 51 bp (Altemose et al., *Science*, 2022). Both mapping strategies show CENP-A chromatin enriched in regions outside of the hypomethylated region.

Reviewer Figure 4. CENP-A chromatin coincides with CpG hypomethylation in the chimpanzee chromosome 12 centromeres but not with the chromosome 20 centromeres. a-d) CpG methylation frequency and CENP-A enrichment profiles for both haplotypes from the **a,b)** chromosome 12 centromere and **c,d)** chromosome 20 centromere. The CpG methylation status was determined via Nanopolish using ONT reads >30 kbp long. The CENP-A chromatin profiles were mapped with two strategies: the original strategy detailed in our initial submission and a marker-assisted strategy with a k -mer size of 51 bp (Altemose et al., *Science*, 2022). Both mapping strategies show CENP-A chromatin enriched in the hypomethylated region on chromosome 12 but not on chromosome 20.

Reviewer Figure 5. CENP-A chromatin does not coincide with CpG hypomethylation in the chimpanzee centromeres from chromosomes 21 and X. a-d) CpG methylation frequency and CENP-A enrichment profiles for **a,b)** both haplotypes from the chromosome 20 centromere and **c,d)** the only haplotype from the chromosome X centromere. The CpG methylation status was determined via Nanopolish using ONT reads >30 kbp long. The CENP-A chromatin profiles were mapped with two strategies: the original strategy detailed in our initial submission and a marker-assisted strategy with a k -mer size of 51 bp (Altemose et al., *Science*, 2022). Both mapping strategies show CENP-A chromatin enriched in regions outside of the hypomethylated region.

Referee #3 (Remarks to the Author):

Centromeres have been largely excluded from genomic studies due to their extremely repetitive and complex sequence compositions, and the first fully resolved sequences of human centromeres have only recently become available. The authors of this manuscript present a second completely resolved set of human centromere sequences, together with completely resolved sequences for 6 chromosomes in 3 non-human primate species. These data enable the first high-resolution assessment of sequence variation in centromeres between human individuals by comparing the completely resolved complements in addition to partial complements from a diversity panel of other human genomes, and an assessment of the recent evolutionary history of primate centromeres. The authors detect large-scale variability in human centromeric sequences, including array sizes, array composition, and kinetochore positioning, as well as multiple new alpha-satellite HORs and a nearly complete turnover of alpha-satellite HORs in primates.

The study is well presented, and most technical aspects are, in my opinion, sound. It represents a significant addition to our understanding of human centromere biology and the results are likely to be of substantial interest to the human genetics and comparative genomics community. I do however have several questions and comments I hope the authors could address (not in order of importance).

I strongly encourage the authors to reconsider some of their coloring schemes, particularly those used to classify alpha-sat HOR-mers. The current scheme uses repeated or very similar divergent colors for different parts of what is a continuous scale. In Fig 1. it is not possible to distinguish between 6/11/17/18/24-mers or 5/14/15-mers, among others. If there is a requirement to use this scale, it is not evident from the text.

Thank you for drawing our attention to this. To make the colors of the HORs more distinguishable, we changed 8 out of 23 of them (34.8%) in **Fig. 1** (see below). The colors that we changed correspond to the following HORs: 3-, 11-, 13-, 14-, 15-, 17-, 18-, 21- and 24-mer, and we have propagated these changes to all relevant figures throughout the manuscript.

New Fig. 1:

Old and new colors for the 6/11/17/18/24-mers and 5/14/15-mers:

Several plots in the extended data are too cramped to make the visualizations useful. I understand the wish to include all chromosomes on the same page, but the subpanels end up being too small to be readable, despite their importance for the manuscript's conclusions. Examples include Ext. Figs 4,5,7,11.

We have expanded and enlarged each of the subpanels for **Extended Data Figs. 4, 5, 6, 7, 9, 10, 11, and 14**, and these are now included as new **Supplementary Figs. 1-3 and 6-62**. Below, we list the corresponding Extended Data and Supplementary Figures.

Extended Data Figure	Supplementary Figures
4	1-3
5	6-8
6	9-10
7	11-12
9	13-14
10	15-16
11	17-39
14	40-62

We have also kept the original Extended Data Figures with all chromosomes on a single page for those who would prefer to view all of them at once.

Page 3, Line 107: The authors use several lines of evidence to ensure that potential somatic alterations in CHM1 do not generate issues with their downstream analyses. They show that a significant fraction of cells contains an aberrant karyotype and several genomic rearrangements, which is not evident from the main text. Given that at least for the tetraploid CHM1 cells chromosomal segregation did at some point not function correctly and the observed rearrangements contain deletions of tumor suppressors, it seems this might be an important caveat regarding the representative nature of the assemblies. It is furthermore not clear why the same degree of caution was not necessary for the NHP cell lines, which are at least in part EBV immortalized and thus potentially subject to the same issues.

This is a fair point. Complete hydatidiform moles are, in fact, aberrations of development that are subsequently passaged after hTERT transformation to generate the large number of cells and DNA needed to generate T2T or near-T2T genomes (in the case of both CHM13 and CHM1). This is why we performed extensive QC of all possible rearrangements. Although not mentioned here, we also characterized all large CNVs (>50 kbp) within the euchromatic position of genome and confirmed that >95% of these were also identified in at least one other human sample (n=2000 samples tested), suggesting that most CHM1 CNVs are of germline origin and present in another human sample. Of course, characterizing the centromere organization is largely *terra incognita*, and that is why the analysis of the 56 human incomplete centromeres was critical. The discovery that about half of the

centromeres were a better fit to CHM1 than the current complete reference CHM13 argues that these are generally representative. We have added the following caveat to the main text, which we agree is important (changes underlined):

“In fact, we find that 46.9% of these haplotypes are a better match to CHM1 than to CHM13 (**Extended Data Table 2, Extended Data Fig. 7**). While the data support the biological relevance of CHM1 centromeres (similar to the T2T-CHM13 centromeres), both genomes are aberrations of normal development followed by cell culture propagation. Thus, caution should be taken until all structures and configurations have been confirmed in additional human samples.”

With respect to the nonhuman primates (NHPs), we believe the situation is sufficiently different for three reasons. Each sample has been extensively characterized previously (as these were previous reference genomes) and the genomes, including the karyotypes, were deemed to be representative of the species. Second, few of these were actually transformed (nonhuman apes do not transform as readily with EBV), so most of the source material is either fibroblast or other primary culture with a limited number of passages when compared to CHM1 and CHM13. Third, the NHP samples were diploid. This gave us the advantage of seeing two versions of each centromere haplotype, which we strove to complete and present for each of the five chromosomes studied in this paper. Consistency in structure and organization between the two NHPs provided, in essence, a replicate of the biological significance of the structural differences.

P3 L109 / Ext. Fig. 4: The figure legend claims “uniform read depth, indicating a lack of large structural errors” and states two exceptions. This statement is insufficiently backed up by the figures, and gauging by eye there are several more, although the figure is difficult to see due to its small size. Examples include chr3 Mb 4-8.5 (PB), chr3 Mb ~2.5-3 (ONT), chr1 Mb ~5.5-6 (PB), chr8 Mb ~2.3-2.8 (ONT), among several other peaks and dips. Could the authors please produce read-depth histograms for these regions and specify what size cutoff they consider ‘large-scale’? I would also be interested in reading their thoughts for the underlying reasons, particularly for the event on chr3. All comments equally apply to the NHP data presented in Ext Figs 15-16, which show similar patterns.

To address this comment, we generated read-depth histograms of both PacBio HiFi and ONT data for all CHM1 centromeric regions (new **Supplementary Figs. 4 and 5**, below). We find that there are six CHM1 centromeres with increased or reduced coverage in PacBio HiFi and/or ONT data (chromosomes 1, 3, 8, 13, 16, and 17). Two of these are due to sequencing biases in PacBio chemistry, which results in increased coverage of HSat2 sequences (chromosome 1) or reduced coverage of HSat1A sequences (chromosome 3). Both of these sequencing biases were previously described in Altemose et al., *Science*, 2022, and we now cite this in the legend of **Extended Data Fig. 4** and **Supplementary Fig. 4**. Two other centromeres (chromosomes 13 and 17) have reduced PacBio and ONT coverage due to a deletion in sequence in a subset of cells, which we described in **Supplementary Notes 1 and 2**. Finally, three other centromeres (chromosomes 3, 8, and 16) have an increase in only ONT coverage but not PacBio HiFi coverage. We speculate that this may indicate a possible collapse in sequence in these centromeres. It is possible that there are smaller (~20 kbp) contigs present in the whole-genome assembly that are not incorporated into the centromere assemblies, and the PacBio reads are mapping to these shorter contigs, but the ONT reads are mapping to the centromere assembly due to alignment length. Because of this, we now describe these three centromeric regions as harboring potential assembly errors in the legends of **Extended Data Fig. 4** and **Supplementary Figs. 4 and 5**. Importantly, none of these regions are the site of

hypomethylation or CENP-A chromatin enrichment and are not thought to contribute to kinetochore assembly.

Supplementary Figure 4. PacBio HiFi and ONT read-depth histograms for CHM1 chromosome 1-12 centromeres. a-l) Histograms of the PacBio HiFi (top) and ONT (bottom) read depths across the

CHM1 chromosome 1-12 centromeres. While most of these distributions are consistent with Poisson sampling, we identify three centromeres with increased or reduced coverage in PacBio HiFi and/or ONT data (chromosomes 1, 3, 8). Two of these are due to sequencing biases in PacBio chemistry (Altemose et al., *Science*, 2022), which results in increased coverage of HSat2 sequences (chromosome 1) or reduced coverage of HSat1A sequences (chromosome 3). However, we also identify increased coverage of ONT data on the centromeres from chromosomes 3 and 8, which may indicate a possible collapse in sequence in these centromeres that is detected with longer ONT reads but not with shorter PacBio HiFi reads. Importantly, neither of these regions are the site of hypomethylation or CENP-A chromatin enrichment and are not thought to contribute to kinetochore assembly.

Supplementary Figure 5. PacBio HiFi and ONT read-depth histograms for CHM1 chromosome 13-22 and X centromeres. a-l) Histograms of the PacBio HiFi (top) and ONT (bottom) read depths across the CHM1 chromosome 13-22 and X centromeres. Most of these distributions are consistent with Poisson sampling. However, we identified three centromeres with increased or reduced coverage in PacBio HiFi and/or ONT data (chromosomes 13, 16, 17). Two of these (chromosomes 13 and 17)

have reduced PacBio and ONT coverage due to a deletion in sequence in a subset of cells (**Supplementary Notes 1 and 2**). However, we also identify increased coverage of ONT data in the chromosome 16 centromere, which may indicate a possible collapse in sequence that is detected with longer ONT reads but not with shorter PacBio HiFi reads. Importantly, none of these regions are the site of hypomethylation or CENP-A chromatin enrichment and are not thought to contribute to kinetochore assembly.

In addition, we performed this analysis for the other human (HG00733) and NHP centromeres and observed typical read depth distributions for all centromeres, indicating they are largely free of sequencing chemistry biases and large assembly errors (see new **Supplementary Figs. 63 and 64**, below). For human chromosome 21, orangutan chromosome 12, and macaque chromosome 21, which have lower coverage, we note that this is due to a smaller region being assessed as a result of either a smaller α -satellite HOR array or inactivated centromeric region.

Human

Chimpanzee

Supplementary Figure 63. PacBio HiFi and ONT read-depth histograms for human and chimpanzee centromeres from chromosomes 5, 10, 12, 20, 21, and X. a-l) Histograms of the PacBio HiFi (top) and ONT (bottom) read depths across the human (HG00733) chromosome a) 5, b) 10, c) 12, d) 20, e) 21, and f) X centromeres and the chimpanzee chromosome g) 5, h) 10, i) 12, j) 20, k) 21, and l) X centromeres. All read-depth distributions are consistent with Poisson sampling, with no significant outliers. We note that the human chromosome 21 centromere has lower coverage due to a smaller region being assessed as a result of a smaller α -satellite HOR array.

Orangutan

Macaque

Supplementary Figure 64. PacBio HiFi and ONT read-depth histograms for orangutan and macaque centromeres from chromosomes 5, 10, 12, 20, 21, and X. a-l) Histograms of the PacBio HiFi (top) and ONT (bottom) read depths across the orangutan chromosome a) 5, b) 10, c) 12, d) 20, e) 21, and f) X centromeres and the macaque chromosome g) 5, h) 10, i) 12, j) 20, k) 21, and l) X centromeres. All read-depth distributions are consistent with Poisson sampling, with no significant outliers. We note that the orangutan chromosome 12 centromere and macaque chromosome 21 centromere have lower coverage due to inactivation of that centromere and, consequently, a smaller region being assessed.

P3 L126 / Ext. Data Table 3: There are substantial differences between, but also within the same alignment strategy when switching query and reference for both identity and alignability. As several important subsequent results are based on this, could the authors please justify their choice of alignment strategy? Why was the tandem-repeat-aware strategy dropped? Please also clarify how the sequence identity was calculated, as the choice of denominator can influence these values significantly.

Most, if not all, of the differences in alignability and sequence identity when switching reference and query are due to natural, biological differences in the α -satellite HOR array that affect the alignments independently of the strategy used. To demonstrate this, take the chromosome 21 centromere, for example. This centromere has a 343 kbp α -satellite HOR array in the CHM13 genome, but it is ~3.6-fold larger in the CHM1 genome (1.23 Mbp). This difference in size affects the amount of sequence that can be aligned to this centromere, depending on which genome is the reference vs. the query. For example, when CHM13 is the reference (smaller centromere), we find that 96.3%, 93.1%, and 87.4% of the CHM1 sequences align for each of the three strategies, respectively. However, when CHM1 is the reference (larger centromere), we find that only 26.6%, 25.2%, and 30.12% of the CHM13 sequences align. This difference in alignability is mainly due to the fact that the CHM1 sequences are only ~28% as abundant as the CHM13 sequences due to the size of the α -satellite HOR array. When comparing the sequence identity when switching reference and query, we observe a much smaller effect (99.0% vs. 98.9% for Strategy #1, 99.3% vs. 99.1% for Strategy #2, and 99.7% vs. 97.9% for Strategy #3, when CHM13 vs. CHM1 is the reference, respectively). **We note that in all three strategies, the sequence identity for all centromeres when switching reference and query is nearly identical (98.6% for both CHM13 and CHM1 in Strategy #1, 99.1% vs. 99.0% for CHM13 vs. CHM1 for Strategy #2, and 98.9% vs. 99.0% for CHM13 vs. CHM1 for Strategy #3).** Thus, the differences can be attributed to the variation in size and structure of the α -satellite HOR array that affect the alignments independently of the strategy used.

We chose to use the first alignment strategy for most of our analyses because this strategy is sequence-independent and can be applied to both repetitive and non-repetitive sequences. This is in contrast to TandemAligner, which is only optimized for tandem repeats and precludes comparisons to non-repetitive sequences (such as those flanking the α -satellite HOR array), which we used in our phylogenetic and mutational analyses. Additionally, the first alignment strategy is permissive to changes in local structure by allowing for both gapped alignments and many-to-one alignments from the same contig, accounting for local deletions and expansions within the region independent of contig length. This is in contrast to the second alignment strategy, which calculates sequence identity from 10 kbp alignments and is less permissive to changes less than 10 kbp in length.

To calculate the sequence identity (as shown in **Extended Data Table 3**), we first partitioned the alignments into 10 kbp non-overlapping windows in the reference genome and then parsed the CIGAR string using a custom python script (https://github.com/glogsdon1/centromere_variation_and_evolution_scripts/blob/main/cigar_parser.py). This script calculates the percent sequence identity within each 10 kbp alignment window “by event”. An “event” can be a mismatch, insertion, or deletion, and they are each counted once even if they occur consecutively. In the script, we calculate percent sequence identity by event as:

$$\frac{(\# \text{ of matches})}{(\# \text{ of matches} + \# \text{ of mismatches} + \# \text{ of insertion events} + \# \text{ of deletion events})}$$

Thus, the denominator for the sequence identity calculation is: (# of matches + # of mismatches + # of insertion events + # of deletion events), with each consecutive set of mismatches, insertion events, and deletion events calculated as one event. For example, a deletion of three consecutive bases would be considered one event, instead of three. Therefore, the maximum a denominator can be is 10 kbp. We now describe this calculation in the Methods (changes underlined):

“We subsequently partitioned the alignments into 10-kbp non-overlapping windows in the reference genome (either CHM1 or CHM13) and calculated the mean sequence identity between the pairwise alignments in each window with the following formula: $(\# \text{ of matches}) / (\# \text{ of matches} + \# \text{ of mismatches} + \# \text{ of insertion events} + \# \text{ of deletion events})$. We, then, averaged the sequence identity across the 10-kbp windows within the α -satellite HOR array(s), monomeric/diverged α -satellites, other satellites, and non-satellites for each chromosome to determine the mean sequence identity in each region.”

P4 149: I'm confused by this sentence. Are the authors suggesting the numbers presented in the preceding sentences do not reflect allelic variation?

We believe that all estimates represent allelic patterns of variation, but the tempo of substitution depends on the sequence context and whether it resides within heterochromatin or euchromatin. The data suggest a gradient of increasing divergence as one moves from euchromatic DNA to pericentromeric satellites and then to higher-order α -satellite defining the heterochromatin. The challenge for the latter is defining alleles in the context of α -satellite HOR turnover, and that is why we were careful to state the mean sequence identity of the alignable portions of CHM1 and CHM13. We have revised this sentence as follows (changes underlined):

“For example, the mean sequence identity for the alignable portions of CHM1 and CHM13 α -satellite HOR arrays is $98.6 \pm 1.6\%$, in contrast to monomeric/diverged α -satellites at $99.8 \pm 0.4\%$ and other pericentromeric satellite DNA (β -satellite, γ -satellite, and human satellites) at $99.1 \pm 1.5\%$ (**Extended Data Table 4, Extended Data Fig. 8**). Extending further into the non-satellite pericentromeric DNA, the sequence identity begins to approximate rates of allelic variation corresponding to the euchromatic portions of the genome ($99.9 \pm 0.3\%$; **Extended Data Table 4, Extended Data Fig. 8**).”

Fig 2a: Please increase the dot sizes, it is currently not possible to distinguish the underlying colors in many cases.

Thank you for this suggestion. We have now increased the dot sizes in this figure by 1.5-fold and propagated all changes to the other relevant figures (**Extended Data Figs. 6 and 9**, and **Supplementary Figs. 7, 8, 10, and 11**). We provide the updated **Fig. 2a** below for convenience:

P4 L159 / Fig2b-c / Ext. Fig 7 / Ext. Fig 10: Do the observed patterns of divergence & diversity on the centromeres recapitulate what is known about the population divergence of these 56 samples based on other markers? While the current analyses of human diversity observing higher sequencing identity for either CHM1 or CHM13 and diversity of array lengths underline the extreme variation, it feels like a missed opportunity to better understand the sources and stratification of variability (particularly beyond the subdivision into “African” and “non-African” haplotypes).

These are great questions, but the patterns appear much more complex and the sample size, at present, is too limited to properly address this with the current cohort of individuals. Unlike unique regions of the genome where the patterns of diversity are much more uniform across different regions, we note considerable diversity among human centromeres—with some showing much greater diversity than others (**Extended Data Fig. 10** and **Supplementary Figs. 13 and 14**). In general, some of the most diverse centromeres (or at least the most divergent when compared to CHM1 and CHM13) do correspond to individuals of African ancestry. This is, however, not always the case. Our phylogenetic analysis of chromosome 11 and 12 centromeres is instructive in this regard (see **Fig. 7**). For chromosome 12, for example, the first branch places a subset of African haplotypes as the outgroup to the most abundant monophyletic clade at ~667 kya. This is contrast with chromosome 11, where the corresponding position is occupied by a set of CEN-haplotypes of diverse ancestry (three Amerindian, one European, and one African). In this case, the increase in diversity appears to be driven by the emergence of a new α -satellite HOR variants (as opposed to ancestry). While it would be tempting to draw some conclusions, the other important consideration is the uneven sampling of the current population of cenhaps. They are primarily of African and Amerindian descent with three individuals of East Asian, two individuals of European, and one of South Asian. Any population diversity conclusion at this time we believe would be too premature.

For Fig 2b / Ext. Fig 10 could the authors please include information on the proportion of the query that aligns?

Yes, we previously included this information in **Extended Data Fig. 7** and **Extended Data Table 6**. In **Extended Data Fig. 7**, you can see the portion of unaligned sequences (black) vs. aligned sequences (pink or yellow). We have enlarged these plots to improve readability, and they are now included in

Supplementary Figs. 9 and 10 (shown below for convenience). We also provide precise quantifications of the portion of unaligned vs. aligned sequences for these 56 genomes compared to CHM1 and CHM13 in **Extended Data Table 6**. We now refer to this table in the figure legends for **Extended Data Fig. 7** and **Supplementary Figs. 9 and 10** (changes underlined).

Supplementary Figure 9. Variation in the sequence and structure of chromosome 1-10 centromeric α -satellite higher-order repeat (HOR) arrays among 56 diverse human genomes. Plots showing the percent sequence identity between centromeric α -satellite HOR arrays from CHM1 (y-axis), CHM13 (x-axis), and 56 other diverse human genomes [generated by the Human Pangenome Reference Consortium (HPRC)⁹ and Human Genome Structural Variation Consortium (HGSCV)¹⁰]. Each data point shows the proportion of aligned bases from each human haplotype to either the CHM1 (left) or CHM13 (right) α -satellite HOR array(s). The proportion of unaligned bases is shown in black. The size of each data point corresponds to the total proportion of aligned bases among the CHM1 and CHM13 centromeric α -satellite HOR arrays. Precise quantification of the sequence identity and proportions of aligned versus unaligned sequences is provided in Extended Data Table 6.

Supplementary Figure 10. Variation in the sequence and structure of chromosome 11-22 and X centromeric α -satellite HOR arrays among 56 diverse human genomes. Plots showing the percent sequence identity between centromeric α -satellite HOR arrays from CHM1 (y-axis), CHM13 (x-axis), and 56 other diverse human genomes (generated by the HPRC⁹ and HGSC¹⁰). Each data point shows the proportion of aligned bases from each human haplotype to either the CHM1 (left) or CHM13 (right) α -satellite HOR array(s). The proportion of unaligned bases is shown in black. The size of each data point corresponds to the total proportion of aligned bases among the CHM1 and CHM13 centromeric α -satellite HOR arrays. Precise quantification of the sequence identity and proportions of aligned versus unaligned sequences is provided in Extended Data Table 6.

Ext. Fig7 is not readable at the current size, do these fractions makes sense considering what is known about the ancestries for CHM1/CHM13?

To make **Extended Data Fig. 7** more readable, we have broken it out onto two pages (**Supplementary Figs. 9 and 10**; shown above).

CHM1 and CHM13 are both known to be primarily of European ancestry (Vollger et al., *Nature*, 2023, and Nurk et al., *Science*, 2022), so the observations shown in **Extended Data Fig. 7** (and **Supplementary Figs. 9 and 10**) that a subset of African samples carry much more diverse centromeres is consistent (however, we note that that observation would be true for comparisons involving any out-of-African population and is not specific to Europeans). It should also be noted that current coalescent trees that we investigated (shown in **Fig. 7** and **Extended Data Figs. 18-20**) tend to suggest very deep origins for the centromere α -satellite predating, by far, continental grouping of humans. Thus, much larger sampling of complete human centromeres will be needed to determine how haplotype structures in centromeres compare with the diversity reported in euchromatic regions.

For Fig. 2c. have the authors explored sources of length variation? Given the extreme differences, is there any evidence that this might encompass not only germline but also somatic variation?

We assume that most of the variation is originating in the germline and not somatically for a few reasons. First, sequence and assembly from cell culture, which represents a population of cells with rare exceptions, reveals one predominant haplotype in the case of a haploid source or two predominant haplotypes (in the case of the NHP diploid samples), upon which all reads from either PacBio HiFi or ONT map back to with consistent coverage. If there was extensive variation, this should be reflected in the alignment of the sequencing reads back to the assemblies. Second, haplotype reconstructions of different human centromeres show that the longer haplotypes tend to cluster on the same or related haplotype due the emergence of new α -satellite HORs. This argues transmission of the changes through the germline. Finally, although not part of this paper, we have also surveyed parent–child transmissions, and the centromere haplotypes remain remarkably consistent if not perfectly identical. All of these suggest that most variation is of germline origin.

P5 L174: Could the authors please comment/hypothesize on the underlying reasons for mostly larger centromeres (and often significantly so) in CHM1 vs CHM13?

This is a good point and something we noted initially as well, though we believe this may simply represent a sampling and ascertainment bias. First, we find that 9 out of 23 α -satellite HOR arrays are larger in CHM13 than in CHM1, while 16 out of 23 arrays are larger in CHM1 than CHM13 (**Figs. 2c, 3a, and Extended Data Table 7**). Second, the difference in terms of fold-length between the CHM1 and CHM13 α -satellite HOR arrays is well within the range we observed from the 56 human samples, which almost always exceed the CHM1-CHM13 differences (**Fig. 2c**). Third, if we assume that there is some minimum length an α -satellite HOR array must be for it to be functional, differences less than 1 will begin to asymptote. Thus, the differences in the lengths of the CHM1 and CHM13 α -satellite HOR arrays are not exceptional and are within the expected range based on our comparative analyses.

P6 L253: Please clarify the criteria used to select the specific chromosomes. Does this set encapsulates the previously described patterns of diversity observed across human centromeres?

We selected these specific chromosomes (5, 10, 12, 20, and 21) because they represent different types of changes we had observed, including dramatic changes in length, replacement of α -satellite HORs, and changes in the potential site of the kinetochore. The X chromosome served as a control of the lower bound because of its lower mutation rate. We think it would be premature to suggest that we have adequately sampled diversity, but short of sequence and assembly of all centromeres from multiple individuals, we believe this is a reasonable place to begin. We have modified the text and made it clear that more work needs to be done (changes underlined):

“Our analyses (**Figs. 1-4**) revealed that human centromeres vary non-uniformly depending on the chromosome. In particular, specific human chromosomes show either highly variable α -satellite HOR array lengths (e.g., chromosome 21), diverse α -satellite HOR organizations (e.g., chromosomes 5, 10, and 12), or divergent epigenetic landscapes (e.g., chromosome 20). In contrast, the X chromosome is among the most conserved, with nearly identical sequences and structures among diverse human genomes (**Extended Data Fig. 11**). These findings imply that centromeres may have different mutation rates and diverse evolutionary trajectories that shape their variation. To test this hypothesis, we sequenced and assembled orthologous centromeres from four primate species, focusing on the completion of these six centromeres, in an effort to reconstruct their evolutionary history over a 25-million-year window of primate evolution. Each were specifically selected because they represent different forms of centromeric diversity (as described above), but additional analyses, such as sampling all centromeres across multiple individuals, will need to be done to fully assess complete diversity. To assemble these centromeres, we first generated PacBio HiFi data...”

P7 L292: Could the authors please provide more details on how this inversion was detected, as there is no mention in the methods? Is this based solely on the orientation of the satellite sequence, and were there any orthogonal analyses?

This inversion was detected with the HumAS-HMMER tool (https://github.com/fedorrik/HumAS-HMMER_for_AnVIL), which classifies α -satellite monomers into SFs and reports their strand orientation. We confirmed the presence of this inversion with StringDecomposer (<https://github.com/ablab/stringdecomposer>), which is another tool that also classifies α -satellite monomers and indicates their orientation, and we validated this inversion by mapping native ONT reads >30 kbp long to the assembly, which show even coverage across the breakpoints as well as a flip in orientation of α -satellite monomers in the raw reads at the breakpoints in this centromere. We provide a figure below (**Extended Data Fig. 19**), which shows the even ONT read coverage over the inversion breakpoints and support by the raw ONT reads for this structure.

Extended Data Figure 19. Detection of a 3.2 Mbp polymorphic inversion in the orangutan chromosome 20 centromere. **a)** Location of a 3.2 Mbp inversion in the orangutan chromosome 20 centromeric α -satellite HOR array in haplotype 2 (H2). This inversion is located at bases 2,097,123-5,280,214 and was detected with both HumAS-HMMER and StringDecomposer. **b,c)** Uniform coverage of orangutan ONT reads >30 kbp long across the **b)** upstream and **c)** downstream inversion breakpoints supports this structural variant.

Additionally, we revised this section in the Methods to provide more details on our analysis (changes underlined):

“Human and NHP α -satellite suprachromosomal family (SF) classification and strand orientation analysis

To determine the α -satellite SF content and strand orientation of human and NHP centromeres, we ran HumAS-HMMER ([https://github.com/fedorrik/HumAS-HMMER for AnVIL](https://github.com/fedorrik/HumAS-HMMER_for_AnVIL)) on centromeric contigs with the following command: `hmmmer-run SF.sh {path to directory with fasta} AS-SFs-`

hmm3.0.290621.hmm {number_of_threads}. This generated a BED file with the SF classification and strand orientation of each α -satellite monomer, which we visualized with R⁵⁶ (v1.1.383) using the ggplot2 package⁵⁷. In cases where an inversion was detected, we ran StringDecomposer⁶³, a tool that detects and reports changes in orientation of tandem repeats, using the default parameters to confirm the presence of reoriented α -satellite monomers at the breakpoints. Finally, we validated the presence of the inversion by aligning native ultra-long ONT reads to the assemblies as described above and confirming even coverage across the breakpoints as well as the presence of inverted α -satellite monomers in the aligned reads.”

P8 L324: The presented model assumes that mutations accumulate at an equal rate on both branches of each human-NHP pair. The authors need to more carefully consider whether this assumption is justified, especially given the extreme variability they observe. It also would be helpful to see the distributions of μ stratified by species and sequence category, as these differences are difficult to gauge from the scatterplots only.

While it certainly is true that saltatory mutations do occur over short periods, these are largely restricted to α -satellite HORs. We believe that equal rates of mutational change are the most reasonable starting point for the monomeric α -satellite sequences in the flanking regions because they align well among primate lineages with an equivalent number of lineage-specific changes and follow a molecular clock, albeit at an elevated tempo. We are careful when estimating mutational changes among the α -satellite HORs to limit it to those haplotypes that are most closely related based on their haplotype. Here, we clearly demonstrate jumps in the mutation rate but stop short of computing genome-wide rates or even centromere rates other than the subset where genetic changes can be deduced. We attempted to make this clearer in the Discussion (changes underlined):

“For example, in the case of chromosome 12, we estimate the new HORs emerged approximately 13-23 kya (thousand years ago; **Fig. 7b**), while for chromosome 11, they emerged approximately 80-153 kya (**Fig. 7c**). This suggests a single origin for the new α -satellite HORs, followed by the saltatory spread of >1 Mbp of new HORs to this subset of human haplotypes. Because we are specifically selecting haplotypes that show a saltatory amplification of α -satellite HORs, these rate estimates should not be considered genome- or even centromere-wide rates of change.”

We also modified the legend of **Fig. 6** and **Extended Data Fig. 17** to make it clear that it is impossible to compute μ between species for the α -satellite HOR due to nearly complete evolutionary turnover (changes underlined):

“Figure 6. Centromeres evolve with different evolutionary trajectories and mutation rates.
a-c) Phylogenetic trees of human, chimpanzee, orangutan, and macaque α -satellites from the higher-order and monomeric α -satellite regions of the chromosome 5, 12, and X centromeres, respectively.
d-f) Plot showing the mutation rate of the chromosome 5, 12, and X centromeric regions, respectively. Individual data points from 10-kbp pairwise sequence alignments are shown. We note that the regions corresponding to the active α -satellite HORs have only approximate mutation rates based on human–human comparisons. Due to unequal rates of mutation and the emergence of new α -satellite HORs, interspecies comparisons are not possible in these regions.”

P8 L345: Please include a distance metric to understand what “remarkably similar” corresponds to, and visually connect corresponding haplotypes in Fig. 7b-c. Do nodes without annotated bootstrap support values (assuming that is what the number corresponds to) have full support? What do the asterisks correspond to?

We have now visibly connected the p- and q- haplotypes in Fig. 7b,c with light teal bars as schematized in Fig. 7a (see below) and have propagated this change to **Extended Data Figs. 18-20** as well. The asterisks do indicate 100% bootstrap support, and nodes with 90-99% bootstrap support are marked numerically. Nodes without an asterisk or number have a bootstrap support <90%. We have revised the legends of Fig. 7 and **Extended Data Figs. 18-20** to reflect this (changes underlined).

Figure 7. Phylogenetic reconstruction of human centromeric haplotypes and the saltatory amplification of new α -satellite HORs. a) Strategy to determine the phylogeny and divergence times

of completely sequenced centromeres using monomeric α -satellite or unique sequence flanking the canonical α -satellite HOR array from both the short (p) and long (q) arms of chromosomes 11 and 12. Chimpanzee is used as an outgroup with an estimated species divergence time of 6 million years ago. **b,c)** Maximum-likelihood phylogenetic trees depicting the p- and q-arm topologies along with the estimated divergence times reveals a monophyletic origin for the emergence of new α -satellite HORs within the **b)** chromosome 12 (*D12Z3*) and **c)** chromosome 11 (*D11Z1*) α -satellite HOR arrays. These arrays show a complex pattern of new α -satellite HOR insertions and deletions over a short period of evolutionary time. Asterisks indicate nodes with 100% bootstrap support, and nodes with 90-99% bootstrap support are indicated numerically. Nodes without an asterisk or number have bootstrap support <90%. The haplotypes from the p- and the q-arm trees are linked with a light teal bar, as schematized in panel a. We note that most differences in the order of the haplotypes occur at the terminal branches where the order of sequence taxa can be readily reshuffled to establish near-complete concordance. Thus, there are no significant changes in the overall topologies of the phylogenetic trees.

We have not computed an official distance metric because many of the sequences belong to monophyletic clades where the order can be easily reshuffled among the termini, but the number of genomes being compared is still relatively modest, and we don't think a calculation at this time would add value. Instead, we have removed "remarkably" and simply refer them as "similar" in the main text.

P8 L334: I'm surprised by the sparsity of comments on the divergence dating for different haplotypes the authors present in Fig. 7b-c and Ext Figs 18-20, many of which exceed the coalescent times for AMH. Am I correctly interpreting that the authors suggest that e.g. on the chr13 p-arm the CHM13 haplotype diverged from other ones 5.2 Mya (<1Mya after the chimp divergence used as calibration) ago and has been maintained ever since? If so, these results are very surprising even for the relatively young dates. Can the authors please clarify how exactly these dates were calculated, as this is not evident from the methods (beyond using chimp to calibrate)? If a simple molecular clock scaled to the human-chimp divergence was used, the authors need to consider whether this is adequate in the face of the extreme variability of sequence, structure, and mutation rates they claim. Are the authors surprised about the tree topology given what we know about the history of the underlying superpopulations, and if not why?

The majority of the coalescent times that we calculated fall well within the expected range of 600-800 kya (Vollger et al., *Nature*, 2023), with most of the coalescents preceding radiation of human continental groups, although we agree there are some interesting outliers. Owing to the lack of recombination in the region, some deeper coalescents might be anticipated. It is noteworthy, for example, that the chromosome 17q21.31 inversion polymorphism, which occurred once in human history, has an estimated coalescent time of approximately 2.3 million years (Steinberg et al., *Nat Genetics*, 2012). We appreciate, however, that the example of the chromosome 13p arm is, in fact, exceptional—something we had not previously considered in detail. It is possible that this is a consequence of ectopic recombination among acrocentric short arms recently described by Guarracino et al., *Nature*, 2023. If such non-homologous exchange events extend to the centromere, it would potentially allow centromeres to be distributed among one of five different short arms in humans, at least for the p-arms. This would, in essence, increase the effective population size of this by fivefold, leading to, in principle, much deeper coalescence. While this is speculative, it is interesting that the p-arm and q-arm topology, unlike other autosomes we have investigated, is very discordant and the

composition of the α -satellite HOR is an outlier for other chromosome 13 centromeres. In response, we have added a note to the legend in **Extended Data Fig. 20**:

“In the case of the chromosome 13 p-arm (panel a), the CHM13 divergence time is exceptional (5.2 mya) compared to all other regions of the genome. The basis for this is unknown, but it may reflect ectopic exchange of the p-arm of human acrocentric chromosomes, leading to non-homologous exchange among five human chromosomes (Guarracino et al., *Nature*, 2023).”

We have also added a description to the Methods of how these coalescent estimates were calculated (changes underlined):

“To determine the phylogenetic relationship and divergence times between centromeric regions from chromosomes 5, 7, and 10-14 in the CHM1, CHM13, and 56 other diverse human genomes (sequenced and assembled by the HPRC⁷ and HGSC¹⁸), we first identified contigs with complete and accurately assembled centromeric α -satellite HOR arrays, as determined by RepeatMasker⁵⁴ (v4.1.0) and NucFreq¹⁷ analysis. Then, we aligned each of these contigs to the T2T-CHM13 reference genome³ (v2.0) via minimap2³⁹ (v2.24). We also aligned the chimpanzee whole-genome assembly to the T2T-CHM13 reference genome³ (v2.0) to serve as an outgroup in our analysis. We identified 20-kbp regions in the flanking monomeric α -satellite or unique regions on the p- or q-arms and ensured that the region we had selected had only a single alignment from each haplotype to the reference genome. Then, we aligned these regions to each other using MAFFT^{64,65} (v7.453) and the following command: `mafft -auto -thread {num_of_threads} {multi-fasta.fasta}`. We used IQ-TREE⁶⁶ (v2.1.2) to reconstruct the maximum-likelihood phylogeny with model selection and 1000 bootstraps. The resulting tree file was visualized in iTOL⁶⁷. Timing estimates were calculated by applying a molecular clock based on branch-length distance to individual nodes and assuming a divergence time between human and chimpanzee of 6 mya. Clusters of α -satellite HOR arrays with a single monophyletic origin were assessed for gains and losses of α -satellite base pairs, monomers, HORs, and distinct structural changes manually.”

P8 L334: The ML tree estimates are sensitive to the alignment quality and correctly established orthology, which is challenging in the face of repetitive regions. Can the authors please clarify the chosen model and present statistics on missing data (gaps) for individuals and alignment columns? Were these alignments filtered in any way prior to the tree inference? The same comment applies to the results presented in Fig. 6a-c and corresponding Ext. Figs. Was Tamura-Nei chosen after testing its adequacy for this data?

Given the difficulty in aligning repetitive regions, we were careful to restrict our analysis to regions or haplotypes where orthology was the least unambiguous. For example, our ML trees were constructed from monomeric α -satellite flanking the α -satellite HOR array, which could be unambiguously aligned as part of the MSA [i.e., these regions can often be extended back to both ape and OWM (~25 mya)]. Second, when dealing with α -satellite HOR differences, we restricted our analysis to haplotypes where the topology was consistent with a monophyletic origin and, thus, more likely to be more recently descended from a common ancestor.

Tamura and Nei's 1993 model is a time-reversible model that corrects for nucleotide frequency differences, transition:transversion biases, and variation of substitution rate among different sites. We made the choice to use this model because we determined previously during our analysis of the

chromosome 8 centromere that testing different substitution models had only a modest effect (i.e., different substitution models gave slightly different results but were compatible with respect to magnitude; Logsdon et al., *Nature*, 2021). This was consistent with a study by Abdadi and colleagues that showed that varying models had limited consequence in changing phylogenetic inferences (Abadi et al., *Nat Commun*, 2019). Thus, because of earlier findings with chromosome 8, we opted for continued use of the Tamura-Nei model in this study.

With respect to gaps and alignment, statistics were only applied to aligned sequence (i.e., gaps were not considered). If there is no alignment, there will be no data to analyze. Some species have a deletion that prevents data from aligning to the reference, which is why there's a gap in that region.

Reviewer Reports on the First Revision:

Referees' comments:

Referee #1 (Remarks to the Author):

Thank you - the authors have very thoroughly and clearly addressed my points. Congratulations on this fascinating study. Ian Henderson

Referee #2 (Remarks to the Author):

The authors have attended to my points. Thanks for the nice Immuno-FISH experiments. I also thank the authors for the additional ChIP-seq experiments, I agree this will take very long and it is not essential to this story. Best wishes, André Marques

Referee #3 (Remarks to the Author):

I thank the authors for providing responses and clarifications to my initial comments and question, all of which are adequately addressed. I have no further comments for the revised version of their manuscript.